# Nucleophosmin supports WNT-driven hyperproliferation and tumor initiation

Georgios Kanellos [1] ✉, Chiara Giacomelli[1], Alexander Raven[1], Nikola Vlahov[1], Hu Jin [2], Pauline Herviou [1], Sudhir B. Malla [1,3], Nadia Nasreddin [1], Patricia P. Centeno [1], Constantinos Alexandrou[1], Kathryn Gilroy [1], Rachel L. Baird[1], Kathryn Pennel [4], June Munro[1], Joseph A. Waldron[1], Holly Hall [1], Leah Officer-Jones [1], Sheila Bryson[1], Douglas Strathdee [1], Sergio Lilla [1], Sara Zanivan[1,4,5], Vivienne Morrison[1], Colin Nixon [1], Rachel A. Ridgway [1], Crispin Miller [1,4], John R. P. Knight[1,7], Andrew D. Campbell [1], Philip D. Dunne [1,3], John Le Quesne [1,4,6], Joanne Edwards [4], Peter J. Park [2], Martin Bushell[1,4] & Owen J. Sansom [1,4]✉

Nucleophosmin (NPM1), a nucleolar protein frequently mutated in hematopoietic malignancies, is overexpressed in several solid tumors with poorly understood functional roles. Here, we demonstrate that *Npm1* is upregulated after APC loss in WNT-responsive tissues and supports WNT-driven intestinal and liver tumorigenesis. Mechanistically, NPM1 loss induces ribosome pausing and accumulation at the 5′-end of coding sequences, triggering a protein synthesis stress response and p53 activation, which mediate this antitumorigenic effect. Collectively, our data identify NPM1 as a critical WNT effector that sustains WNT-driven hyperproliferation and tumorigenesis by attenuating the integrated stress response and p53 activation. Notably, *NPM1* expression correlates with elevated WNT signaling and proliferation in human colorectal cancer (CRC), while CRCs harboring *NPM1* deletions exhibit preferential *TP53* inactivation, underscoring the clinical relevance of our findings. Being dispensable for adult epithelial homeostasis, NPM1 represents a promising therapeutic target in p53-proficient WNT-driven tumors, including treatment-refractory *KRAS*-mutant CRC, and hepatic cancers.

The response to specific drivers of proliferation and oncogenic transformation varies greatly among tissues[1]. This tissue-specific 'permissiveness' is exemplified by selective mutations in pathway components driving cancers of distinct origins. For instance, the WNT pathway is crucial for tumor initiation in several tissues. However, mutations in *CTNNB1* are prevalent in hepatocellular carcinomas (HCCs), but less frequent in colorectal cancer (CRC)[2]. Conversely, inactivating mutations in the tumor suppressor adenomatous polyposis coli (*APC*) are rare in HCC, but among the most frequent in CRC, occurring in ~80% of cases. These mutations result in ligand-independent WNT-pathway activation and tumor initiation[2,3]. APC-deficient intestinal cells rely on mechanistic target of rapamycin complex 1 (mTORC1) to promote translation elongation and hyperproliferation, which can be effectively suppressed by rapamycin[4]. However, acquisition of additional mutations, such as oncogenic *KRAS*, found in ~40% of patients with CRC, renders rapamycin ineffective[5,6]. This highlights

[1]Cancer Research UK Scotland Institute, Glasgow, UK. [2]Department of Biomedical Informatics, Harvard Medical School, Boston, MA, USA. [3]The Patrick G Johnston Centre for Cancer Research, Queen's University Belfast, Belfast, UK. [4]School of Cancer Sciences, University of Glasgow, Glasgow, UK. [5]Department of Experimental Therapeutics, University of Texas MD Anderson Cancer Center, Houston, TX, USA. [6]Department of Histopathology, Queen Elizabeth University Hospital, Glasgow, UK. [7]Present address: Division of Cancer Sciences, University of Manchester, Manchester, UK. ✉e-mail: g.kanellos@crukscotlandinstitute.ac.uk; o.sansom@crukscotlandinstitute.ac.uk

the need for additional druggable target identification to tackle WNT-driven disease.

Nucleophosmin (NPM1) is a multifunctional nucleolar protein essential for embryonic development[7,8]. It has established roles in cell proliferation, survival, genomic integrity, ribosome biogenesis and posttranscriptional ribosomal RNA modifications among others[7-11]. It also functions as a molecular chaperone, preventing protein aggregation within the confined nuclear space and during stress[12,13]. *NPM1* is the most frequently mutated gene in acute myeloid leukemia (AML)[14]. These mutations typically produce a truncated, cytoplasmic form of NPM1 that facilitates increased expression of genes driving leukemogenesis[15-17]. In solid tumors, *NPM1* is rarely mutated but it is frequently overexpressed in multiple cancers, including CRC, hepatic and lung[18-22]. Despite extensive research in hematological malignancies, any roles of NPM1 in solid tumors, its impact on adult tissue homeostasis and the potential for repurposing emerging AML-targeted NPM1 therapies to benefit more patients remain unclear.

Pharmacological inhibition of WNT signaling in tumor cells dependent on WNT ligands for growth (for example, with *RNF43* loss-of-function mutations) induces MYC-dependent downregulation of NPM1 and other ribosome biogenesis and rRNA processing factors, identifying NPM1 as a key effector in the WNT–MYC axis[23]. *NPM1* has also been identified as both a MYC target gene and a regulator of MYC activity[24-26], indicating a potential role in tumorigenesis through MYC.

Furthermore, NPM1 has been shown to regulate tumor suppressor pathways. Under stress conditions, NPM1 may activate p53 by directly binding to p53 or its ubiquitin ligase MDM2, thereby stabilizing p53 expression[27,28]. Conversely, *Npm1* deletion can also result in increased p53 levels and p53-dependent growth arrest and apoptosis[7,8]. While part of these effects may be indirect, they highlight a context-dependent NPM1-p53 relationship.

In this study, we identify *Npm1* as a key gene upregulated in organs permissive to WNT-driven oncogenic growth and demonstrate its relevance to human disease. Using genetically engineered mouse models, we show that while NPM1 is dispensable for epithelial tissue homeostasis, it is essential for WNT-driven transformation by attenuating a protein synthesis stress response and posttranscriptional

p53 activation. Given that NPM1 is already a therapeutic target in hematopoietic cancers, our findings suggest that targeting NPM1 could also benefit patients with WNT-driven solid tumors, such as *KRAS*-mutant CRC and hepatic cancers, which remain challenging to treat.

## Results

### *Npm1* is upregulated in WNT-permissive tissues

Genetic alterations in the WNT signaling pathway are common in gastrointestinal tract tumors ('WNT-permissive tissues'), while less frequent in other tissues[2,29]. We sought to identify genes that are commonly upregulated across multiple WNT-permissive tissues after global hyperactivation of WNT. This was achieved by acutely deleting *Apc* (*Apc^loxP/loxP*) within multiple adult mouse tissues using a tamoxifen-inducible Cre recombinase driven by the ubiquitously expressing *Rosa26* promoter (*R26-Cre^ERT2*), followed by transcriptomic analysis (Fig. 1a). The small intestine (SI) was the only tissue exhibiting a pronounced phenotype, displaying crypt expansion (Extended Data Fig. 1a). Nuclear β-catenin in *Apc^loxP/loxP* animals confirmed successful WNT activation in tissues (Extended Data Fig. 1b). The SI, esophagus, colon and stomach had the strongest activation of a WNT transcriptional program, confirming their WNT-permissive status, while the bladder, kidneys, lungs and spleen had comparatively minimal activation, reflecting their nonpermissiveness (Fig. 1b). We identified only 107 genes upregulated in at least four of the WNT-permissive tissues (Fig. 1c). These included known WNT mediators and negative feedback regulators of WNT, for example, *RNF43*, *ZNRF3*, *AXIN2*, *TCF7* and *NOTUM*, confirming successful activation of the WNT transcriptional program in WNT-permissive tissues (Fig. 1c). We also confirmed enrichment in MYC transcriptional signatures, which is consistent with the roles of MYC downstream of APC (Extended Data Fig. 1c).

Given its association with MYC and altered expression in tumors, NPM1 was a target of interest (Fig. 1c). *Npm1* was strongly upregulated in WNT-permissive tissues, except the liver, where the APC-MYC module is uncoupled (see below) (Fig. 1d). *Npm1* was also moderately increased in the bladder and kidney, both of which were enriched in translation-related signatures, a process where NPM1 and MYC have important roles (Fig. 1d and Extended Data Fig. 1c).

**Fig. 1 | *Npm1* is upregulated in WNT-permissive tissues after APC loss and correlates with increased proliferation in patients with CRC. a**, Schematic representation of the experimental strategy used to identify targets commonly upregulated in tissues with WNT signaling activation after *Apc* deletion. Cre recombinase was expressed from the *Rosa26* locus (*R26-Cre^ERT2*); *Apc^loxP/loxP* alleles underwent Cre-mediated recombination upon tamoxifen administration. Multiple tissues were collected four days after induction and subjected to RNA-seq. **b**, Heatmap displaying GSEA in WNT-permissive and nonpermissive tissues for the indicated Reactome and published WNT activation signatures in the *Apc^loxP/loxP* tissues compared to WT (see Methods for more details on the signatures used). Enrichment level is based on each group average. **c**, Venn diagram depicting the number of unique and commonly upregulated genes with log_2(fold change) > 1 among WNT-permissive tissues (top), and list of upregulated genes in at least four of the permissive tissues (bottom). Known common regulators of WNT signaling are highlighted in the green boxes. **d**, *Npm1* relative expression in WNT-permissive and nonpermissive tissues, highlighted in green and blue, respectively (for esophagus, mid-colon, stomach corpus, liver, bladder, kidney, lung and spleen; *Apc^+/+* n = 4; *Apc^loxP/loxP* n = 5; for proximal SI n = 3 per group; for stomach antrum n = 4 per group). Statistical significance was assessed using multiple two-sided *t*-tests, with *P* values adjusted using the Holm–Šidák method. Adjusted *P* values are shown. **e**, TCGA cancer types sorted according to median *NPM1* expression in tumor and median *NPM1* expression in adjacent normal. *n* denotes the number of samples. The *P* values shown above the boxes assessing the tumor-normal difference were obtained using a two-sided *t*-test. **f**, Scatter plot showing the correlation between the mean tumor-normal difference of the GSVA score of the signature (WNT signature a) compared to the mean tumor-normal difference of *NPM1* expression across different tumor

types in the TCGA dataset. **g**, Scatter plot showing the correlation between *NPM1* expression and the GSVA score of the signature (WNT signature a) among COAD/READ tumors (*n* = 375) in the TCGA dataset. **f,g**, Data were statistically assessed using a two-sided Pearson correlation. The correlation coefficient (*r*) is displayed to indicate the degree of association. **h**, Representative examples of low-intensity and high-intensity NPM1 immunohistochemistry (IHC) staining from a human tissue microarray (TMA) of patients with CRC. **i**, Plot of CRC tumors ranked according to high or low NPM1 protein levels in the CRC TMA and their respective percentage in Ki-67⁺ cells. Each point corresponds to an individual patient with CRC; the mean ± s.e.m. is shown (*n* = 279 patients with low and *n* = 193 patients with high NPM1 expression within tumors). Data were statistically assessed using an unpaired two-tailed Mann–Whitney *U*-test. **j**, *NPM1* expression in the GSE39582 patient cohort of fresh-frozen CRC microarray data (tumor: *n* = 566; normal: *n* = 19). Data were statistically assessed using a two-sided Wilcoxon rank-sum test. **k**, Scatter plot showing the correlation between *NPM1* expression and the proliferation index in the GSE39582 CRC patient cohort. Data were statistically assessed using a two-sided Pearson correlation. A linear regression line is shown with the gray bands representing the 95% confidence intervals; the correlation coefficient (*r*) is displayed to indicate the degree of association. **l**, *NPM1* expression across PDS in the GSE39582 CRC cohort (PDS1 *n* = 186; PDS2 *n* = 140; PDS3 *n* = 122). Data were statistically assessed using a two-sided Wilcoxon rank-sum test. In all box plots, the boxes extend from the 25th to 75th percentiles, the whiskers extend to the minimum and maximum values, the line in every box is plotted at the median and outliers as dots outside whiskers. Statistically significant *P* values are shown in red. **a**, Illustrations created in BioRender.com (https://BioRender.com/s46am60 and https://BioRender.com/8kuycvt). **h**, Scale bar, 100 μm.

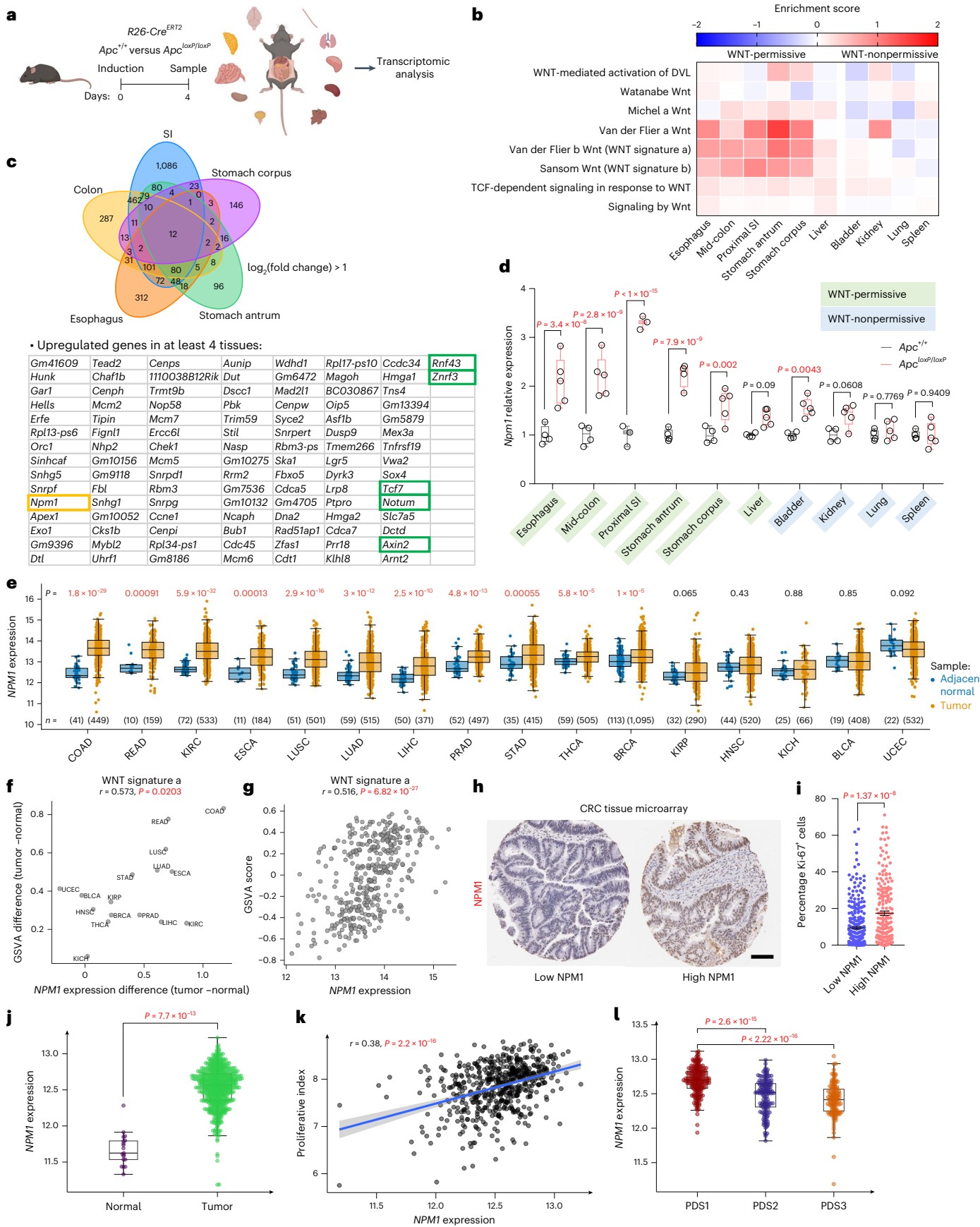

### *NPM1* is deregulated in CRC and correlates with WNT/MYC signaling

To assess if this link between NPM1 and WNT/MYC signaling was detectable in human cancer, we first examined human tumors from The Cancer Genome Atlas (TCGA). Indeed, *NPM1* is significantly overexpressed in many human cancers compared to adjacent normal tissues, but the strongest increase is within colon adenocarcinoma (COAD) and rectal adenocarcinoma (READ) (Fig. 1e). Gene set enrichment analysis (GSEA) of WNT signatures revealed a significant correlation between WNT activation and *NPM1* expression across tumor types, and within COAD/READ tumors, the cancer subtype exhibiting the strongest WNT activation (Fig. 1f,g and Extended Data Fig. 1d,e). *NPM1* expression was also highly correlated with *MYC*, supporting its status as a MYC target gene (Extended Data Fig. 1f).

We next compared proliferation across a cohort of patients with CRC and found significantly higher Ki-67 positivity in samples with high NPM1 expression (Fig. 1h,i). Microarray data from an independent CRC patient cohort confirmed significantly higher *NPM1* expression in tumors compared to adjacent normal tissue and a significant correlation with proliferative index (Fig. 1j,k). Consistent with this, *NPM1* expression was significantly higher in the stem-cell-rich, highly proliferative, pathway-derived subtype 1 (PDS1) of the PDS classification[30] (Fig. 1l). Taken together, these data support a role for NPM1 in WNT-driven disease with human relevance. Therefore, we next investigated its functions during homeostasis and WNT-driven tumorigenesis.

### NPM1 is critical for WNT-driven hyperproliferation, not homeostasis

As *Npm1* deletion is embryonically lethal, we investigated whether its loss is tolerated in adult mice. To this end, we generated mice with conditional *Npm1* alleles (*Npm1*^*loxP/loxP*) (Extended Data Fig. 1g) and crossed them with *R26-Cre*^*ERT2* mice. Long-term NPM1 loss across multiple tissues caused no signs of ill health or gross tissue abnormalities (Extended Data Fig. 2a,b). Consistent with what has been described

previously, the only anomaly detected was differences in erythropoiesis, with *Npm1*^*loxP/loxP* animals having fewer and larger circulating red blood cells, and more immature reticulocytes, compared to wild-type (WT)[7,31,32] (Extended Data Fig. 2c–n). This confirms NPM1's roles in hematopoietic differentiation but, importantly, demonstrates that its loss does not cause immediate tissue toxicity. Moreover, proliferative organs like the intestine and skin retained *Npm1* deletion without gross phenotypes, suggesting that although associated with cancer cell proliferation, NPM1 is dispensable for normal homeostatic proliferation, implying probable tolerance for drug targeting.

To investigate whether *Npm1* upregulation after APC loss has functional significance, we crossed *Npm1*^*loxP/loxP* with *R26-Cre*^*ERT2*Apc^*loxP/loxP* mice to achieve concurrent deletion of both genes in vivo. *Apc* deletion induced robust proliferation in WNT-permissive tissues (SI, stomach and liver), while proliferation remained unchanged in WNT-nonpermissive tissues (lungs, kidneys and pancreas) (Extended Data Fig. 3a–i). Notably, concurrent *Npm1* deletion significantly attenuated this response, highlighting NPM1 as a key mediator of WNT-driven hyperproliferation.

### NPM1 is dispensable for murine intestinal epithelium homeostasis

Given this functional association of NPM1 downstream of APC loss and its marked deregulation in CRC, we investigated its role further using murine CRC models. We generated *villin-Cre*^*ERT2*Npm1^*loxP/loxP* mice and confirmed efficient, tissue-specific *Npm1* deletion within the intestinal epithelium (Extended Data Fig. 4a). Long-term NPM1 loss was tolerated without disrupting tissue architecture, intestinal stem or differentiated cell populations (for example, goblet and Paneth cells) or proliferation (Extended Data Fig. 4b,c). Despite being a nucleolar factor with roles in ribosome biogenesis[33–35], NPM1 depletion did not alter 47S pre-rRNA production (Extended Data Fig. 4d,e). Consistent with this, global protein synthesis remained unaffected in ex vivo-induced *Npm1*^*loxP/loxP* intestinal organoids, in line with previous findings[9]

**Fig. 2 | NPM1 loss suppresses WNT-driven hyperproliferation and tumor formation in vivo. a**, *Npm1* relative expression in SI and colon tissues from WT mice and a range of genetically engineered murine models of CRC (GSE309379; *n* = 5: WT (SI and colon), *VilCre*^*ERT2*Braf^*V600E/+* (colon), *VilCre*^*ERT2*Braf^*V600E/+*Trp53^*loxP/loxP*ALK5^*loxP/loxP*R26^*LSL-N1icd/+* (SI); *n* = 4: *VilCre*^*ERT2* (SI and colon), *VilCre*^*ERT2*Braf^*V600E/+*Trp53^*loxP/loxP*ALK5^*loxP/loxP* (SI), *VilCre*^*ERT2*Apc^*loxP/loxP*KRas^*G12D/+* (colon), *VilCre*^*ERT2*Apc^*loxP/loxP*KRas^*G12D/+*Trp53^*loxP/loxP* (SI), *VilCre*^*ERT2*Apc^*loxP/loxP*Trp53^*loxP/loxP* (SI), *AhCre*^*ERT2*Apc^*loxP/loxP*Myc^*loxP/loxP* (SI); *n* = 3: *VilCre*^*ERT2*KRas^*G12D/+*Trp53^*loxP/loxP* (SI), *VilCre*^*ERT2*Apc^*loxP/loxP*KRas^*G12D/+*Trp53^*loxP/loxP* (colon); *n* = 7: *VilCre*^*ERT2*Braf^*V600E/+*Trp53^*loxP/loxP* (SI and colon), *VilCre*^*ERT2*KRas^*G12D/+*Trp53^*loxP/loxP*R26^*LSL-N1icd/+* (SI); *n* = 10: *VilCre*^*ERT2*Apc^*loxP/loxP* (SI); *n* = 12: *VilCre*^*ERT2*Apc^*loxP/loxP* (colon); *n* = 8: *VilCre*^*ERT2*Apc^*loxP/loxP*KRas^*G12D/+* (SI); *n* = 6: *AhCre*^*ERT2*Apc^*loxP/loxP* (SI)). Data were statistically assessed by one-way analysis of variance (ANOVA) followed by Dunnett's multiple comparisons test. **b**, Representative micrographs from *Apc*^*loxP/loxP*Npm1^*+/+* (*n* = 5) and *Apc*^*loxP/loxP*Npm1^*loxP/loxP* (*n* = 5) SI sections stained with hematoxylin and eosin (H&E) and anti-bromodeoxyuridine (BrdU) from mice collected 4 days after induction. The red bars indicate the expanded crypt depth after APC loss. **c**, Quantification of BrdU⁺ cells in SI half-crypts of animals from the groups shown in **a** (*n* = 5 per group). Data were statistically assessed using an unpaired, two-tailed Mann–Whitney *U*-test. **d**, Survival curves of *Apc*^*loxP/+* (*n* = 14) and *Apc*^*loxP/+*Npm1^*loxP/loxP* (*n* = 16) mice sampled at the clinical endpoint. Median survival in days is indicated in parentheses. Censored mice are denoted as tick marks at the indicated times after induction. The *P* value was obtained using a log-rank (Mantel–Cox) test. **e**, Tumor numbers from *Apc*^*loxP/+* (*n* = 8) and *Apc*^*loxP/+*Npm1^*loxP/loxP* (*n* = 9) mice sampled at the clinical endpoint. Data were statistically assessed using an unpaired, two-tailed *t*-test. **f**, Representative staining for NPM1 on SI tissue sections from *Apc*^*loxP/+*Npm1^*loxP/loxP* animals at the clinical endpoint (*n* = 5). **g**, Quantification of the percentage of tumors being positive, negative or mosaic for NPM1 expression in each *Apc*^*loxP/+*Npm1^*loxP/loxP* animal at the clinical endpoint (*n* = 5). **h**, Survival curves of *Lgr5-Cre*^*ERT2*Apc^*loxP/loxP* (*n* = 13) and *Lgr5-Cre*^*ERT2*Apc^*loxP/loxP*Npm1^*loxP/loxP* (*n* = 11) mice sampled at the clinical endpoint. Median survival in days is indicated in parentheses. Censored mice

are denoted as tick marks at the indicated times after induction. The *P* value was obtained using a log-rank (Mantel–Cox) test. **i**, Tumor numbers from *Lgr5-Cre*^*ERT2*Apc^*loxP/loxP* (*n* = 11) and *Lgr5-Cre*^*ERT2*Apc^*loxP/loxP*Npm1^*loxP/loxP* (*n* = 9) mice sampled at the clinical endpoint. Data were statistically assessed using an unpaired, two-tailed Mann–Whitney *U*-test. **j**, Representative staining for NPM1 on SI tissue sections from *Lgr5-Cre*^*ERT2*Apc^*loxP/loxP* (*n* = 3) and *Lgr5-Cre*^*ERT2*Apc^*loxP/loxP*Npm1^*loxP/loxP* (*n* = 7) animals at the clinical endpoint. **k**, Quantification of the percentage of tumors being positive, negative or mosaic for NPM1 expression in each *Lgr5-Cre*^*ERT2*Apc^*loxP/loxP*Npm1^*loxP/loxP* animal at the clinical endpoint (*n* = 7). **l**, Representative micrographs from *Apc*^*loxP/loxP*KRas^*G12D/+*Npm1^*+/+* (*n* = 4) and *Apc*^*loxP/loxP*KRas^*G12D/+*Npm1^*loxP/loxP* (*n* = 4) SI sections stained with H&E and anti-BrdU from mice collected 3 days after tamoxifen induction. The red bars indicate the expanded crypt depth after APC loss and KRAS^G12D activation. **m**, Quantification of BrdU⁺ cells in the SI of animals from the groups shown in **l** (*n* = 4 per group). Data were statistically assessed using an unpaired, two-tailed *t*-test. **n**, Survival curves of *Apc*^*loxP/+*KRas^*G12D/+* (*n* = 17) and *Apc*^*loxP/+*KRas^*G12D/+*Npm1^*loxP/loxP* (*n* = 20) mice sampled at the clinical endpoint. Median survival in days is indicated in parentheses. The *P* value was obtained using a log-rank (Mantel–Cox) test. **o**, Tumor numbers from *Apc*^*loxP/+*KRas^*G12D/+* (*n* = 17) and *Apc*^*loxP/+*KRas^*G12D/+*Npm1^*loxP/loxP* (*n* = 20) mice sampled at the clinical endpoint. Data were statistically assessed using an unpaired, two-tailed Mann–Whitney *U*-test. **p**, Representative staining for NPM1 on colon tissue sections from *Apc*^*loxP/+*KRas^*G12D/+* and *Apc*^*loxP/+*KRas^*G12D/+*Npm1^*loxP/loxP* animals at the clinical endpoint. **q**, Quantification of the percentage of tumors being positive, negative or mosaic for NPM1 expression in each *Apc*^*loxP/+*KRas^*G12D/+*Npm1^*loxP/loxP* animal at the clinical endpoint (*n* = 9) compared to that of *Apc*^*loxP/+*Npm1^*loxP/loxP* animals (*n* = 5) presented in Fig. 2g. Data were statistically assessed using a two-way ANOVA with Šidák's correction for multiple comparisons. The bar charts present data as the mean ± s.e.m.; the boxes in the box plots extend from the 25th to 75th percentile, the whiskers extend to the minimum and maximum values, and the line in every box is plotted at the median. Statistically significant *P* values are shown in red. **b,l**, Scale bar, 50 μm. **f**, Scale bar, 1 mm. **j**, Scale bar, 300 μm. **p**, Scale bar, 500 μm.

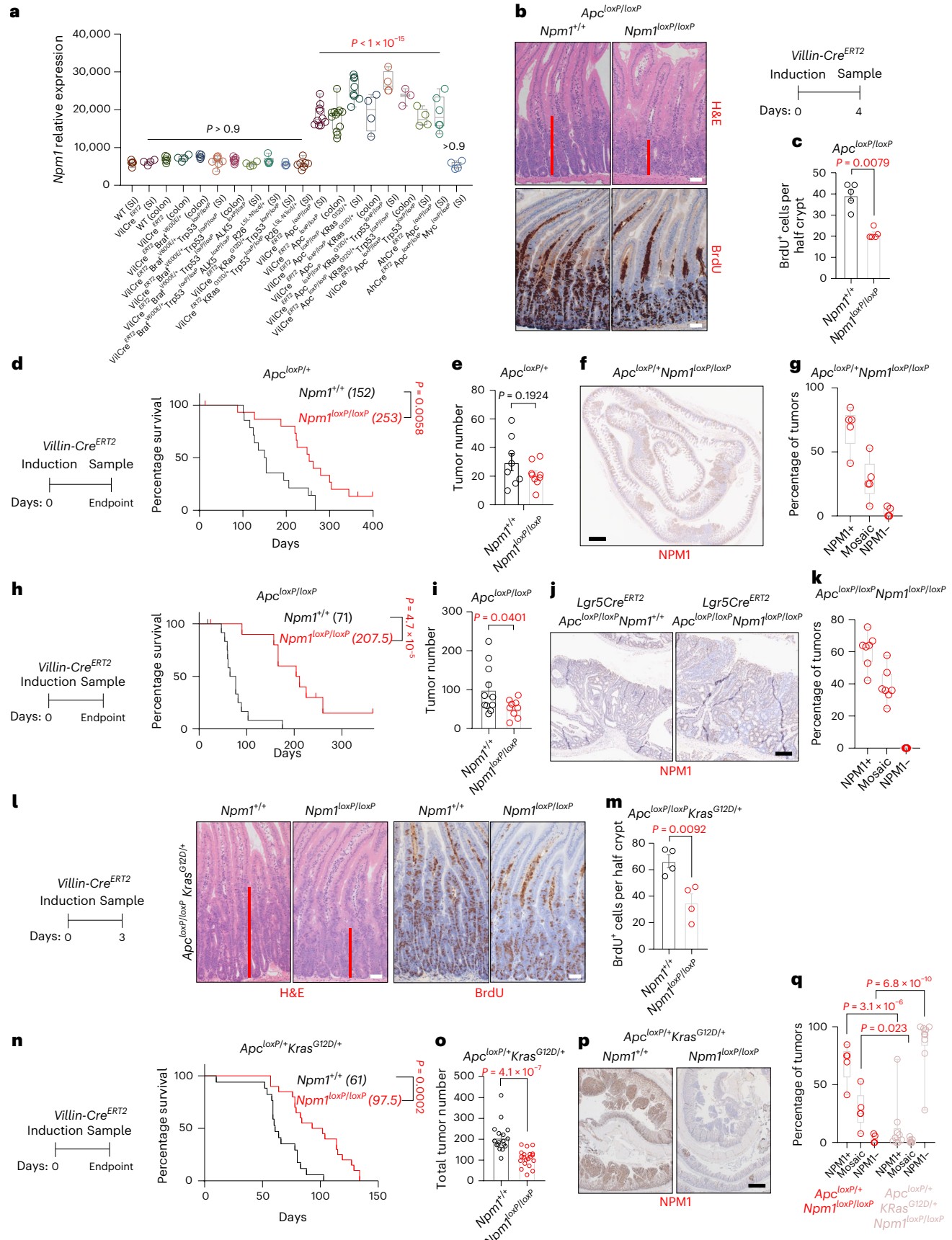

(Extended Data Fig. 4f,g). Nucleolar number was also unchanged after *Npm1* deletion, although fibrillarin staining revealed a minor variation in nucleolar circularity (Extended Data Fig. 4h–q).

## NPM1 loss limits proliferation and extends survival in WNT-high CRC

We next investigated whether *Npm1* upregulation depends specifically on high WNT signaling after APC loss, rather than other oncogenic mutations unrelated to the WNT pathway. Notably, *Npm1* overexpression was observed exclusively in models with *Apc* deletion but not in those harboring other CRC-relevant mutations such as *Braf*, *Kras* or *Trp53* (Fig. 2a and Extended Data Fig. 5a–c). *Myc* co-deletion prevented *Npm1* overexpression, suggesting MYC-dependent regulation (Fig. 2a). NPM1 depletion did not abrogate activation of the BRAF-associated fetal-like transcriptional program[36] (Extended Data Fig. 5d,e), further supporting its association with high WNT. Consistent with this, *APC* mutant human CRCs exhibit enhanced WNT signaling and significantly higher *NPM1* expression compared to *APC* WT counterparts (Extended Data Fig. 5f–h).

*Apc* deletion in *villin-Cre^ERT2*-expressing cells triggered intestinal hyperproliferation, which was markedly suppressed by concurrent *Npm1* deletion (Fig. 2b,c). Interestingly, the *Olfm4^+* stem cell population remained unaffected (Extended Data Fig. 5i,j). Stochastic loss of the second *Apc* allele in *villin-Cre^ERT2 Apc^loxP/+* mice promotes tumor formation. *Npm1* deletion significantly extended survival, although most tumors escaped recombination and retained NPM1 expression (Fig. 2d–g and Extended Data Fig. 5k). Notably, the surrounding normal epithelium maintained *Npm1* deletion, suggesting a tumor-specific requirement for NPM1 during initiation. Given that the stem cell population appeared unaffected, we investigated whether tumorigenesis driven by *Apc* deletion in *Lgr5*-expressing cells (*Lgr5-Cre^ERT2 Apc^loxP/loxP*) was suppressed by NPM1 loss. Remarkably, *Npm1* deletion suppressed this rapid adenoma model, extending survival threefold, while tumors again retained NPM1 expression (Fig. 2h–k and Extended Data Fig. 5l).

Additionally, NPM1 loss significantly reduced hyperproliferation without affecting stem cells after oncogenic *Kras* activation (*Kras^G12D/+*) in *villin-Cre^ERT2 Apc^loxP/loxP* mice (Fig. 2l,m and Extended Data Fig. 5m,n). *KRAS* mutations, found in ~40% of cases with CRC, often follow *APC* loss, confer therapy resistance and accelerate disease progression[6]. NPM1 loss also significantly extended survival in *villin-Cre^ERT2 Apc^loxP/+ Kras^G12D/+* mice (Fig. 2n). Unlike with APC loss alone, oncogenic *Kras* enabled the establishment of NPM1⁻ tumors, although their number was significantly reduced (Fig. 2o–q and Extended Data Fig. 5o). Taken together, our data suggest that NPM1 is required for WNT-driven intestinal cell proliferation and tumor initiation; this dependency persists after oncogenic KRAS activation, which drives resistance to epidermal growth factor receptor and mTOR inhibition.

## p53 mediates the antitumorigenic effects of *Npm1* deletion

Given the marked phenotypic suppression after *Apc* deletion, we examined transcriptional changes to uncover how NPM1 loss mediates this effect. Surprisingly, RNA sequencing (RNA-seq) revealed minimal transcriptional differences (Extended Data Fig. 6a–c). Previous studies linked *Npm1* embryonic knockout (KO) to increased p53 and p21 levels[7,8], prompting us to investigate a potential p53 response in NPM1-depleted intestines. Both p53 and p21 were upregulated in intestinal crypts, irrespective of oncogenic KRAS^G12D activation or APC loss (Fig. 3a–i). Increased p53 levels were also evident in the crypts and tumors of *Apc^loxP/+ Kras^G12D/+* mice collected at the clinical endpoint (Fig. 3j,k and Extended Data Fig. 6d). Consistent with our transcriptomic data, the mRNA levels of *Trp53*, cyclin-dependent kinase inhibitor 1a (*Cdkn1a*, encoding p21) and several p53 target genes (for example, *Bax*, BCL2 binding component 3 (*Bbc3*), phorbol-12-myristate-13-acetate-induced protein 1 (*Pmaip1*)) remained unchanged (Extended Data Fig. 6e,f), suggesting posttranscriptional regulation of p53 and p21 after NPM1 loss.

Concurrent conditional deletion of *Trp53* (*Trp53^loxP/loxP*) alongside *Npm1* and *Apc* restored proliferation in the *Apc^loxP/loxP Trp53^loxP/loxP*

---

**Fig. 3 | p53 upregulation suppresses WNT-driven hyperproliferation and tumorigenesis in the Npm1-deficient intestine. a**, Representative p53 (yellow) staining from SI tissue sections of *Npm1^+/+* and *Npm1^loxP/loxP* (*n* = 4 per group) animals collected 4 days after induction. Nuclei (blue) are visualized with 4′,6-diamidino-2-phenylindole (DAPI). **b**, Quantification of p53⁺ cells in the crypts from the groups in **a**. Data were statistically assessed using an unpaired, two-tailed Mann–Whitney *U*-test. **c**, Representative p21 staining from SI tissue sections of *Npm1^+/+* (*n* = 4) and *Npm1^loxP/loxP* (*n* = 3) animals collected 120 days after induction (top row) and after additional activation of KRAS^G12D (*Kras^G12D/+ n* = 3; *Kras^G12D/+ Npm1^loxP/loxP n* = 3) collected 30 days after induction (bottom row). **d,e**, Quantification of p21⁺ cells in SI half-crypts of *Npm1^+/+* and *Npm1^loxP/loxP* animals without (**d**) and with (**e**) additional KRAS^G12D activation from the groups presented in **c**. Data were statistically assessed using an unpaired, two-tailed *t*-test. **f**, Representative staining for NPM1 and p21 on SI tissue sections from *Apc^loxP/loxP* and *Apc^loxP/loxP Npm1^loxP/loxP* animals collected 4 days after induction (*n* = 6 per group). The red dotted line indicates the outer edges of the intestinal crypts. **g**, Quantification of p21⁺ cells in the SI half-crypts of animals from the groups in **f**. Data were statistically assessed using an unpaired, two-tailed *t*-test. **h**, Representative staining for p21 (top) and p53 (yellow, bottom) on SI tissue sections from *Apc^loxP/loxP Kras^G12D/+* and *Apc^loxP/loxP Kras^G12D/+ Npm1^loxP/loxP* animals collected 3 days after induction. The red dotted line indicates the outer edges of the intestinal crypts. Nuclei (blue) were visualized with DAPI. **i**, Quantification of p21⁺ cells in SI half-crypts and p53⁺ cells in the crypts of animals from the groups in **h** (*Apc^loxP/loxP Kras^G12D/+* (*n* = 3 for p21, *n* = 4 for p53); *Apc^loxP/loxP Kras^G12D/+ Npm1^loxP/loxP* (*n* = 4)). Data were statistically assessed using an unpaired, two-tailed *t*-test. **j,k**, Quantification of p53⁺ cells in SI crypts (**j**) and within tumors (**k**) of *Apc^loxP/+ Kras^G12D/+* and *Apc^loxP/+ Kras^G12D/+ Npm1^loxP/loxP* animals at the clinical endpoint. Data were statistically assessed using an unpaired, two-tailed *t*-test. Related to Extended Data Fig. 6d (*n* = 4 per group). **l**, Representative images of SI tissue sections stained with H&E, anti-BrdU and anti-NPM1 from *Apc^loxP/loxP Trp53^loxP/loxP* (*n* = 4) and *Apc^loxP/loxP Trp53^loxP/loxP Npm1^loxP/loxP* (*n* = 5) animals sampled 4 days after induction. The red bars indicate crypt depth. **m**, Quantification of BrdU⁺ cells in the SI half-crypts of animals from the groups shown in **l** and statistically compared to that of animals without *Trp53* deletion presented in Fig. 2c. Data were statistically assessed using a one-way ANOVA followed by Tukey's multiple comparisons test. **n**, Survival curves of *Apc^loxP/+* (*n* = 14) and *Apc^loxP/+ Npm1^loxP/loxP* (*n* = 16) mice presented as dotted lines (also shown in Fig. 2d), compared to that of *Apc^loxP/+ Trp53^loxP/loxP* (*n* = 14) and *Apc^loxP/+ Trp53^loxP/loxP Npm1^loxP/loxP* (*n* = 15) mice sampled at the clinical endpoint. Median survival in days is indicated in parentheses. *P* values were obtained using a log-rank (Mantel–Cox) test. **o**, Tumor number of *Apc^loxP/+* (*n* = 8) and *Apc^loxP/+ Npm1^loxP/loxP* (*n* = 9) (also shown in Fig. 2e), plotted with that of *Apc^loxP/+ Trp53^loxP/loxP* (*n* = 8) and *Apc^loxP/+ Trp53^loxP/loxP Npm1^loxP/loxP* (*n* = 14) mice sampled at the clinical endpoint. Data were statistically assessed using a one-way ANOVA followed by Tukey's multiple comparisons test. **p**, Representative staining for NPM1 on SI tissue sections from *Apc^loxP/+ Npm1^loxP/loxP* (*n* = 5) (also shown in Fig. 2f) and *Apc^loxP/+ Trp53^loxP/loxP Npm1^loxP/loxP* (*n* = 5) mice at the clinical endpoint. **q**, Quantification of the percentage of tumors being positive, negative or mosaic for NPM1 expression in *Apc^loxP/+ Trp53^loxP/loxP Npm1^loxP/loxP* mice at the clinical endpoint (*n* = 5) compared to that of *Apc^loxP/+ Npm1^loxP/loxP* animals (*n* = 5) presented in Fig. 2g. Data were statistically assessed using a two-way ANOVA followed by Šidák's multiple comparisons test. **r**, *NPM1* expression in WT and mutant (Mut) *TP53* COAD/READ tumors in the TCGA dataset (*n* = 375). Data were statistically assessed using a two-sided *t*-test. **s,t**, Table (**s**) and graphical representation (**t**) of COAD/READ tumors in the TCGA dataset (*n* = 375) with shallow *NPM1* deletions and *TP53* mutations. Data were analyzed using a two-sided Fisher's exact test. All bar charts present data as the mean ± s.e.m.; the boxes in the box plots extend from the 25th to 75th percentile, the whiskers extend to the minimum and maximum values, the line in every box is plotted at the median and outliers as dots outside the whiskers. Statistically significant *P* values are shown in red. **a**, Scale bar, 20 μm. **c,l**, Scale bar, 50 μm. **f**, Scale bar, 100 μm. **h**, Scale bar, 100 μm (top), 20 μm (bottom). **p**, Scale bar, 1 mm.

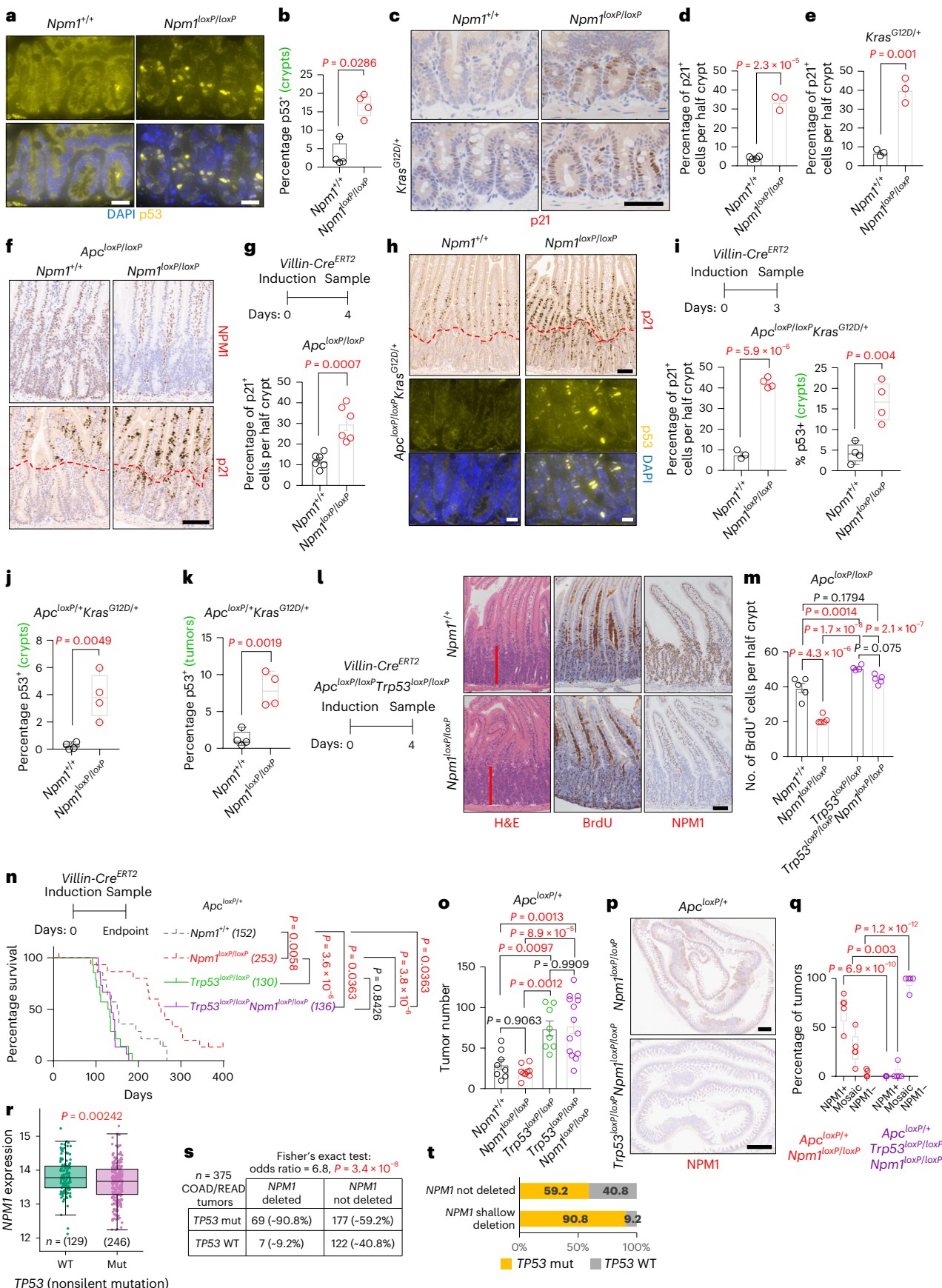

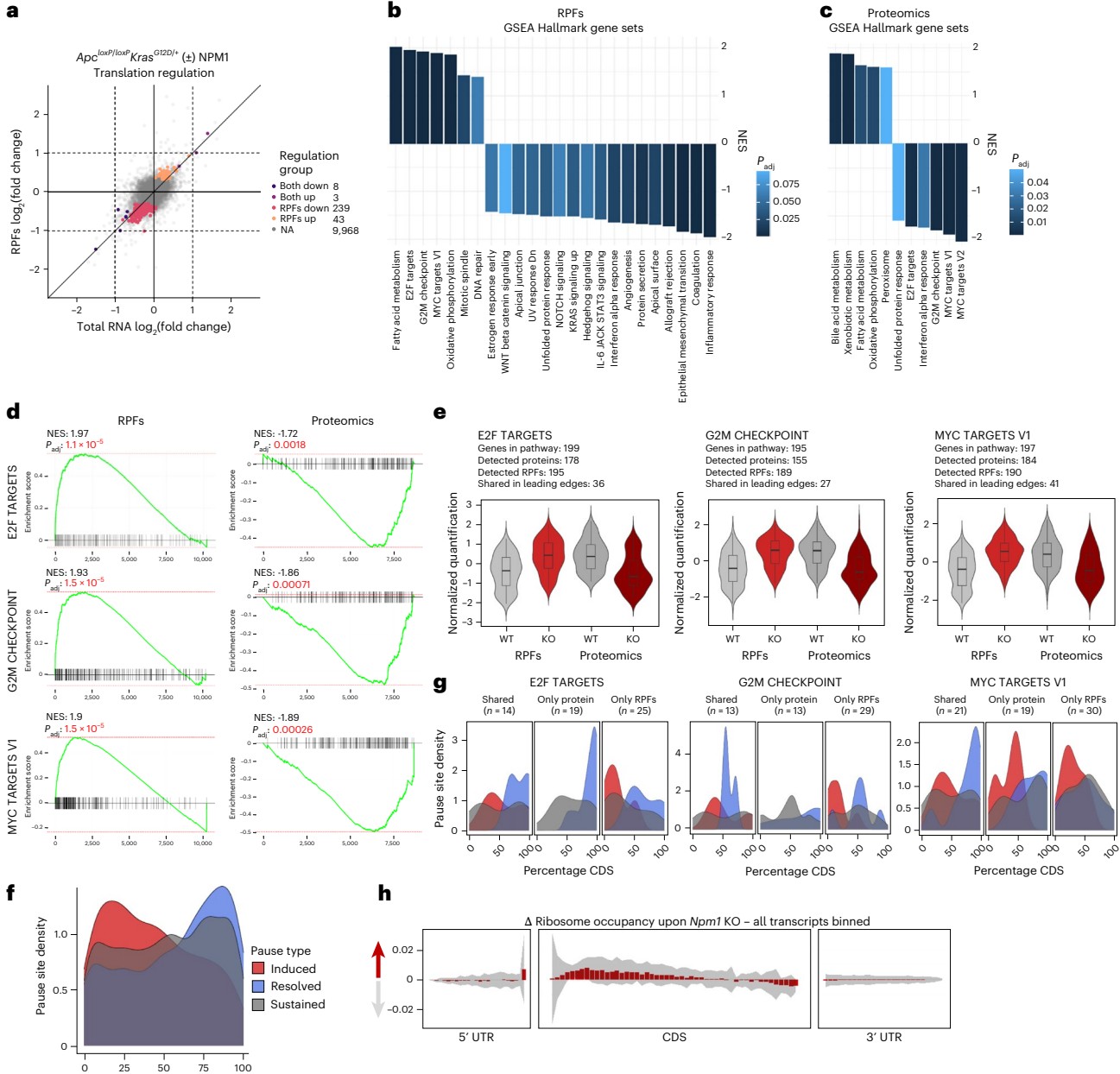

**Fig. 4 | NPM1 depletion triggers ribosome pausing.** Data comparing *Apc^{loxP/loxP}Kras^{G12D/+}Npm1^{loxP/loxP}* and *Apc^{loxP/loxP}Kras^{G12D/+}* intestinal epithelial cells, sampled 3 days after induction (*n* = 4 per group) **a**, Translation efficiency scatter with differentially expressed transcripts at the total cytoplasmic RNA level (*x* axis) compared to changes at the RPF level (*y* axis). Transcripts are color-coded depending on their change being significant at the total and RPF level (both up and both down), or exclusively at the RPF level (RPF up and RPF down) (*P*_{adj} < 0.1). No transcript was detected to be changed exclusively at the total cytoplasmic RNA level. **b**, GSEA for Hallmark gene sets on transcripts ranked according to changes at the RPF level. **c**, GSEA for Hallmark gene sets on genes ranked according to changes at the proteome level. **d**, Enrichment profiles from the GSEA for the Hallmark sets E2F targets, G2M checkpoint and MYC targets V1. Left, Enrichment profiles of RPFs. Right, Enrichment profiles of proteomics data. Enrichment scores were calculated using the weighted Kolmogorov–Smirnov test implemented in the fGSEA package. Significance and normalization were assessed using empirical permutation testing; *P* values were adjusted using the Benjamini–Hochberg method. Normalized enrichment scores (NES) and *P*_{adj} values are shown in the plots. **e**, Quantification of genes shared by the leading edges of both RPFs and proteomics (within the same pathway) are displayed as violins (*Npm1^{+/+}* (WT) condition in gray, *Npm1^{loxP/loxP}* (KO) condition in red). Each plot represents scaled abundances of RPF-normalized reads and protein

intensities; the number of genes in the shared leading edge is displayed above each panel. The box plots within the violins extend from the 25th to 75th percentile, with the lines plotted at the median. The whiskers extend to the minimum and maximum values, no further than 1.5 times the interquartile range from the hinges. Outliers beyond the end of the whiskers are not plotted. **f**, Ribosome pause site distribution across transcript CDS, with induced sites in red, resolved sites in blue and maintained sites in gray. **g**, Ribosome pause site distribution across genes belonging to the leading edges of the pathways indicated. For each pathway, the pause distribution is shown for those genes belonging to the leading edge of both RPFs and proteins, or to the leading edge exclusively for proteins (only protein) or RPFs (only RPFs). The number of genes in each of these categories is indicated above the plots. Color coding of the pause sites as in **f**. Note that the number of genes in the shared category is different than in **e** as we are displaying only transcripts for which ribosomes pause sites were confidently identified. **h**, Metagene plot of all transcripts detected in the Riboseq experiment with more than 50 average reads aligned to their CDS. The plot represents the variation in ribosome occupancy across the 5′ untranslated region (UTR) (first panel), their CDS (second panel) or their 3′ UTR (third panel). The gray shaded area represents the standard deviation of the delta ribosome occupancy across the four biological replicates assayed per group.

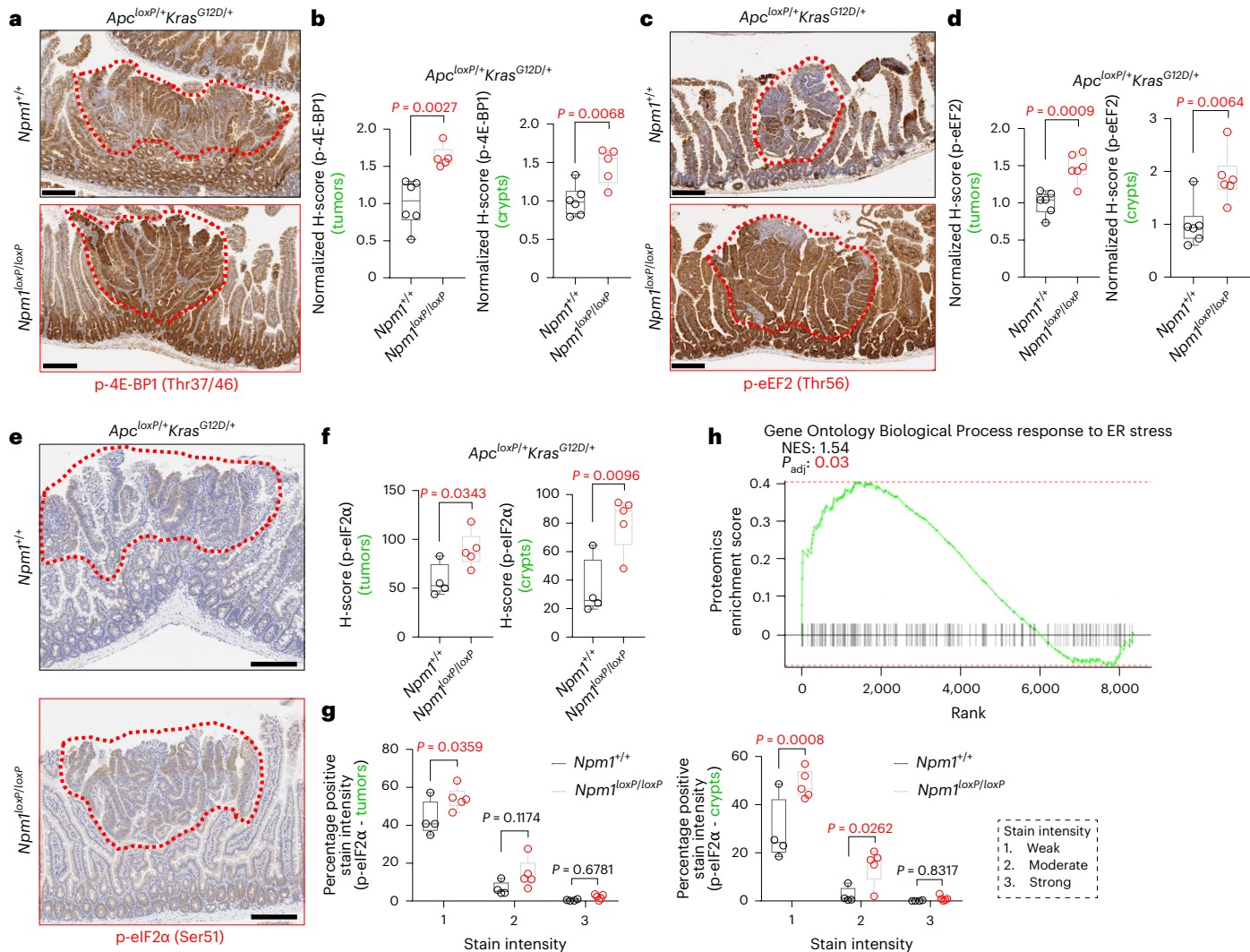

**Fig. 5 | NPM1 loss induces protein synthesis stress. a**, Representative staining for p-4E-BP1 (Thr37/46) on SI tissue sections from $Apc^{loxP/+}Kras^{G12D/+}$ ($n = 6$) and $Apc^{loxP/+}Kras^{G12D/+}Npm1^{loxP/loxP}$ ($n = 5$) animals collected at the endpoint. The red dotted lines indicate intestinal adenomas. **b**, HALO H-score quantification normalized to the average control value of p-4E-BP1 expression in tumors (left) and crypts (right) of the groups shown in **a**. **c**, Representative staining for p-eEF2 (Thr56) on SI tissue sections from $Apc^{loxP/+}Kras^{G12D/+}$ and $Apc^{loxP/+}Kras^{G12D/+}Npm1^{loxP/loxP}$ animals collected at the endpoint ($n = 6$ per group). The red dotted lines indicate intestinal adenomas. **d**, HALO H-score quantification normalized to the average control value of p-eEF2 expression in tumors (left) and crypts (right) of the groups shown in **c**. **e**, Representative staining for p-eIF2α (Ser51) on SI tissue sections from $Apc^{loxP/+}Kras^{G12D/+}$ ($n = 4$) and $Apc^{loxP/+}Kras^{G12D/+}Npm1^{loxP/loxP}$ ($n = 5$) animals collected at the endpoint. The red dotted lines indicate intestinal

adenomas. **f**, HALO H-score quantification of p-eIF2α expression in the tumors (left) and crypts (right) of the groups shown in **e**. **g**, HALO quantification of p-eIF2α stain intensity in the tumors (left) and crypts (right) of the groups shown in **e**. **h**, Enrichment profile for the pathway response to ER stress (from the Gene Ontology Biological Process subset) from the proteomics data. The NES and $P_{adj}$ value were calculated as in Fig. 4d and are shown in the plot. **b,d,f**, All data were statistically assessed using an unpaired, two-tailed $t$-test. **g**, Statistical significance was assessed using multiple two-sided $t$-tests, with $P$ values adjusted using the Holm-Šidák method. $P_{adj}$ values are shown. The boxes in box plots extend from the 25th to 75th percentile, the whiskers extend to the minimum and maximum values, and the line in every box is plotted at the median. Statistically significant $P$ values are shown in red. **a,c**, Scale bar, 300 μm. **e**, Scale bar, 200 μm.

$Npm1^{loxP/loxP}$ intestinal epithelium, confirming a functional relationship between p53 and NPM1 (Fig. 3l,m). Furthermore, p53 loss negated the survival benefit of NPM1 depletion in $Apc$-driven CRC models, while most tumors retained NPM1 loss (Fig. 3n–q and Extended Data Fig. 6g–l). To corroborate the role of the p53 pathway, we deleted $Cdkn1a$ in the $Lgr5$-$Cre^{ERT2}Apc^{loxP/loxP}$ tumor model. Like p53 loss, co-deletion of $Cdkn1a$ and $Npm1$ significantly reduced survival compared to p21-proficient $Npm1^{loxP/loxP}$ mice, and enabled the establishment of significantly more NPM1⁻ tumors (Extended Data Fig. 7a–e).

To probe a CRC model without the $Apc$ mutation, we investigated NPM1 loss in an aggressive model driven by oncogenic $Kras$, active NOTCH signaling and p53 loss[37]. $Npm1$ deletion had no impact (Extended Data Fig. 7f,g). In a less aggressive model driven by KRAS and

p53 loss, $Npm1$ deletion slightly but significantly accelerated tumorigenesis (Extended Data Fig. 7h,i). These data confirm NPM1 dependency in WNT-driven models and the requirement for p53.

Given this relationship with p53, we analyzed human tumors and found that $TP53$ WT samples from the TCGA COAD/READ dataset exhibited significantly higher $NPM1$ expression than the $TP53$-mutated samples (Fig. 3r). Among the $n = 375$ COAD/READ tumors, 76 harbored shallow $NPM1$ deletions (none harbored $NPM1$ deep deletions). Of these, 69 (~91%) also carried $TP53$ mutations, which is significantly higher than the 177 of 299 $TP53$-mutated tumors without $NPM1$ deletions (Fig. 3s,t). This suggests preferential $TP53$ inactivation in CRCs with $NPM1$ deletions, mirroring murine data where p53 loss is essential for establishing $Npm1$-null tumors. Together, our results

demonstrate that functional p53 signaling is essential for mediating the tumor-suppressive effects of NPM1 loss. However, it is important to note that the modest p53 induction after *Npm1* deletion in the WT intestine did not perturb normal homeostatic proliferation. This suggests that NPM1 loss triggers a specific stress after APC loss, rather than a broad effect on proliferation per se.

## NPM1 depletion induces ribosome pauses during translation

Given the minimal transcriptional changes after NPM1 loss despite the increase in p53/p21 protein levels, we investigated potential posttranscriptional regulation by performing ribosome profiling in intestinal epithelia from *Apc*<sup>loxP/loxP</sup>*Kras*<sup>G12D/+</sup> mice. Library quality control confirmed good ribosome protected fragment (RPF) quantification, with high enrichment in protein-coding sequences; principal component analysis showed separation between sample groups (Extended Data Fig. 8a–g). NPM1 loss induced significant changes in RPFs, but, strikingly, had minimal impact on total RNA levels, which is consistent with the bulk RNA-seq data (Fig. 4a and Extended Data Fig. 8h–k). To distinguish transcriptional from translational changes, we compared differentially expressed transcripts at the total RNA and RPF levels. We identified 239 transcripts with decreased and 43 transcripts with increased RPF abundance, without corresponding changes in total cytoplasmic RNA (Fig. 4a). Interestingly, only few transcripts changed at both levels (eight decreased, three increased) with none changing exclusively at the mRNA level.

To identify pathways significantly affected by NPM1 loss, we performed GSEA based on RPF-level changes. Among the top positively enriched pathways (Fig. 4b), we confirmed DNA repair activation by increased γH2AX positivity in NPM1-depleted cells and tumors of *Apc*<sup>loxP/+</sup>*Kras*<sup>G12D/+</sup> mice, without a corresponding increase in apoptosis, as assessed by cleaved-PARP levels. Interestingly, *NPM1* expression was significantly correlated with DNA repair and replication stress signatures in human CRCs (Extended Data Fig. 8l–u). However, the positive enrichment of pathways such as E2F targets, G2M checkpoint and MYC targets (v1) was in stark contrast to the reduced proliferation phenotype observed upon NPM1 loss (Fig. 4b).

To complement ribosome profiling, we quantified protein abundance in the same samples using mass spectrometry (MS). Notably, while E2F targets, G2M checkpoint and MYC targets were positively enriched in RPFs, they were significantly negatively enriched at the protein level (Fig. 4b,c). Comparing the normalized abundance of RPFs and proteins for the genes belonging to the leading edge subsets of these pathways further confirmed this opposite enrichment (Fig. 4d,e). To interrogate this discrepancy, we exploited the positional resolution of RPF data. Across the transcriptome, NPM1 loss caused a striking induction of ribosome pausing in the first half of coding sequences (CDS) (Fig. 4f). Interestingly, genes participating in the leading edge subset of the RPF enrichment exhibited more pause inductions, while genes participating in the leading edge of the proteomics were devoid of them (Fig. 4g). Further positional analyses across all transcripts revealed a pronounced shift in ribosome occupancy after *Npm1* deletion, characterized by increased footprint density at the 5′-end and reduced at the 3′-end of the CDS (Fig. 4h). Notably, *Trp53* and *Cdkn1a* transcripts lacked this 5′-end occupancy increase, while NPM1 loss promoted resolution of ribosome pausing sites on *Trp53*, suggesting more efficient transcript translation that may contribute to the increased p53 protein levels (Extended Data Fig. 8v–x).

## NPM1 depletion triggers the integrated stress response

The widespread ribosome pauses induced upon *Npm1* deletion prompted us to investigate a potential response to protein synthesis stress. We examined key translation pathway components in the tumors and crypts of *Apc*<sup>loxP/+</sup>*Kras*<sup>G12D/+</sup> mice, which develop NPM1-deficient tumors. Phosphorylation of the translation activators eukaryotic initiation factor 4E-binding protein 1 (p-4E-BP1) and eukaryotic initiation factor 4E (p-eIF4E), and elongation inhibitory phosphorylation of eukaryotic elongation factor 2 (p-eEF2), were upregulated after NPM1 loss, while *Apc*<sup>loxP/loxP</sup>*Kras*<sup>G12D/+</sup> NPM1-deficient organoids exhibited reduced protein synthesis (Fig. 5a–d and Extended Data Fig. 9a–e). These contradictory signals suggest disrupted translation regulation, potentially inducing proteostatic stress. Therefore, we examined phosphorylation of the integrated stress response (ISR) marker eukaryotic

**Fig. 6 | ISR inhibition rescues proliferation after APC loss in NPM1-deficient tissue. a**, Representative BrdU staining on SI tissue sections from *Apc*<sup>loxP/loxP</sup>*Npm1*<sup>loxP/loxP</sup> animals treated with vehicle or ISRIB (*n* = 4 per group) and collected 4 days after induction. **b**, Quantification of BrdU⁺ cells in SI half-crypts of animals from the groups shown in **a**, separated by a dashed line from untreated *Apc*<sup>loxP/loxP</sup>*Npm1*<sup>+/+</sup> and *Apc*<sup>loxP/loxP</sup>*Npm1*<sup>loxP/loxP</sup> (*n* = 5 per group) animals used for comparison and also shown in Fig. 2c. Data were statistically assessed using a one-way ANOVA followed by Tukey's multiple comparisons test. **c**, Representative BrdU staining on SI tissue sections from *Apc*<sup>loxP/loxP</sup>*Kras*<sup>G12D/+</sup>*Npm1*<sup>+/+</sup> (*n* = 4) and *Apc*<sup>loxP/loxP</sup>*Kras*<sup>G12D/+</sup>*Npm1*<sup>loxP/loxP</sup> (*n* = 3) animals treated with ISRIB and collected 3 days after induction. **d**, Quantification of BrdU⁺ cells in SI half-crypts of animals from the groups shown in **c**, separated by a dashed line from untreated *Apc*<sup>loxP/loxP</sup>*Kras*<sup>G12D/+</sup>*Npm1*<sup>+/+</sup> and *Apc*<sup>loxP/loxP</sup>*Kras*<sup>G12D/+</sup>*Npm1*<sup>loxP/loxP</sup> (*n* = 4 per group) animals used for comparison and also presented in Fig. 2m. Data were statistically assessed using a one-way ANOVA followed by Tukey's multiple comparisons test. **e**, Representative BrdU (top) and p-eIF2α (Ser51) (bottom) staining on SI tissue sections from *Apc*<sup>loxP/loxP</sup>*Npm1*<sup>loxP/loxP</sup> animals treated with vehicle or PERK inhibitor (PERKi) (*n* = 4 per group) and collected 4 days after induction. **f**, Quantification of BrdU⁺ cells in the SI half-crypts of animals from the groups shown in **e**, separated by a dashed line from untreated *Apc*<sup>loxP/loxP</sup>*Npm1*<sup>+/+</sup> (*n* = 5) and *Apc*<sup>loxP/loxP</sup>*Npm1*<sup>loxP/loxP</sup> (*n* = 5) animals used for comparison and also presented in Fig. 2c. Data were statistically assessed using a one-way ANOVA followed by Tukey's multiple comparisons test. **g**, Representative BrdU staining on SI tissue sections from *Apc*<sup>loxP/loxP</sup>*Kras*<sup>G12D/+</sup>*Npm1*<sup>+/+</sup> (*n* = 4) and *Apc*<sup>loxP/loxP</sup>*Kras*<sup>G12D/+</sup>*Npm1*<sup>loxP/loxP</sup> (*n* = 3) animals treated with PERKi and collected 3 days after induction. **h**, Quantification of BrdU⁺ cells in the SI half-crypts of animals from the groups shown in **g**, separated by a dashed line from untreated *Apc*<sup>loxP/loxP</sup>*Kras*<sup>G12D/+</sup>*Npm1*<sup>+/+</sup> (*n* = 4) and *Apc*<sup>loxP/loxP</sup>*Kras*<sup>G12D/+</sup>*Npm1*<sup>loxP/loxP</sup> (*n* = 4) animals used for comparison and also presented in Fig. 2m. Data were statistically assessed using a one-way ANOVA followed by Tukey's multiple comparisons test. **i**, Representative BrdU staining on SI tissue sections from untreated *Apc*<sup>loxP/+</sup>*Kras*<sup>G12D/+</sup> (*n* = 5) and *Apc*<sup>loxP/+</sup>*Kras*<sup>G12D/+</sup>*Npm1*<sup>loxP/loxP</sup> (*n* = 6) animals, or ISRIB-treated *Apc*<sup>loxP/+</sup>*Kras*<sup>G12D/+</sup> (*n* = 4) and *Apc*<sup>loxP/+</sup>*Kras*<sup>G12D/+</sup>*Npm1*<sup>loxP/loxP</sup> (*n* = 6) animals. ISRIB treatment lasted for 72 h before sampling and started once animals were showing cancer symptoms. The red dotted lines indicate intestinal adenomas. **j**, Quantification of BrdU⁺ cells in tumors (left) and crypts (right) from the groups shown in **i**. Data were statistically assessed using a one-way ANOVA followed by Tukey's multiple comparisons test (left) and an unpaired, two-tailed *t*-test (right). **k**, Representative p-eIF2α (Ser51) staining on SI tissue sections from ISRIB-treated *Apc*<sup>loxP/+</sup>*Kras*<sup>G12D/+</sup> (*n* = 4) and *Apc*<sup>loxP/+</sup>*Kras*<sup>G12D/+</sup>*Npm1*<sup>loxP/loxP</sup> (*n* = 7) animals. ISRIB treatment lasted for 72 h before sampling and started once animals were showing cancer symptoms. The red dotted lines indicate intestinal adenomas. **l**, Quantification of p-eIF2α H-score in tumors (left) and crypts (right) from the groups shown in **k**. Data were statistically assessed using an unpaired, two-tailed *t*-test. **m**, Representative NPM1 (left), p21 (middle) and p53 (right) staining on SI tissue sections from *Apc*<sup>loxP/loxP</sup>*Npm1*<sup>loxP/loxP</sup> animals treated with vehicle or ISRIB (*n* = 4 per group) and collected 4 days after induction. The red dotted line indicates the outer edges of the intestinal crypts. **n**, Quantification of p21⁺ cells in the SI half-crypts of animals from the groups shown in **m**, separated by a dashed line from untreated *Apc*<sup>loxP/loxP</sup>*Npm1*<sup>+/+</sup> (*n* = 6) and *Apc*<sup>loxP/loxP</sup>*Npm1*<sup>loxP/loxP</sup> (*n* = 6) animals used for comparison and also presented in Fig. 3g. Data were statistically assessed using a one-way ANOVA followed by Tukey's multiple comparisons test. **o**, Quantification of p53⁺ cells in the SI half-crypts of animals from the groups shown in **m**. Data were statistically assessed using an unpaired, two-tailed *t*-test. All bar charts present data as the mean ± s.e.m. The boxes in the box plots extend from the 25th to 75th percentile, the whiskers extend to the minimum and maximum values, and the line in every box is plotted at the median. Statistically significant *P* values are shown in red. **a**,**e**, Scale bar, 50 μm. **c**,**g**,**i**,**k**,**m**, Scale bar, 100 μm.

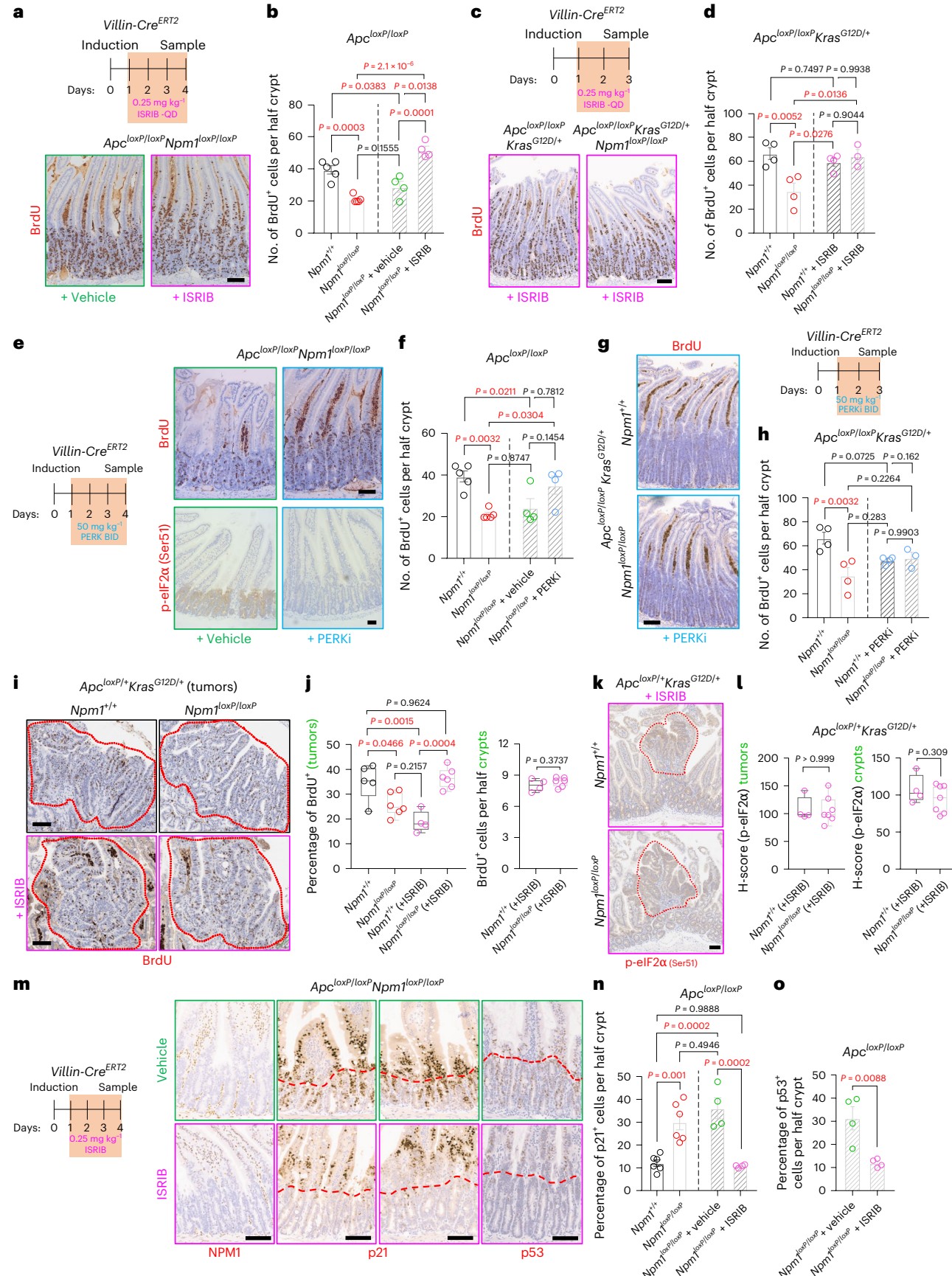

initiation factor-2α (p-eIF2α), an indicator of deregulated proteostasis. p-eIF2α levels were elevated in the tumors and intestinal crypts of *Npm1*-depleted *Apc*^loxP/+^*Kras*^G12D/+^ mice, along with upregulation of the endoplasmic reticulum (ER) stress response pathway in the proteomics data (Fig. 5e–h). Only a modest upregulation was evident in the homeostatic *Npm1*^loxP/loxP^ gut epithelium (Extended Data Fig. 9f–h). Ex vivo analysis of *Apc*^loxP/loxP^ and *Apc*^loxP/loxP^*Kras*^G12D/+^ intestinal cells and organoids revealed comparable p-eIF2α levels between genotypes and even a slight decrease in downstream ISR targets like activating transcription factor 4 in NPM1-depleted cells, while p53/p21 upregulation persisted (Extended Data Fig. 9i–l). ER stress induced by tunicamycin in *Apc*^loxP/loxP^*Kras*^G12D/+^ organoids showed that NPM1-depleted cells could activate the 'canonical' stress response, upregulating p-eIF2α, activating transcription factor 4, C/EBP homologous protein and growth arrest and DNA damage-inducible protein 34; however, p53/p21 upregulation was lost under these conditions. These data suggest that NPM1 loss causes dysregulated translation and global protein synthesis stress in response to hyperproliferative signaling, particularly after APC loss and KRAS^G12D^ activation. However, this stress may be distinct from the 'canonical' ISR.

### Inhibition of the ISR restores proliferation after NPM1 loss

To investigate which ISR step is critical for the phenotypes after NPM1 loss, we treated mice with the small-molecule inhibitor of the integrated stress response (ISRIB), which reverses the effects downstream of p-eIF2α[38]. ISRIB rescued proliferation in the *Apc*^loxP/loxP^*Npm1*^loxP/loxP^ intestines, with proliferating cells expanding further into villi compared to vehicle-treated controls (Fig. 6a,b). Similarly, ISRIB restored proliferation in *Apc*^loxP/loxP^*Kras*^G12D/+^*Npm1*^loxP/loxP^ intestines to levels matching *Npm1*-proficient controls (Fig. 6c,d). Interestingly, ISRIB treatment did not restore protein synthesis in NPM1-depleted *Apc*^loxP/loxP^*Kras*^G12D/+^ organoids, suggesting that reduced proliferation after NPM1 loss is more specific to NPM1's roles than a general protein synthesis defect (Extended Data Fig. 9m,n).

To confirm the role of p-eIF2α downstream of NPM1 loss, we inhibited the eIF2α protein kinase R (PKR)-like endoplasmic reticulum kinase (PERK) using the small-molecule inhibitor GSK2606414 (ref. 39). This inhibited eIF2α phosphorylation and phenocopied the ISRIB effects, restoring proliferation of NPM1-depleted intestinal cells in both *Apc*^loxP/loxP^ and *Apc*^loxP/loxP^*Kras*^G12D/+^ models (Fig. 6e–h).

Given the known pancreatic toxicity of GSK2606414 (ref. 40), and lack of long-term ISRIB tolerance[41], we used the licensed antidepressant trazodone hydrochloride for long-term studies. While not a specific p-eIF2α inhibitor, trazodone mimics ISRIB in counteracting p-eIF2α-mediated effects and is well tolerated in vivo[42]. Like ISRIB and PERK inhibition, trazodone hydrochloride treatment restored short-term proliferation in *Apc*^loxP/loxP^*Npm1*^loxP/loxP^ intestinal cells and significantly reduced the survival benefit from *Npm1* deletion in the *Apc*^loxP/+^*Kras*^G12D/+^ CRC model (Extended Data Fig. 9o–r). GSK2606414 exhibits off-target effects and related inhibitors can induce eIF2α kinases[43,44]. To address these limitations, we used ISRIB in subsequent experiments as a more specific tool to probe the ISR.

Given that ISR and p-eIF2α may have pro-tumorigenic effects in tumors[45,46], to complement our findings on targeting ISR during the early proliferative burst after *Apc* deletion, we treated mice with ISRIB when harboring established tumors. ISRIB supressed proliferation in NPM1-proficient tumors, but significantly increased proliferative capacity in NPM1-deficient tumors, suggesting that both NPM1 and the ISR contribute toward tumor maintenance (Fig. 6i,j). Proliferation within intestinal crypts and p-eIF2α levels in crypts and tumors were comparable in *Apc*^loxP/+^*Kras*^G12D/+^ and *Apc*^loxP/+^*Kras*^G12D/+^*Npm1*^loxP/loxP^ mice after ISRIB treatment (Fig. 6j–l).

Lastly, to examine whether ISRIB modulates the induction of p53 and p21, which are crucial mediators of tumor suppression after NPM1 loss, we quantified their expression in ISRIB-treated mice (Fig. 6m–o). Notably, ISRIB treatment prevented an increase in p21/p53 levels within NPM1-depleted intestinal crypts. Together, these data demonstrate a functional role for protein synthesis stress and the ISR in reducing WNT-mediated hyperproliferation upon NPM1 depletion, through posttranscriptional p53 activation.

### *Npm1* deletion improves survival in WNT-driven HCC

HCC is the second most commonly WNT-mutated cancer, with ~44% of cases exhibiting high WNT signaling primarily because of β-catenin-activating mutations[47]. In the liver, WNT-pathway mutations drive hepatocyte differentiation altering hepatic zonation[48], rather than promoting tumorigenesis[49]. Furthermore, unlike CRC and other cancers, the WNT–MYC module is uncoupled in the liver and *Myc* deletion does not rescue the acute effects of APC loss[48,50]. Therefore, efficient tumorigenesis in mice requires ectopic MYC expression to

**Fig. 7 | Npm1 deletion in hepatocytes attenuates WNT/MYC-driven proliferation and HCC tumor formation. a**, Representative staining for BrdU, NPM1 and p21 on liver tissue sections of *Ctnnb1*^+/(Dex3)^*R26*^lsl-MYC/lsl-MYC^ animals with or without *Npm1*^loxP/loxP^ collected at the indicated time points (four days, *n* = 3 per group, or ten days, *n* = 5 per group after high-dose AAV8.TBG. Cre viral induction). **b**, Quantification of BrdU^+^ cells in ten ×10 fields of view (FOVs) per biological replicate of *Ctnnb1*^+/(Dex3)^*R26*^lsl-MYC/lsl-MYC^ animals with or without *Npm1*^loxP/loxP^ sampled 4 days after induction (*n* = 3 per group). Data were statistically assessed using an unpaired, two-tailed *t*-test. **c**, Quantification of BrdU^+^ cells in ten ×10 FOVs per biological replicate, **d**, Percentage of liver-to-body weight ratios. **e**, HALO quantification of the percentage of p21^+^ hepatocytes from *Ctnnb1*^+/(Dex3)^*R26*^lsl-MYC/lsl-MYC^ animals with or without *Npm1*^loxP/loxP^ collected 10 days after induction (*n* = 5 per group except in **c** where *n* = 4 for the *Npm1*^loxP/loxP^ group). **f**, Significantly positively enriched Reactome pathways in RNA-sequenced *Ctnnb1*^+/(Dex3)^*R26*^lsl-MYC/lsl-MYC^*Npm1*^loxP/loxP^ mice compared to controls, sampled 10 days after induction (*n* = 4 per group). The fraction of regulated genes within each pathway is indicated by the gene ratio and the gene number (according to circle size); the circle color indicates the significance of enrichment. Overrepresentation analysis was conducted using a hypergeometric model implemented in the ReactomePA package, with significance assessed using a one-sided Fisher's exact test. **g,h**, Survival curves of male (**g**) and female (**h**) *Ctnnb1*^+/(Dex3)^*R26*^lsl-MYC/lsl-MYC^ animals induced with low-dose AAV8.TBG.Cre and sampled at the clinical endpoint. Median survival in days is indicated in parentheses. Censored mice are denoted as tick marks at the indicated times after induction. *P* values were obtained using a log-rank (Mantel–Cox) test

(*n* = 13 *Npm1*^+/+^ and 15 *Npm1*^loxP/loxP^ in **g**, and *n* = 15 per group in **h**). **i**, Percentage of *Ctnnb1*^+/(Dex3)^*R26*^lsl-MYC/lsl-MYC^ (*n* = 11) and *Ctnnb1*^+/(Dex3)^*R26*^lsl-MYC/lsl-MYC^*Npm1*^loxP/loxP^ (*n* = 18) animals with lung metastases at the endpoint. Data were compared using a two-sided Fisher's exact test. **j,k**, Percentage of liver-to-body weight ratios at the endpoint for the male and female animals shown in **g,h** (*n* = 12 *Npm1*^+/+^ and 14 *Npm1*^loxP/loxP^ in **j**, and *n* = 14 per group in **k**). **l,m**, Tumor number from male and female animals presented in **g,h** at the endpoint (*n* = 13 *Npm1*^+/+^ and 14 *Npm1*^loxP/loxP^ in **l**, and *n* = 14 per group in **m**). **n**, Representative NPM1 staining on liver tissue sections of animals at the endpoint (*n* = 4 per group). The red dotted lines indicate individual tumors. **o**, Quantification of percentage of tumors negative for NPM1 expression in *Npm1*^loxP/loxP^ animals at the endpoint (*n* = 4). **p**, Representative BrdU staining on liver tissue sections of *Ctnnb1*^+/(Dex3)^ *R26*^lsl-MYC/lsl-MYC^ animals with or without *Npm1*^loxP/loxP^ treated with ISRIB and collected 4 days after induction (*n* = 3 per group). **q**, Quantification of BrdU^+^ cells in ten ×10 FOVs from liver sections of animals from the groups shown in **p**, separated by a dashed line from untreated animals of the same genotypes (*n* = 3 per group), used for comparison and also presented in **b**. **b**–**e,j**–**k,m**, Data were statistically assessed using an unpaired, two-tailed *t*-test. **l**, Data were statistically assessed using an unpaired, two-tailed Mann–Whitney *U*-test. **q**, Data were statistically assessed using a one-way ANOVA followed by Tukey's multiple comparisons test. **b**–**e**, The bar charts present data as the mean ± s.e.m.; the boxes in the box plots extend from the 25th to 75th percentile, the whiskers extend to the minimum and maximum values, and the line in every box is plotted at the median. Statistically significant *P* values are shown in red. **a,p**, Scale bar, 100 μm. **n**, Scale bar, 500 μm.

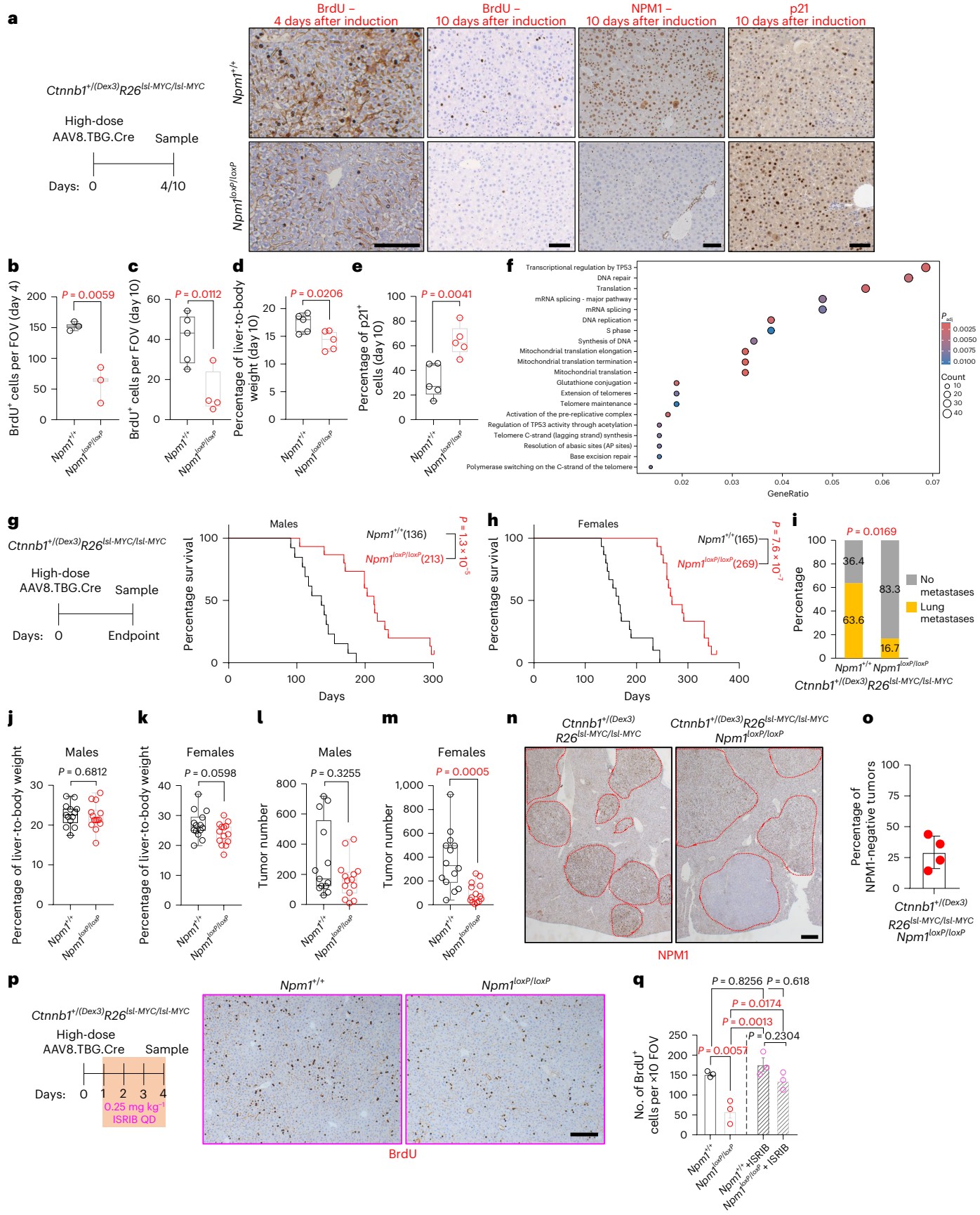

synergize with β-catenin to drive HCC; *MYC* amplification is common in β-catenin-mutated human HCC[49,51].

*Npm1* expression was marginally changed after APC loss or β-catenin activation (*Ctnnb1*[+/(Dex3)]) in the liver, but significantly enhanced after conditional *MYC* transgene overexpression (*R26*[lsl-MYC/lsl-MYC]) (Fig. 1d and Extended Data Fig. 10a). Nevertheless, *Npm1* deletion suppressed short-term hepatocyte hyperproliferation after *Apc* loss (Extended Data Fig. 3d), confirming a WNT effector role in WNT-permissive tissues. Similarly, *NPM1* is overexpressed in human HCC (Fig. 1e), with higher expression in *MYC*-amplified cases; it correlates with WNT activation and is associated with significantly worse prognosis (Extended Data Fig. 10b–h).

*Npm1* deletion in adult liver did not disrupt homeostasis. Liver tissue exhibited normal histology, zonation, hepatocyte proliferation and function after long-term NPM1 loss (Extended Data Fig. 10i–l). To model WNT-driven HCC, we generated mice with an activating β-catenin mutation (*Ctnnb1*[+/(Dex3)]) combined with a *MYC* transgene (*R26*[lsl-MYC/lsl-MYC]) to induce hepatocyte proliferation and HCC formation. Induction with high AAV8.TBG.Cre viral titers triggered recombination across hepatocytes, leading to increased proliferation and hepatomegaly, both of which were suppressed after concurrent NPM1 depletion (Fig. 7a–d). NPM1 loss modestly suppressed WNT-driven hepatocyte differentiation, indicated by reduced glutamine synthetase staining, while reduced proliferation after *Apc* deletion was further confirmed (Extended Data Fig. 10m–q). In an oncogenic *BRAF*-driven model, proliferation remained unchanged by NPM1 depletion 4 days after induction, and modestly reduced by 8 days (with p21 levels increased), confirming NPM1 as a key effector of elevated WNT signaling (Extended Data Fig. 10r–v).

NPM1 depletion induced p21 expression in the liver irrespective of β-catenin/MYC or oncogenic *BRAF* activation (Fig. 7a,e and Extended Data Fig. 10t,v). In the β-catenin/MYC model, pathways related to p53 regulation, DNA repair and the cell cycle were among the most enriched after NPM1 loss (Fig. 7f and Extended Data Fig. 10w), indicating a conserved response to NPM1 depletion in both the gut and liver. Consistent with this, *TP53* is more frequently inactivated in HCCs with *NPM1* deletions compared to those with intact *NPM1*, recapitulating our observations in human CRC (Extended Data Fig. 10x,y).

Low AAV8.TBG.Cre viral titers induce recombination in few hepatocytes, permitting clonal outgrowth and tumor formation. NPM1 loss significantly extended survival in mice collected at the endpoint, with comparable liver mass and reduced lung metastases (Fig. 7g–k). *Npm1*[loxP/loxP] mice also developed fewer tumors, with only ~25% of them being NPM1[−] (Fig. 7l–o). Finally, to assess whether ISR activation directly mediates reduced proliferation after NPM1 loss in the liver, we treated β-catenin/MYC mice with ISRIB (Fig. 7p). Consistent with the gut findings, ISRIB restored hepatocyte hyperproliferation in *Npm1*[loxP/loxP] animals (Fig. 7p,q), suggesting that ISR activation underlies the proliferative defect induced by NPM1 loss in both tissues.

To conclude, we describe an overarching mechanism whereby NPM1 expression is upregulated upon oncogenic insult and is essential to support WNT-driven hyperproliferation and tumor initiation. Mechanistically, this relies on attenuation of both the p53 pathway and the ISR, which become activated upon NPM1 depletion. Importantly, this holds true for multiple WNT-permissive tissues with increased mutational load, solidifying NPM1 as a promising therapeutic target.

## Discussion

MYC, a central oncogenic driver[1] and WNT mediator, is challenging to target directly. In this study, we identified its target gene *Npm1* as a therapeutic candidate in WNT-permissive tissues. While essential during development, NPM1 proved dispensable in adult tissues but critical for WNT-driven hyperproliferation and tumorigenesis. Notably, NPM1 depletion suppressed *Kras*-mutant WNT-driven CRC and restricted WNT-driven liver cancer, both high-mortality diseases with limited treatment options[4,6,52].

NPM1 loss disrupted translation in WNT-activated cells by inducing ribosome pausing on mRNA CDS. While the mechanism remains unclear, NPM1 association with active ribosomes[53,54] suggests it may directly modulate translation. This could be crucial during hyperproliferative signaling, where increased translational demand risks proteotoxic stress and growth arrest. Our data highlight NPM1 as a potential key regulator of translational homeostasis under these conditions.

NPM1 also binds small nucleolar RNAs and mediates site-specific 2-O′-methylation of rRNAs via fibrillarin, reducing translation of select mRNAs[9]. Fibrillarin was also upregulated within WNT-permissive tissues (Fig. 1c), suggesting a potential link between dysregulated rRNA modifications and oncogenic signaling. p53 represses fibrillarin expression, preventing rRNA methylation changes that enhance oncogene translation (for example, *MYC*) by affecting ribosome fidelity[55]. Conceivably, NPM1 depletion may activate p53 to maintain translation control, thus abrogating transformation.

ISR inhibition restored proliferation and suppressed the p53 response after NPM1 loss in WNT-activated tissues. However, this may reflect a noncanonical ISR, as typical targets downstream of p-eIF2α were not upregulated and ISRIB treatment failed to restore protein synthesis in NPM1-depleted organoids. Recent studies highlight the plasticity of ISR activation and downstream effects[56]. This may extend to NPM1-related ISR, where *Trp53* notably escapes ribosome pausing after NPM1 loss. *Trp53* deletion independently rescued NPM1 loss phenotypes, implicating ISR activation causally. The PERK-regulated p53 isoform (p47), identified under ER stress, supports this functional connection[57]. After NPM1 loss, p53 activation is subtle, avoiding full p53 transcriptional program activation, permitting homeostatic proliferation but restraining the hyperproliferation required for transformation. p53 stabilization upon NPM1 loss has been observed previously, and concomitant loss of both proteins accelerates leukemogenesis in mice[7,8,32]. Our findings propose a link between p53 and NPM1, mediated by proteostatic stress from increased protein synthesis. Of note, p53-deficient, *KRAS*-mutant non-small-cell lung cancer cells were sensitive to NPM1 depletion, indicating that this interplay may be context-dependent[21].

*NPM1*-mutated leukemia is generally associated with favorable patient outcomes and improved treatment response[58,59]. Disrupting NPM1 oligomerization sensitizes cells to DNA-damaging agents[60]. NPM1 is involved in the DNA damage response[22], with our data demonstrating increased γH2AX after NPM1 loss. We found that NPM1 depletion particularly affects transformed cells, suggesting that aside from suppressing hyperproliferation, targeting NPM1 could also enhance susceptibility to additional stress, which could be exploited therapeutically in combination therapies.

Lastly, biosynthetic capacity via increased activity of RNA polymerase I in tumor cells that are enriched in nucleolar, protein folding and ribosome biogenesis pathways, is critical for sustained growth in CRC[61]. Standard therapies already target RNA synthesis and nucleolar function[62–65]. The enrichment of core nucleolar factors, NPM1 and fibrillarin, in WNT-permissive tissues, suggests an increased reliance on protein synthesis in WNT-driven CRC. Co-targeting translation control components can sensitize *KRAS*-mutant CRC to standard treatment[6]. As deregulated translation and protein synthesis emerge as a therapeutic strategy, NPM1 stands out as a target, particularly affecting transformed cells that are highly dependent on these processes for rapid growth. Our data revealed that NPM1 couples major oncogenic and tumor suppressor pathways via translational control, offering a potential avenue toward targeting difficult-to-treat cancer drivers and tumor heterogeneity.

## Online content

Any methods, additional references, Nature Portfolio reporting summaries, source data, extended data, supplementary information, acknowledgements, peer review information; details of author contributions

and competing interests; and statements of data and code availability are available at https://doi.org/10.1038/s41588-025-02408-7.

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

## Methods

All animal experiments were performed in accordance with UK Home Office regulations (project licenses 70/8646 and PP3907577), with the approval, and under the oversight, of the animal welfare and ethical review board of the University of Glasgow. All experiments performed adhered to Institutional guidelines in full. Key experimental procedures are outlined below; details of additional methods are available in the Supplementary Note.

### In vivo studies

Mice were maintained in conventional cages in a specific pathogen-free facility of constant temperature between 19 °C and 23 °C, and 55 ± 10% humidity, under a 12 h light–12 h dark cycle, and ad libitum access to food and water. Genotyping was performed on ear punch biopsies by TransnetYX. All experiments were conducted on mice that had been bred on a C57BL/6 background for at least three generations ($n \geq 3$) for the homeostasis experiments, and at least four generations ($n \geq 4$) for the CRC models. No formal randomization was performed and researchers were not blinded to the experimental groups.

For the intestinal experiments, male and female mice were induced between 8 and 15 weeks of age or once they had reached 20 g of body weight. Recombination under the villin promoter (referred to as *Villin-Cre^ERT2*) was induced via intraperitoneal injection of 2 mg tamoxifen (cat. no. T5648, Sigma-Aldrich) dissolved in corn oil (cat. no. C8267, Sigma-Aldrich)[66]. Mice for the short-term experiments received two doses of tamoxifen on consecutive days and were sampled at 4 days after induction, with the exception of mice bearing two *Apc* floxed alleles[67] and oncogenic *Kras^G12D* (ref. 68) activation (*Apc^loxP/loxP Kras^G12D/+*), which were induced with a single tamoxifen injection and sampled after 3 days. The tumorigenesis experiments were induced by a single 2-mg tamoxifen intraperitoneal injection and samples were collected either at predetermined time points or when they reached the clinical endpoint as determined by hunching, paling or weight loss in the case of tumor-bearing mice. Clinical endpoints were not exceeded at any point during this study. Recombination under the *Lgr5* promoter (*Lgr5-Cre^ERT2*) was induced with a 3-mg tamoxifen intraperitoneal injection[69]; samples were collected once they had reached the clinical endpoint as defined above. The *Trp53* floxed[70], *Cdkn1a* KO[71], intracellular expression of the NOTCH1 transgene (*Rosa26^NIicd*)[72], *Braf^V600E* (ref. 73) and *Braf^LSL-V637E* (ref. 74) alleles have been described previously.

Whole-body *Npm1* depletion in adult tissues was achieved using *Cre^ERT2* expression induced from the *Rosa26* locus (*Rosa26^CreERT2*)[75], and an initial 3-mg tamoxifen followed by three doses of 2-mg tamoxifen intraperitoneal injections on consecutive days. For whole-body deletion of floxed *Apc* alleles[76] a single 3-mg intraperitoneal injection was used. For hepatocyte-specific genetic recombination of the *Npm1^loxP/loxP* alleles, male and female mice between 2 and 4 months of age were induced with adeno-associated virus (AAV) expressing Cre under the control of the thyroxine binding globulin (TBG) promoter (AAV8.TBG.Cre). AAV.TBG.PI.Cre.rBG was a gift from J. M. Wilson (viral prep 107787-AAV8, Addgene; http://n2t.net/addgene:107787; research resource identifier: Addgene_107787). Viral delivery was conducted via intravenous tail vein injections of $2 \times 10^{11}$ genome copies (GCs) per mouse. To drive liver proliferation and tumorigenesis, β-catenin was activated via conditional excision of exon 3 (*Ctnnb1^+/(Dex3)*)[77], in combination with human *MYC* transgene expression from the *Rosa26* locus (*R26^lsl-MYC/lsl-MYC*)[78]. Short-term experiments in the liver tumor models were performed only in male mice because of large cell proliferation variability between the sexes. Samples were collected at specified time points, usually at 4 or 10 days after induction with $2 \times 10^{11}$ AAV8.TBG.Cre GC per mouse. For the long-term experiments assessing tumor formation, mice of both sexes were induced with a lower dose of $6.4 \times 10^8$ GC per mouse, monitored for tumor growth; samples were collected at the clinical endpoint as determined by abdominal swelling, loss of body conditioning or weight. In all tumor model cohorts, mice that had to be

euthanized for reasons other than tumor burden (for example, epidermal wounds) or significantly surpassed median survival were censored.

Cell proliferation was assessed using BrdU (cat. no. RPN201, Amersham Biosciences) incorporation, which was administered via intraperitoneal injection (250 µl) 2 h before sampling. Complete blood count analysis was carried out on blood samples collected from animals under terminal anesthesia, cervically dislocated, by cardiac puncture and in tubes containing EDTA anticoagulant (Sarstedt) using the Pro-Cyte Dx (IPU v.00-33-Build51) hematology analyzer (IDEXX). Blood plasma biochemistry was performed on blood samples collected in lithium-heparin-coated tubes (Sarstedt) and after plasma separation using centrifugation at 2,350*g* for 10 min at room temperature stored at −80 °C.

### *Npm1* floxed allele generation

A conditional allele of the *Npm1* gene (Ensembl ID: ENS-MUSG00000057113 in genome assembly GRCm39; Ensembl Release 110) was generated using gene targeting in mouse embryonic stem cells (mESCs). This modified allele introduces *loxP* sites flanking exons 2–6 (ENSMUSE00000473268, ENSMUSE00000472357, ENSMUSE00000463763, ENSMUSE00000134295 and ENSMUSE00000251772) of the mouse *Npm1* gene (Ensembl Transcript ID: Npm1-201; ENSMUST00000075641.10) (see the schematic depiction in Extended Data Fig. 1g).

To generate this allele, a targeting vector for the *Npm1* locus (PG00256_Z_5_E12) was imported from EUMMCR[79]. The imported construct was linearized and used to transfect 129/P2-derived HM1 mESCs[80] by electroporation. After transfection, mESCs were selected under G418 (250 µg ml⁻¹) and resistant colonies were picked and screened for the correct targeting of the *Npm1* gene using long-range PCR (Expand Long Template PCR System, Roche). Screening was carried out at both the 5′ and 3′ ends of the targeting vector, from within the neomycin selection cassette to endogenous *Npm1* sequences outside the homology arms. Oligonucleotide sequences used to screen cells, and to confirm the presence of the isolated 3′ *loxP* site using PCR, are provided in Supplementary Table 1.

After identification of correctly targeted clones for the *Npm1* allele, mouse lines were derived by injection of targeted mESCs cells into C57BL/6J blastocysts according to standard protocols. After breeding of chimeras, germline offspring were identified according to coat color; the presence of the modified allele was confirmed with the 3′ *loxP* primers described above. Mice were subsequently crossed with a C57Bl/6J mouse strain expressing FLPe (Tg(ACTFLPe)9205Dym) to delete the selectable marker by recombination at the *FRT* sites[81]. Deletion of the selectable marker was confirmed using PCR across the remaining *FRT* site, using the oligonucleotides provided in Supplementary Table 1. After successful validation of the mouse strain carrying the *Npm1* target allele, genotyping was subsequently carried out by the commercial genotyping service provider (TransnetYX).

### Drug treatments

ISRIB (cat. no. SML0843, Sigma-Aldrich) was prepared as a 1 mg ml⁻¹ stock solution in dimethylsulfoxide and subsequently prepared as a 1:20 dilution in a vehicle consisting of 50% PEG 400 and 45% 0.9% saline solution. A 0.25 mg kg⁻¹ dose was administered in vivo once daily via intraperitoneal injection. Trazodone hydrochloride (cat. no. T6154, Sigma-Aldrich) was dosed once daily via intraperitoneal injection at 40 mg kg⁻¹, dissolved in 0.9% saline solution. Mice were dosed with PERKi (GSK2606414, cat. no. S7307, Selleck Chemicals) twice daily via oral gavage at 50 mg kg⁻¹, which was dissolved in 0.5% hydroxypropyl methylcellulose/0.1% Tween-80 vehicle.

### TMA

NPM1 protein expression was assessed in a retrospectively collected cohort of patients with stage I–IV CRC ($n = 787$). The cohort consisted

 

of patients with CRC undergoing surgical resection with curative intent within Greater Glasgow and Clyde National Health Service between 1997 and 2013. Tissue was previously constructed into TMA consisting of 0.6-mm cores in triplicate for each patient to account for tumor heterogeneity. Patients were excluded from the study due to mortality within 30 days of surgery or administration of neoadjuvant therapy. Data are deposited with the Glasgow Safehaven (no. GSH21ON009) and ethical approval was in place for the study (no. MREC/01/0/3).

Immunohistochemical staining was performed as described in the Supplementary Note. Staining was assessed in tumor cells semi-quantitively using HALO (v.3.6.4134) (Indica Labs) via weighted histoscore. Scores were validated through correlation analysis with manual scores by two observers (G.K., K.P.) for 10% of cases. Averaged scores were input into a clinical database in SPSS (v.28, IBM Corporation). The optimal cutoff point for high-expression and low-expression groups was determined using the Survminer package in RStudio (v.2023.12.1, Posit) based on cancer-specific survival.

## Human tumor data and analyses

For the TCGA pan-cancer analysis in Fig. 1e,f and Extended Data Fig. 1d,f, the COAD/READ analysis in Figs. 1g and 3r–t, Extended Data Figs. 1e and 5f–h, and the LIHC analysis in Extended Data Fig. 10b–g,x,y, the harmonized TCGA Pan-Cancer Atlas data was downloaded from UCSC Xena[82]. For the GSEA, the gsva function from the Python package GSEApy[83] was used, which reimplemented the gene set variation analysis[84]. Specifically, when comparing between tumors and adjacent normal samples (Fig. 1e,f and Extended Data Fig. 1d), gsva was run on the combined expression matrix containing both tumor and adjacent normal samples for each tumor type separately. For Fig. 1g, and Extended Data Figs. 1e, 5f–h and 10d–g, gsva was run on the expression matrix containing only tumor samples from the corresponding tumor type to focus on the variation within tumor samples alone. For stratifying tumors based on *APC* mutation status, we defined damaging mutations as nonsense mutations or frameshift indels. For copy number status, we used the absolute copy number values calculated by GISTIC2 (ref. 85) and interpreted them as follows: −2 (deep deletion); −1 (shallow deletion); 0 (diploid); 1 (low-level gain); 2 (high-level amplification). Of note, NPM1 is never deep-deleted in COAD/READ or LIHC. Survival analysis of patients with HCC with high *NPM1* mRNA expression presented in Extended Data Fig. 10h was performed using Kaplan–Meier Plotter[86].

The microarray dataset (GSE39582)[87] from patients with colon cancer consisting of n = 566 primary tumors and n = 19 adjacent normal tissue samples was downloaded from the Gene Expression Omnibus (GEO), followed by probe-to-gene collapse using maxMean method WGCNA[88] (v.1.70-3) R package, which takes the probe with the highest average score across the samples as gene expression. Tumor samples were called for PDS classification with the PDSclassifier[30] (v.1.0.0) R package. To measure the proliferative index, the ProliferativeIndex[89] (v.1.0.1) R package was used. The gene signature DNA repair (Hallmark) was accessed using the msigdbr[90] (v.7.4.1) R package to obtain the single-sample GSEA score using the GSVA[84] (v.1.42.0) R package. The replication stress score was generated as stated previously[30]. For visualizations, a list of R packages was used, which consists of ggplot2 (v.3.5.1), ggpubr (v.0.6.0) and ggbeeswarm (v.0.7.2) within R (v.4.3.3).

## RNA-seq and pathway enrichment analyses

RNA quality was tested using an Agilent 220 TapeStation with RNA screentape and only samples with an RNA integrity number > 6 were used for the downstream analysis. Libraries for cluster generation and RNA-seq were prepared according to a previously described method[91], using a TruSeq RNA Sample Prep Kit v2 (Illumina), then run on an Illumina NextSeq system using the 75 cycles High Output Kit (2 × 36 cycles, paired-end reads, single index). Raw sequence quality was assessed using FastQC (v.0.11.8), then sequences were trimmed to remove adapter sequences and low-quality base calls, defined as those

with a Phred score of less than 20, using Trim Galore (v.0.6.4). Trimmed sequences were aligned to the mouse genome build GRCm38.98 using HISAT2 (v.2.1.0); raw counts per gene were determined using Feature-Counts (v.1.6.4). Differential expression analysis was performed using the R package DESeq2 (v.1.22.2), using a negative binomial generalized linear model, with significance assessed using a Wald test and Benjamini–Hochberg multiple testing correction. Reactome pathway enrichment was performed using enrichPathway function from the R package ReactomePA (v.1.36.0). For GSEA in the *Rosa26^{CreERT2}Apc^{+/+}* versus *Rosa26^{CreERT2}Apc^{loxP/loxP}*, the following Reactome or Hallmark pathways are shown: Hallmark Myc V1, Hallmark Myc V2, 5991099_Translation, 5991458_Translation_initiation_complex_formation, 5991097_Cap-dependent_Translation_Initiation, 5991098_Eukaryotic_Translation_Initiation, 5991746_WNT_mediated_activation_of_DVL, 5991743_TCF_dependent_signaling_in_response_to_WNT and 5991561_Signaling_by_Wnt. The additional WNT signatures are as Watanabe Wnt[92], Michel a Wnt[93], Van der Flier a Wnt and Van der Flier b Wnt[94], and Sansom Wnt[95]. The last two were also referred to throughout the manuscript as 'WNT signature a' and 'WNT signature b', respectively.

## Ribosome profiling (Riboseq)

Pellets for Riboseq from intestinal epithelial extracts were prepared as described in the Supplementary Note, were placed at −20 °C for 30 min, then on ice for 5 min before being lysed in 550 μl of ice-cold lysis buffer (15 mM Tris-Cl, pH 7.5, 150 mM NaCl, 15 mM MgCl$_2$, 100 μg ml$^{-1}$ cycloheximide, 1% Triton X-100, 0.05% Tween-20, 2% *n*-dodecyl-β-D-maltopyranoside detergent, 0.5 mM dithiothreitol, 1× cOmplete mini (cat. no. 04693124001, Roche)) and 1,000 U ml$^{-1}$ Ribolock (cat. no. EO0382, Thermo Fisher Scientific). Lysis was achieved by resuspension and subsequent mechanical stress by shear force, passing the lysate 5× through a 21-G needle, while keeping the tube on ice as much as possible. Lysates were then cleared using centrifugation (16,000g, 5 min, 4 °C). An aliquot of 25 μl undigested cleared lysates was supplemented with 1 ml TRIzol (cat. no. 15596026, Invitrogen) for total cytoplasmic RNA extraction. TRIzol extraction was performed according to the manufacturer's instruction.

Then, 450 μl of cleared lysate were transferred to a fresh tube and digested with 5 μl RNase I (cat. no. AM2295, Ambion) at 22 °C for 15 min with gentle agitation (600 r.p.m.). Digestion was stopped with 10 μl of SUPERase In (cat. no. AM2696, Invitrogen). Digested lysates were loaded onto a 10–50% sucrose gradient and ultracentrifuged for 2 h at 256,800g at 4 °C in a SW 40 Ti rotor (Beckman Coulter). Spun samples were fractionated on a Triax Gradientmaster and fractions corresponding to monosomes were extracted in acid-phenol:chloroform, pH 4.5 (with IAA, 125:24:1, Invitrogen), washed twice with chloroform and precipitated with 2 μl glycogen, 1/10th volume of 3 M NaOAc (pH 5.2) and an equal volume of isopropanol. RNA was size-selected on a 15% Urea-TBE gel. rRNA depletion was performed with a custom set of complementary biotinylated DNA oligonucleotides (Supplementary Table 1), subsequently captured with Dynabeads MyOne Streptavidin C1 beads (cat. no. 65001, Invitrogen). After treatment with T4 Polynucleotide Kinase (cat. no. M0201L, New England Biolabs) and final purification with acid-phenol:chloroform and isopropanol precipitation as above. RPF libraries were prepared using the NEXTflex Small RNA Kit v3 (PerkinElmer) according to the manufacturer's instructions, using all purified material and 13 PCR cycles in the amplification step. Alternative step F and step H2 were used. Total cytoplasmic RNAs, extracted from undigested lysate, were depleted of rRNA with RiboCop V2 (Lexogen); then libraries were prepared using the CORALL kit V1 (Lexogen), all according to the manufacturer's instructions; 900 μg of total cytoplasmic RNA were used as input for the RiboCop. Thirteen PCR cycles were used in the library amplification stage.

RPF and total libraries were quantified using DNA High Sensitivity Qubit assays (Invitrogen) and size-checked using D1000 High-sensitivity ScreenTape (Agilent Technologies). After equimolar

pooling, libraries were single-end sequenced on a NextSeq 500 High Output 75 cycle kit (Illumina).

## Riboseq analysis

Riboseq analysis was performed according to the publicly available pipeline at the Bushell's lab GitHub page (https://github.com/Bushell-lab/Ribo-seq). The custom scripts listed below are available on the GitHub page. All analyses relative to transcript isoforms and their characteristics were performed using the Gencode vM27 dataset for reference. Briefly, both total cytoplasmic RNA reads and RPF reads had adapter sequences removed with cutadapt and unique molecular identifiers were appended to FASTA headers using UMItools[96]. Then, total RNA reads were aligned to protein-coding transcripts using Bowtie 2 (ref. [97]) and subsequently deduplicated, again using UMItools. Isoform quantification for each gene was performed using rsem-calculate-expression[97]; then, the most abundant transcript (MAT) per gene was extracted with a custom R script. A FASTA file for the MATs was then generated with a custom Python script. RPF reads were aligned to rRNA, tRNA and mitochondrial sequences to assess contamination of input RNA in the library preparation procedure using BBMap (sourceforge.net/projects/bbmap/). The remaining reads were aligned to the MAT FASTA with BBMap and then deduplicated using UMItools. Reads of length 27–38 were extracted and quality-checked using custom Python and R scripts. Finally reads for the protein CDS were extracted for each read length identified as correctly representing RPFs on the basis of quality control (Extended Data Fig. 8).

Differential expression analysis was performed with DESeq2 (ref. [98]). During differential expression analysis, batch correction was performed, where the batch corresponded to the day in which the epithelial extractions were processed (from lysis to acid-phenol:chloroform extraction). Any transcript with average reads across all samples lower than ten was excluded from differential expression analysis and downstream processing. Of the 20,622 transcripts in the MAT table, 10,861 were retained for cytoplasmic RNA and 11,279 were retained for RPFs (Supplementary Tables 2–5). Merging the data tables to evaluate changes at RPF and cytoplasmic RNA concomitantly retained 10,261 transcripts. Given that we had previously selected a unique transcript per gene, during the MAT analysis, 10,261 unique genes are represented each by one transcript. Functional analyses were performed with the fGSEA package[99]. Pause analysis was performed as described previously[100]. For each condition, a pause site was defined as an RPF peak whose height was ten times greater than the average RPF peak height (excluding zeros) on the same mRNA. The change in peak height between conditions was calculated as the difference between RPFs normalized to mRNA abundance. Pause sites were then classified as 'resolved' or 'induced' if the decrease or increase, respectively, in their peak height was ten times greater than the average change across the same mRNA. Pauses that were neither induced nor resolved were classified as 'sustained'. For subsequent analyses, pauses occurring at the translation start site were excluded.

## Proteomics

Pellets from epithelial extracts (Supplementary Note) from the same animals used for Riboseq were lysed in 500 µl of 4% SDS, 100 mM Tris-HCl, pH 7.5, prepared in mass spectrmetry-grade water. Samples were boiled for 60 s at 95 °C, then passed through 23-G needles 7× times to fully lyse cells. Samples were then sonicated for five cycles 10 s on/10 s off with an immersion probe at 50% (FB 50, Thermo Fisher Scientific), then centrifuged at 16,000g for 10 min at room temperature. The supernatant was collected in a fresh tube and protein concentration quantified with a bicinchoninic acid assay (Pierce BCA Protein Assay Kit, Thermo Fisher Scientific).

Then, 10 µg of proteins per sample were reduced with 10 mM dithiothreitol and subsequently alkylated in the dark with 55 mM iodoacetamide at room temperature. Alkylated proteins were precipitated by adding four volumes of acetone at −20 °C overnight. Washed pellets were reconstituted in 50 µl of 200 mM HEPES and digested first with endoproteinase Lys-C (1:33 ratio enzyme:lysate) for 1 h, followed by trypsin, overnight (1:33 ratio enzyme:lysate). The digested peptides from each experiment were differentially labeled using the TMTpro 16plex reagent (Thermo Fisher Scientific). The reaction was carried out at room temperature for 2 h. Fully labeled samples were mixed in equal amounts and desalted using a 50-mg Sep Pak C18 reverse-phase solid-phase extraction cartridges (Waters). TMT-labeled peptides were fractionated using high-pH reverse-phase chromatography on a C18 column (150 × 2.1 mm i.d.; Kinetex EVO (5 µm, 100 Å)) on a high-performance liquid chromatography system (LC 1260 Infinity II, Agilent Technologies). A two-step gradient was applied, from 1–28% B in 42 min, then from 28–46% B in 13 min to obtain a total of 21 fractions for MS analysis.

Peptides resulting from all samples were separated using nanoscale C18 reverse-phase liquid chromatography using an EASY-nLC II 1200 (Thermo Fisher Scientific) coupled to an Orbitrap Q Exactive HF (Thermo Fisher Scientific). Elution was carried out at a flow rate of 300 nl min⁻¹ using a binary gradient with buffer A (2% acetonitrile) and B (80% acetonitrile), both containing 0.1% formic acid. Samples were loaded with 6 µl of buffer A into a 50-cm fused silica emitter (New Objective) packed in-house with ReproSil-Pur C18-AQ, 1.9 µm resin (Dr. Maisch). For both systems, the packed emitter was kept at 50 °C using a column oven (Sonation) integrated into the nanoelectrospray ion source (Thermo Fisher Scientific); the Xcalibur software (v.4.1.31.9, Thermo Fisher Scientific) was used for data acquisition. Peptides were eluted using different gradients optimized for three sets of fractions: 1–7; 8–15; and 16–21 (ref. [101]). Each fraction was acquired for a duration of 190 min. A full scan over mass range of 375–1400 $m/z$ was acquired at 60,000 resolution at 200 $m/z$, with a target value of $3 \times 10^6$ ions for a maximum injection time of 20 ms. Higher energy collisional dissociation fragmentation was performed on the 20 most intense ions selected within an isolation window of 0.8 $m/z$. Peptide fragments were analyzed in the Orbitrap at a 45,000 resolution.

## Proteomics data analysis

The MS raw data were processed with the MaxQuant software[102] (v.1.6.14.0) and searched with Andromeda search engine[103] querying SwissProt[104] *Mus musculus* (63,668 entries). First and main searches were performed with precursor mass tolerances of 20 p.p.m. and 4.5 p.p.m., respectively, and MS/MS tolerance of 20 p.p.m. The minimum peptide length was set to six amino acids; specificity for trypsin cleavage was required, allowing up to two missed cleavage sites. The peptide, protein and site false discovery rate was set to 1%. Modification by iodoacetamide on cysteine residues (carbamidomethylation) were specified as fixed, whereas methionine oxidation and N-terminal acetylation modifications were specified as variable. For the proteome analysis, MaxQuant was set to quantify on 'Reporter ion MS2'; TMT-16plex was chosen as the isobaric label. Interference between TMT channels was corrected by MaxQuant using the correction factors provided by the manufacturer. The 'Filter by PIF' option was activated and a 'reporter ion tolerance' of 0.003 Da was used.

The proteinGroups.txt file from the MaxQuant output was used for protein quantitation analysis using the Perseus software[105] (v.1.6.15.0). The 'reverse', 'potential contaminants' and 'only identified by site' protein, as specified in MaxQuant, were removed, as well as protein groups identified with no unique peptides. Only proteins robustly quantified in all replicates in at least one group were allowed in the list of quantified proteins. Significantly different proteins were selected using a permutation-based Student's $t$-test with the false discovery rate set at 5%.

Functional analyses were performed with the fgsea package as for Riboseq, preranking genes by test difference (Supplementary Tables 2–6). If a protein group contained more than one protein, all available gene symbols were kept tied in the ranking.

## Statistics and reproducibility

Statistical analyses were performed with Prism (v.7.0.4, GraphPad Software). To ensure appropriate statistical comparison test selection, data were initially assessed to determine whether they followed a Gaussian distribution. Sample sizes of eight or more were assessed using the D'Agostino-Pearson omnibus K2 test and those less than eight with the Shapiro–Wilk test. Statistical significance of the parametric data was assessed using a two-tailed $t$-test, while nonparametric data were compared using a two-tailed Mann–Whitney $U$-test. Comparisons involving more than two groups were performed using an ANOVA, followed by appropriate post hoc tests as indicated. Survival data were statistically compared using the log-rank (Mantel–Cox) test. The null hypothesis was rejected and statistical significance was assumed for $P < 0.05$. Data are presented as the mean ± s.e.m. unless otherwise indicated. Sample sizes, specific $P$ values and tests performed for each experiment are indicated in the corresponding figures or figure legends. From the TMA cohort, patients were excluded from the study because of mortality within 30 days of surgery or administration of neoadjuvant therapy. As part of qaulity control in the Riboseq analysis, any transcript with average reads across all samples lower than ten was excluded from any downstream analysis. No other data were excluded from the analyses. The experiments were not randomized. The investigators were not blinded to allocation during the experiments and outcome assessment. No formal statistical methods were used to predetermine sample sizes as part of this study. All in vivo cohort sizes were determined based on power analyses in studies previously carried out in similar models, respecting the limited use of animals in line with the 3R system: replacement, reduction, refinement.

## Reporting summary

Further information on research design is available in the Nature Portfolio Reporting Summary linked to this article.

## Data availability

The RNA-seq data generated in this study have been deposited at the GEO under accession numbers GSE230110, GSE309379 and GSE250047. The ribosome profiling data have been deposited at GEO under the accession number GSE249958. The proteomics data have been deposited at the ProteomeXchange Consortium via the PRIDE partner repository with the dataset identifier PXD062969. The murine liver *Npm1* expression data presented in Extended Data Fig. 10a were derived from RNA-seq data available at the GEO under accession number GSE230137. Human cancer analyses were conducted using data that are in whole or part based on data generated by the TCGA Research Network (www.cancer.gov/tcga), as well as the publicly available colon cancer microarray dataset GSE39582 (ref. 87). Source data are provided with this paper.

## Code availability

The code used in this project for the Ribo-seq analysis and integration with the proteomics data is available on the project-specific Zenodo repository page (https://doi.org/10.5281/zenodo.17187088)[106]. The code is based on the Bushell lab pipeline available on the lab's GitHub page (https://github.com/Bushell-lab/Ribo-seq).

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

## Acknowledgements

We thank the Core Research Services at the Cancer Research UK (CRUK) Scotland Institute, particularly the Biological Services Unit, Histology and Molecular Technologies services for technical support. We also thank C. Winchester for critical assessment of the manuscript. This work was supported by: CRUK core funding to the CRUK Scotland Institute (no. A31287); CRUK core funding to the CRUK Glasgow Centre (no. A25142); CRUK core funding to the CRUK Scotland Centre (no. CTRQQR-2021\100006); CRUK core funding to the lab of C.M. (no. A29801); CRUK core funding to the lab of M.B. (no. A29252); and CRUK core funding to the lab of O.J.S. (nos. A21139 and DRCQQR-May21\100002). The J.L.Q. lab is funded by the Mazumdar Shaw Chair; CSO fellowships TCS/22/02 and CSO EPD/22/13 (K.P.); the SPECIFICANCER Cancer Grand Challenge funded by CRUK and The Mark Foundation for Cancer Research (no. A29055) (O.J.S., P.J.P.); and the International Accelerator Award, ACRCelerate, jointly funded by CRUK (nos. A26825 and A28223), FC AECC (no. GEACC18004TAB) and AIRC (no. 22795) (A.D.C., P.D.D., O.J.S.).

## Author contributions

G.K. conceptualized the study, carried out the investigation, devised the methodology, wrote the original manuscript draft, and reviewed and edited the manuscript under the supervision of O.J.S. C.G. reviewed and edited the manuscript, and carried out the investigation (analyzed the ribosome profiling and proteomics data) with support from J.M., who performed the ribosome profiling. P.H. and J.A.W. carried out the investigation (performed the pause analysis), and reviewed and edited the manuscript, under the supervision of M.B. A.R. performed the whole-body *Apc* deletion experiments, and reviewed and edited the manuscript. N.V. performed the [35]S-methionine labeling experiments and supported the vivo and in vitro investigation along with C.A., P.P.C. and R.A.R. under the supervision of O.J.S. H.J. and S.B.M. analyzed the publicly available human cancer data, under the supervision of P.J.P. and P.D.D., respectively. N.N. and R.L.B. analyzed the IF-IHC experiments, with support from L.O.-J., under the supervision of J.L.Q. K.G. carried out the RNA-seq analysis. K.P. and J.E. sourced and curated the TMA, which was under the governance of the NHS Greater Glasgow and Clyde Biorepository. K.P. assisted with its analysis. H.H. analyzed publicly available human liver cancer data under the supervision of C.M. D.S. and S.B. generated the *Npm1* floxed mouse line. S.L. and S.Z. acquired the MS data. C.N., V.M. and the CRUK Scotland Institute histology services performed the histology processing and tissue staining experiments. J.R.P.K. and A.D.C. supported the investigation, and reviewed and edited the manuscript. M.B. reviewed and edited the manuscript, and acquired the funding. O.J.S. conceptualized the study, reviewed and edited the manuscript, and acquired the funding.

## Competing interests

The authors declare no competing interests.

## Additional information

**Extended data** is available for this paper at https://doi.org/10.1038/s41588-025-02408-7.

**Correspondence and requests for materials** should be addressed to Georgios Kanellos or Owen J. Sansom.

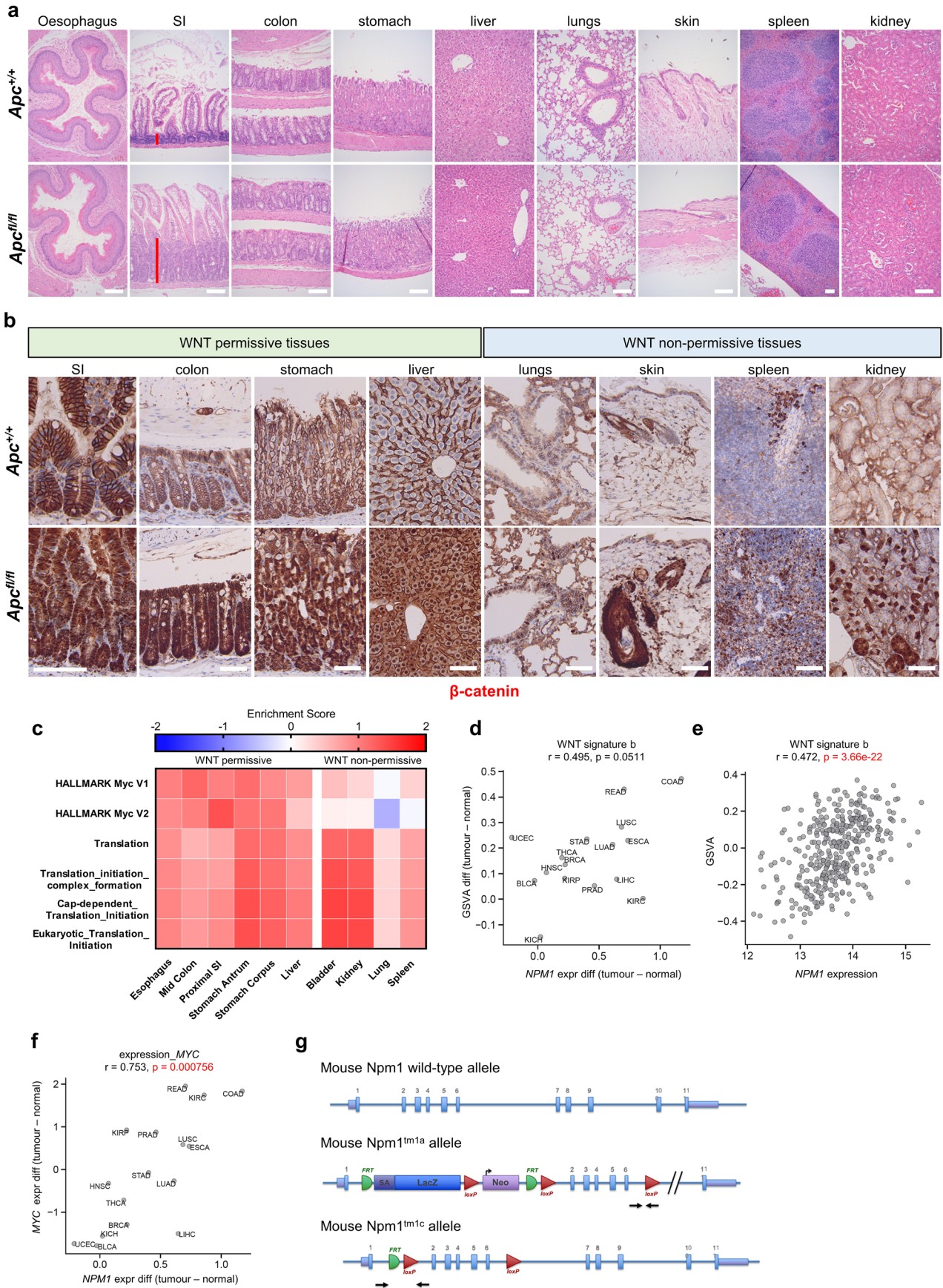

**Extended Data Fig. 1 | See next page for caption.**

**Extended Data Fig. 1 | *NPM1* expression correlates with WNT and MYC activation in cancer patients. a, b**, Representative micrographs of sections of the indicated tissues stained with H&E (**a**) and anti-BrdU (**b**) derived from *R26-Cre^{ER-T2}Apc^{+/+}* and *R26-Cre^{ER-T2}Apc^{fl/fl}* (n = 3 per group) mice harvested four days post-induction. Red bars indicate the expanded crypt depth in the small intestine (SI) following APC loss. Scale bars, 100 µm in (**a**) and 50 µm in (**b**). **c**, Heat map displaying gene set enrichment analysis in WNT permissive and non-permissive tissues for the indicated HALLMARK MYC and Reactome translation signatures in the *Apc^{fl/fl}* tissues compared to wild-type (see methods for more details on the signatures used). Enrichment level is based on each group average. **d**, Scatterplot showing the correlation between mean tumor-normal difference of GSVA score of the signature (WNT signature b) compared to mean tumor-normal difference of *NPM1* expression across different tumor types in TCGA dataset. **e**, Scatterplot showing the correlation between *NPM1* expression and the GSVA score of the signature (WNT signature b) among *NPM1* copy neutral COAD/READ tumours (n = 255) in the TCGA dataset. **f**, Scatterplot showing the correlation between mean tumor-normal difference of *MYC* compared to mean tumor-normal difference of *NPM1* expression across different tumor types in TCGA dataset. In (d-f) data were statistically assessed by two-sided Pearson correlation. Correlation coefficient (r) is displayed to indicate the degree of association. Statistically significant p-values are shown in red. **g**, Schematic of the floxing strategy used to generate a conditional *Npm1* knockout allele in mice. See methods for more information.

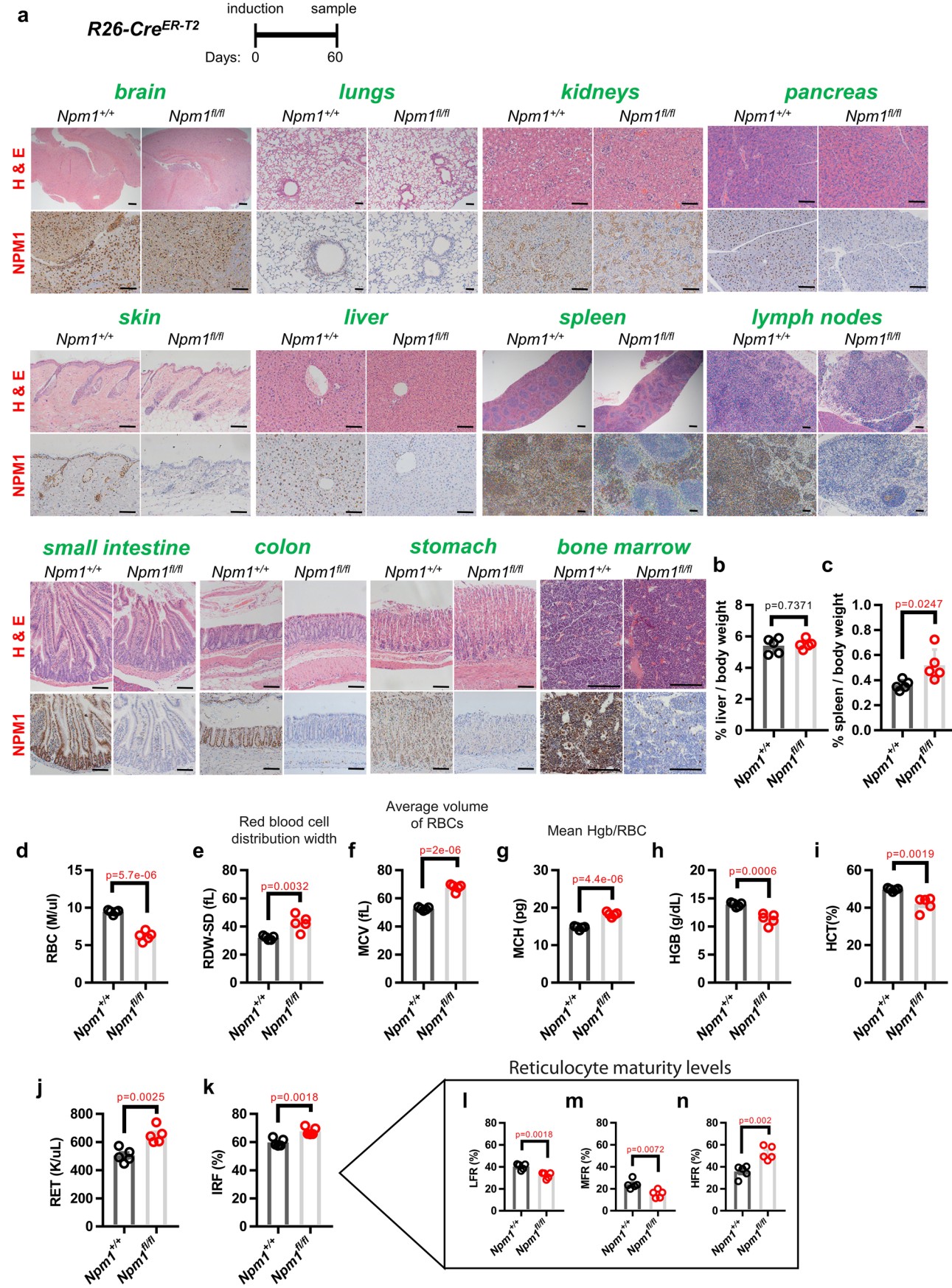

**Extended Data Fig. 2 | See next page for caption.**

**Extended Data Fig. 2 | Whole-body *Npm1* deletion is tolerated in adult mouse tissues. a**, Representative micrographs of sections of the indicated tissues stained with H&E and anti-NPM1 derived from animals harvested 60 days post-induction of *Npm1* deletion by CRE expression from the *Rosa26* locus (*R26-Cre*^*ER-T2*) (n = 5 per group). Scale bars, 100 μm for all apart from the brain and spleen H&E sections, which are 400 μm. **b, c**, Liver and spleen to body weight ratios from the groups presented in (**a**). **d-n**, Complete red blood cell (RBC) count (**d**), distribution width (RDW) of RBCs (**e**), RBC mean corpuscular volume (MCV) (**f**), RBC mean corpuscular haemoglobin (MCH) (**g**), haemoglobin (HGB) levels (**h**), haematocrit (HCT) value (**i**), reticulocyte count (**j**), and immature reticulocytes fraction (IRF) (**k**), with the reticulocyte maturity indices also shown inside the box (l-n) (LFR: low fluorescence ratio – least immature reticulocytes; MFR: medium fluorescence ratio; HFR: high fluorescence ratio – most immature reticulocytes), in circulating blood from the groups shown in (**a**) (n = 5 per group). Data, mean ± SEM. All data statistically assessed by unpaired two-tailed t tests. Statistically significant p-values are shown in red.

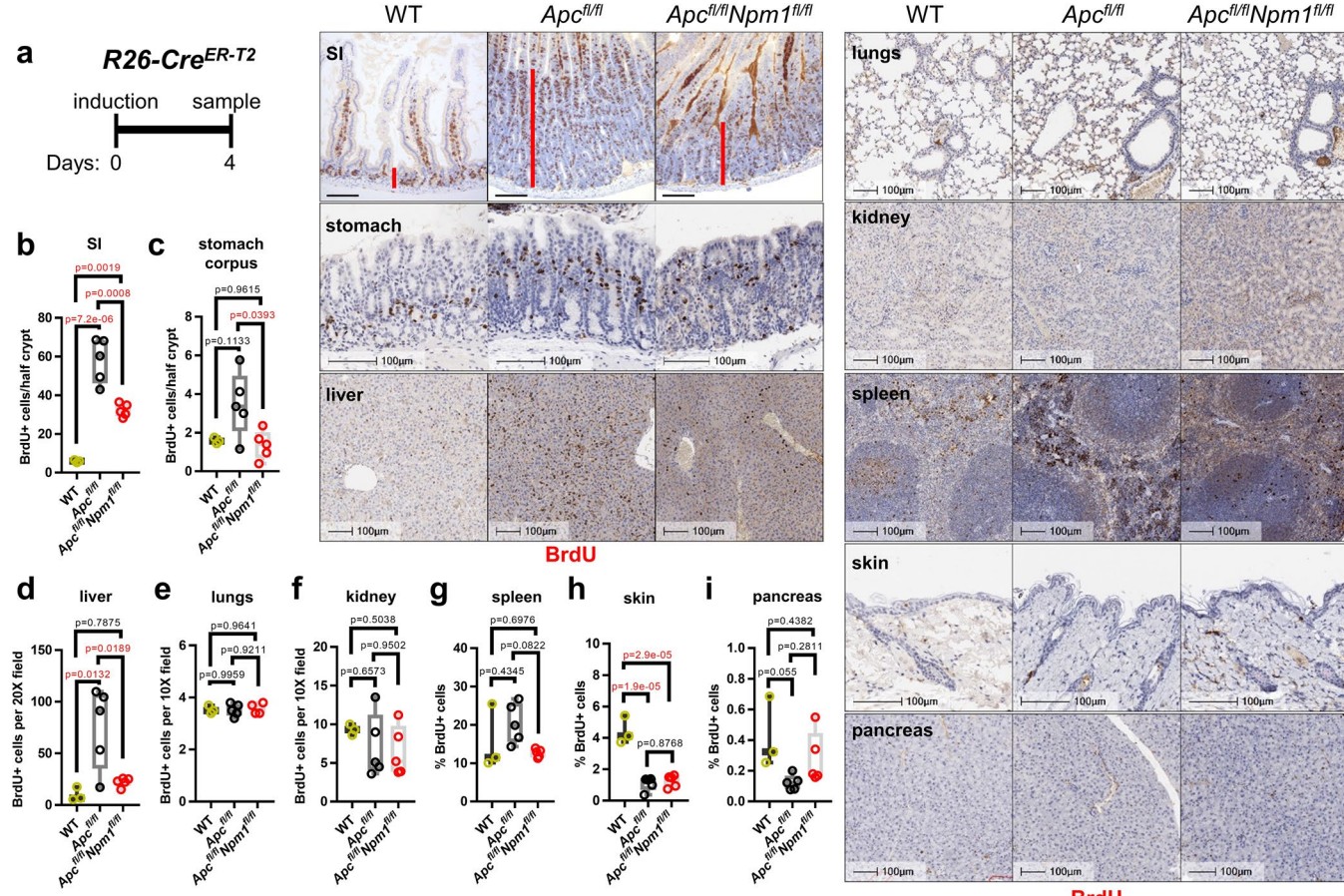

**Extended Data Fig. 3 | NPM1 is required for WNT-driven proliferation in WNT-permissive tissues. a,** Representative micrographs of indicated tissues from *R26-Cre^(ER-T2)* (WT, n = 3), *R26-Cre^(ER-T2)Apc^(fl/fl)* (n = 5) and *R26-Cre^(ER-T2)Apc^(fl/fl)Npm1^(fl/fl)* mice collected four days post-induction and stained for BrdU. Red bars indicate the expanded crypt depth following APC loss in the small intestine (SI). Scale bars, 100 μm. **b-i,** Quantification of BrdU+ cells for the tissues and sample sizes shown in (**a**). Boxes in boxplots extend from the 25th to 75th percentiles, whiskers extend to the minimum and maximum values, and the line in every box is plotted at the median. Data statistically assessed by one-way ANOVA followed by Tukey's multiple comparisons test. Statistically significant p-values are shown in red.

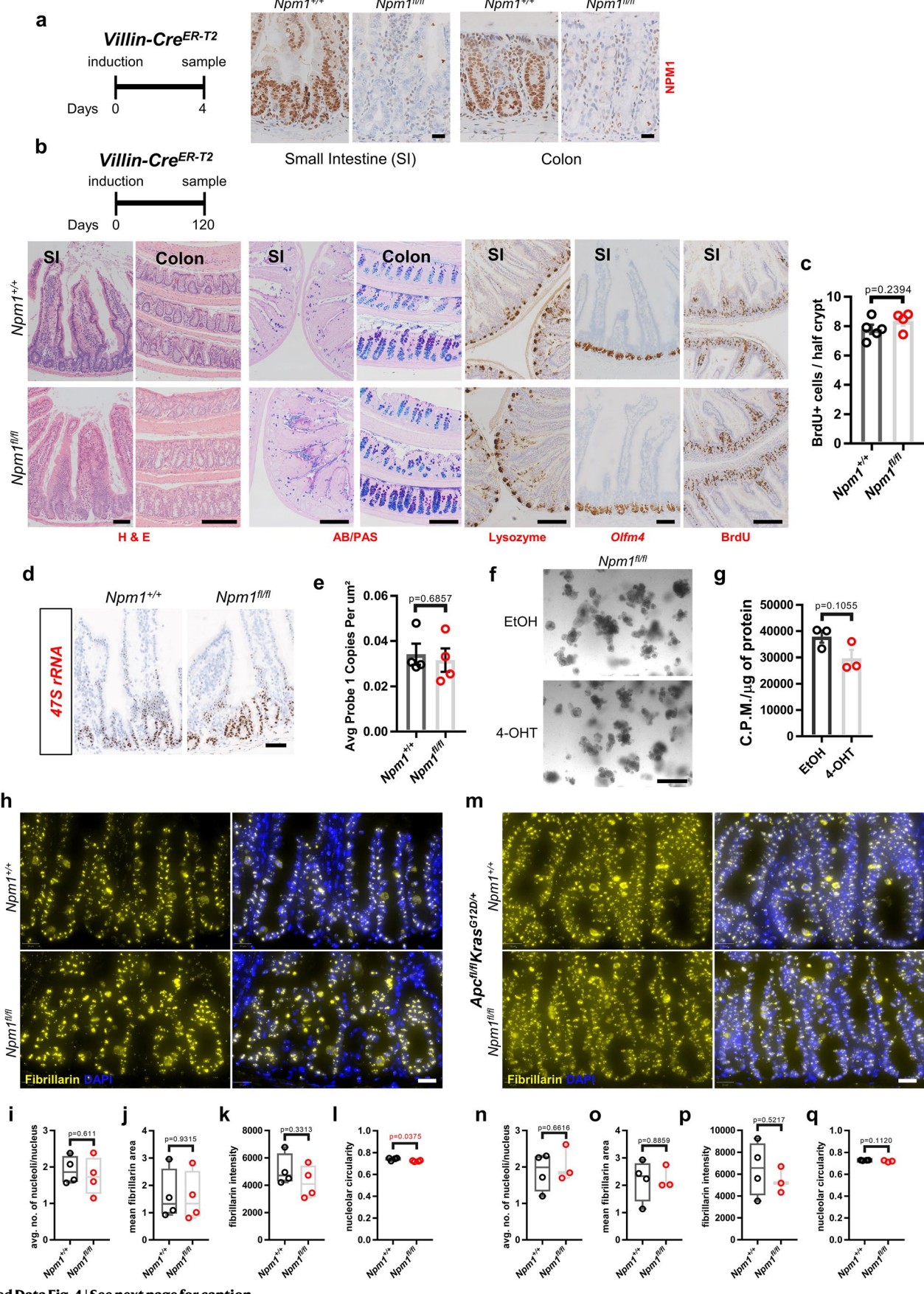

**Extended Data Fig. 4 | See next page for caption.**

**Extended Data Fig. 4 | NPM1 is dispensable for adult intestinal tissue homeostasis. a**, Representative images of NPM1-stained (SI) (left panels) and colon (right panels) sections from $Npm1^{+/+}$ (n = 4) and $Npm1^{fl/fl}$ (n = 5) animals collected four days post-induction. Scale bars, 50 μm. **b**, Representative staining with H&E, AB/PAS, anti-lysozyme, anti-BrdU and *in situ* hybridisation (ISH) for *Olfm4* of SI and colon of $Npm1^{+/+}$ (n = 5) and $Npm1^{fl/fl}$ (n = 5) animals collected 120 days post-induction. Scale bars, 100 μm. **c**, Quantification of BrdU+ cells in the SI of $Npm1^{+/+}$ (n = 5) and $Npm1^{fl/fl}$ (n = 4) animals collected four days post-induction. Data, mean ± SEM, statistically assessed by unpaired two-tailed t test. **d, e**, Representative micrographs of *47S rRNA* ISH in sections from intestinal tissue of $Npm1^{+/+}$ and $Npm1^{fl/fl}$ mice (n = 4 per group) sampled four days post-induction with HALO quantification in (e). Scale bar, 50 μm. Data, mean ± SEM, statistically assessed by unpaired two-tailed Mann Whitney test. **f**, Representative images of $Npm1^{fl/fl}$ SI organoids induced with ethanol (EtOH) or 4-OHT in culture, taken six days post-induction (n = 3 independent lines derived from individual mice). Scale bar, 100 μm. **g**, Protein synthesis rate quantification by $^{35}$S-methionine incorporation in the organoid lines presented in (f) (n = 3 per group). Data, mean ± SEM, statistically assessed by unpaired two-tailed t test. **h-q**, Representative fibrillarin (yellow) staining from SI tissue sections of $Npm1^{+/+}$ and $Npm1^{fl/fl}$ (n = 4 per group) animals collected four days post-induction, and $Apc^{fl/fl}Kras^{G12D/+}$ (n = 4) and $Apc^{fl/fl}Kras^{G12D/+}Npm1^{fl/fl}$ (n = 3) animals collected three days post-induction. Nuclei (blue) visualised with DAPI. Scale bars, 20 μm. Quantification of average number of nucleoli per nucleus based on fibrillarin stain, mean fibrillarin area, mean fibrillarin intensity and mean nucleolar circularity based on fibrillarin stain is shown. Data in (i-l) and (n-q) statistically assessed by unpaired two-tailed t tests. Boxes in boxplots extend from the 25th to 75th percentiles, whiskers extend to the minimum and maximum values, and the line in every box is plotted at the median. Statistically significant p-values are shown in red.

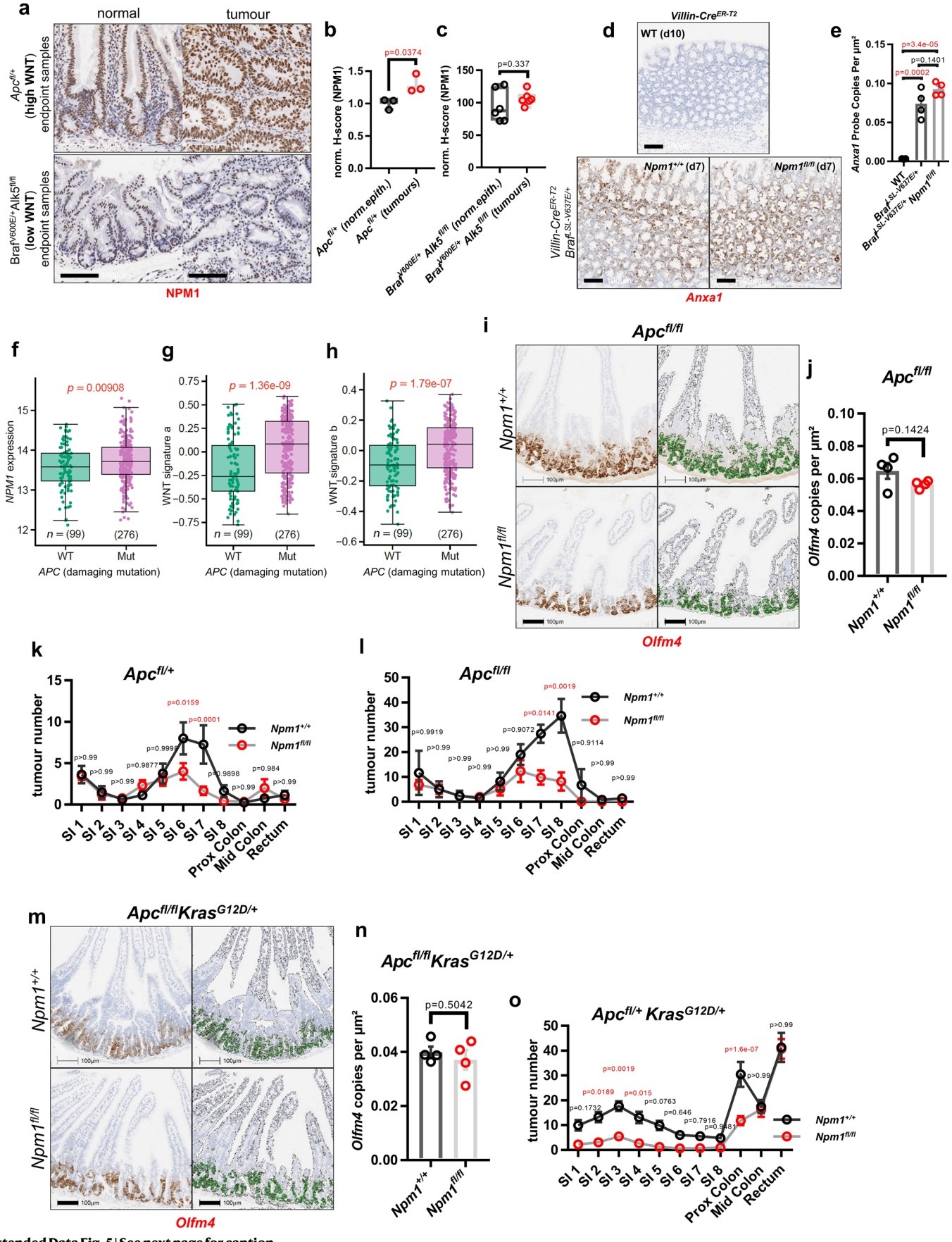

**Extended Data Fig. 5 | See next page for caption.**

**Extended Data Fig. 5 | NPM1 is upregulated and necessary for WNT-driven tumourigenesis. a**, Representative micrographs of NPM1-stained normal intestinal tissue sections and tumours from $Apc^{fl/+}$ (n = 3) (high WNT-driven tumours) and $Braf^{V600E/+}AlkS^{fl/fl}$ (n = 6) (low WNT-driven tumours) mice harvested at clinical endpoint. **b, c**, NPM1 staining HALO H-score quantification of the groups shown in (**a**), normalised to the mean normal epithelium value. **d, e**, *Anxa1* ISH in proximal colon tissue sections from *Villin-Cre*$^{ER-T2}$ (n = 3) sampled 10 days, *Villin-Cre*$^{ER-T2}Braf^{L-SL-V637E/+}$ (n = 4) and *Villin-Cre*$^{ER-T2}Braf^{L-SL-V637E/+}Npm1^{fl/fl}$ (n = 4) sampled 7 days post-induction, and HALO quantification shown in (**e**). Data statistically assessed by one-way ANOVA followed by Tukey's multiple comparisons test. **f**, *NPM1* expression in wild-type (WT) and mutant (Mut) *APC* COAD/READ tumours in the TCGA dataset (n = 375). **g, h**, GSVA score of WNT signatures a and b in WT and Mut *APC* TCGA COAD/READ tumours. **i**, *Olfm4* ISH and HALO rendering on intestinal tissue sections from $Apc^{fl/fl}$ mice with or without $Npm1^{fl/fl}$ and HALO quantification in (**j**) (n = 4 per group). **k**, Tumor number distribution within equidistant proximal to distal SI and colon sections from $Apc^{fl/+}$ (n = 8) and $Apc^{fl/+}Npm1^{fl/fl}$ (n = 9) mice sampled at clinical endpoint. Data statistically assessed by two-way ANOVA followed by Sidak's multiple comparisons test. **l**, Tumor number distribution within equidistant proximal to distal SI and colon sections from *Lgr5-Cre*$^{ER-T2}Apc^{fl/fl}$ (n = 11) and *Lgr5-Cre*$^{ER-T2}Apc^{fl/fl}Npm1^{fl/fl}$ (n = 9) mice sampled at clinical endpoint. Data statistically assessed by two-way ANOVA followed by Sidak's multiple comparisons test. **m**, *Olfm4* ISH and HALO rendering on intestinal tissue sections from $Apc^{fl/fl}Kras^{G12D/+}$ mice with or without $Npm1^{fl/fl}$ and HALO quantification in (**n**) (n = 4 per group). **o**, Tumor number distribution within equidistant proximal to distal SI and colon sections from $Apc^{fl/+}Kras^{G12D/+}$ (n = 17) and $Apc^{fl/+}Kras^{G12D/+}Npm1^{fl/fl}$ (n = 20) mice sampled at clinical endpoint. Data statistically assessed by two-way ANOVA followed by Sidak's multiple comparisons test. Bar chart data, mean ± SEM. Boxes in boxplots extend from the 25th to 75th percentiles, whiskers extend to the minimum and maximum values, and the line in every box is plotted at the median. All data in bar charts and boxplots statistically assessed by unpaired two-tailed t tests. Statistically significant p-values are shown in red. Scale bars, 100 μm.

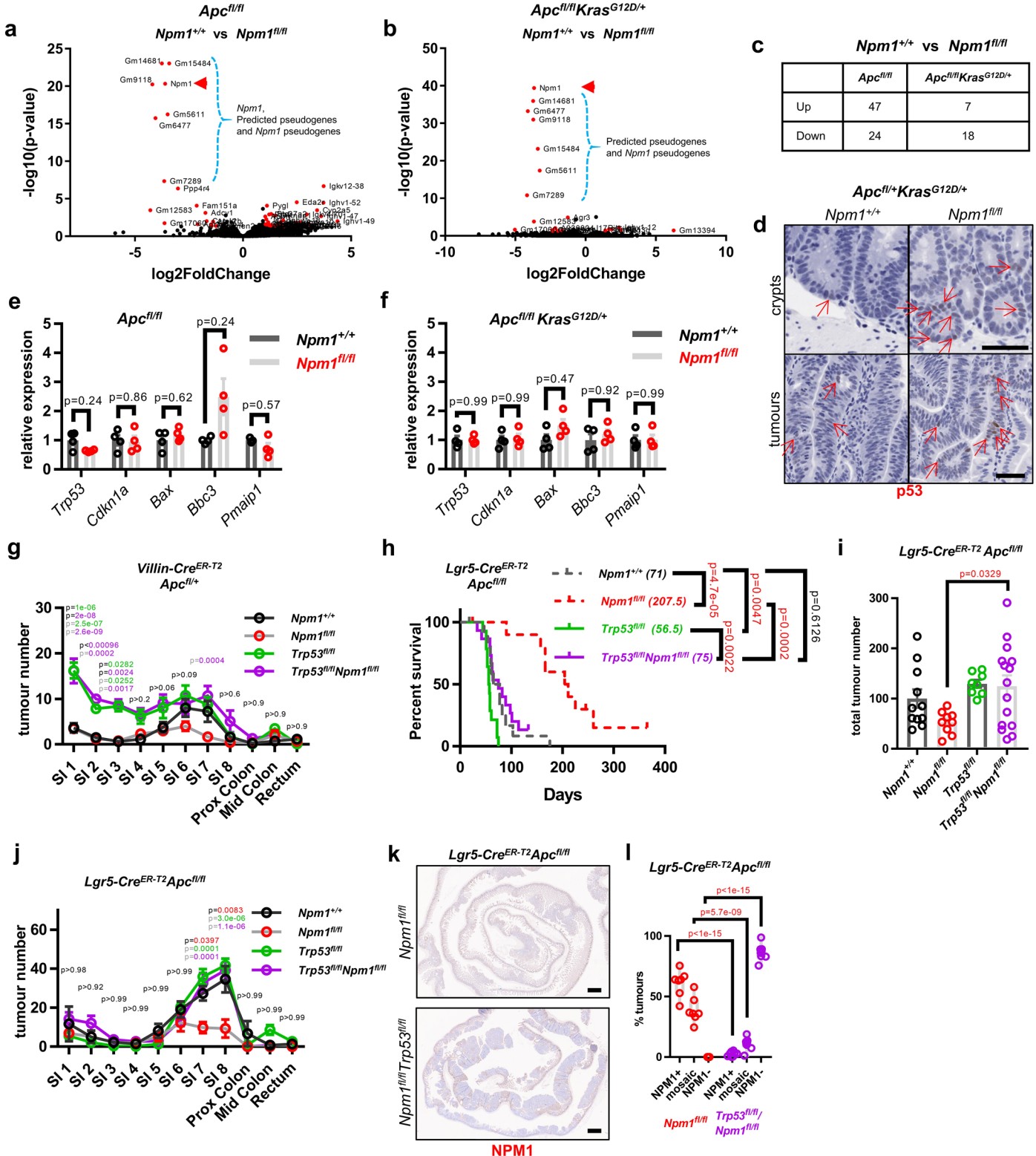

**Extended Data Fig. 6 | See next page for caption.**

**Extended Data Fig. 6 | Post-transcriptional activation of p53 mediates the survival benefit of NPM1 loss in WNT-driven CRC models. a-c**, Volcano plots of SI total RNA differential expression comparing *Apc^{fl/fl}Npm1^{fl/fl}* with *Apc^{fl/fl}* (**a**) and *Apc^{fl/fl}Kras^{G12D/+}Npm1^{fl/fl}* with *Apc^{fl/fl}Kras^{G12D/+}* (**b**) (n = 4/group). Significantly deregulated transcripts (adj p < 0.05, log2FC < -1 or >1) shown in red. DESeq2 with two-sided Wald test and Benjamini–Hochberg correction used. *Npm1* and *Npm1* pseudogenes highlighted. **c**, Number of deregulated transcripts. **d**, p53 staining in *Apc^{fl/+}Kras^{G12D/+}* and *Apc^{fl/+}Kras^{G12D/+}Npm1^{fl/fl}* (n = 4/group) SI crypts and tumours. Arrows indicate p53+ cells. Scale bar, 50 μm. Quantification in Fig. 3j,k. **e, f**, qPCR analysis of p53 target gene expression in *Apc^{fl/fl}* (**e**) and *Apc^{fl/fl}Kras^{G12D/+}* (**f**) mice ±*Npm1^{fl/fl}* (n = 4/group; *Pmaip1* in *Apc^{fl/fl}* n = 3). Multiple two-sided t-tests with Holm–Sidak correction. **g**, Tumor distribution within equidistant proximal to distal SI and colon sections from *Apc^{fl/+}* (n = 8), *Apc^{fl/+}Npm1^{fl/fl}* (n = 9) (also shown in Extended Data Fig. 5k), *Apc^{fl/+}Trp53^{fl/fl}* (n = 8) and *Apc^{fl/+}Trp53^{fl/fl}Npm1^{fl/fl}* (n = 14) endpoint mice. Two-way ANOVA with Sidak's test. Colour combinations denote groups compared. Black p-values indicate non-significance. **h**, Survival curves of *Lgr5-Cre^{ER-T2}Apc^{fl/fl}* (n = 13), *Lgr5-Cre^{ER-T2}Apc^{fl/fl}Npm1^{fl/fl}* (n = 11) (also shown in Fig. 2h and Extended Data Fig. 7a), *Lgr5-Cre^{ER-T2}Apc^{fl/fl}TrpS3^{fl/fl}* (n = 14) and

*Lgr5-Cre^{ER-T2}Apc^{fl/fl}TrpS3^{fl/fl}Npm1^{fl/fl}* (n = 15) endpoint mice. Median survival (days) indicated in brackets. P-values obtained by Log-rank (Mantel-Cox) tests. Censored mice denoted as tick marks. **i**, Endpoint tumor numbers from *Lgr5-Cre^{ER-T2}Apc^{fl/fl}* (n = 11), *Lgr5-Cre^{ER-T2}Apc^{fl/fl}Npm1^{fl/fl}* (n = 9) (also shown in Fig. 2i and the *Npm1^{fl/fl}* cohort in Extended Data Fig. 7b), *Lgr5-Cre^{ER-T2}Apc^{fl/fl}TrpS3^{fl/fl}* (n = 8) and *Lgr5-Cre^{ER-T2}Apc^{fl/fl}TrpS3^{fl/fl}Npm1^{fl/fl}* (n = 15) mice. One-way ANOVA with Tukey's test. **j**, Tumor distribution within equidistant proximal to distal SI and colon sections from groups in (i). Two-way ANOVA with Sidak's test. Colour combinations denote groups compared. Black p-values indicate non-significance. **k**, NPM1-stained SI from *Lgr5-Cre^{ER-T2}Apc^{fl/fl}Npm1^{fl/fl}* (n = 7) (also shown in Extended Data Fig. 7d) and *Lgr5-Cre^{ER-T2}Apc^{fl/fl}TrpS3^{fl/fl}Npm1^{fl/fl}* (n = 8) endpoint animals. Scale bar, 1 mm. **l**, Quantification of NPM1 + /−/mosaic tumours in endpoint *Lgr5-Cre^{ER-T2}Apc^{fl/fl}TrpS3^{fl/fl}Npm1^{fl/fl}* (n = 8) compared to *Lgr5-Cre^{ER-T2}Apc^{fl/fl}Npm1^{fl/fl}* (n = 7) (also presented in Fig. 2k. and Extended Data Fig. 7e) animals. Two-way ANOVA with Sidak's test. Data mean ± SEM (bar charts); boxplots boxes extend from 25^{th}-75th percentiles, whiskers from minimum to maximum values, and the line denotes the median. Significant p-values in red.

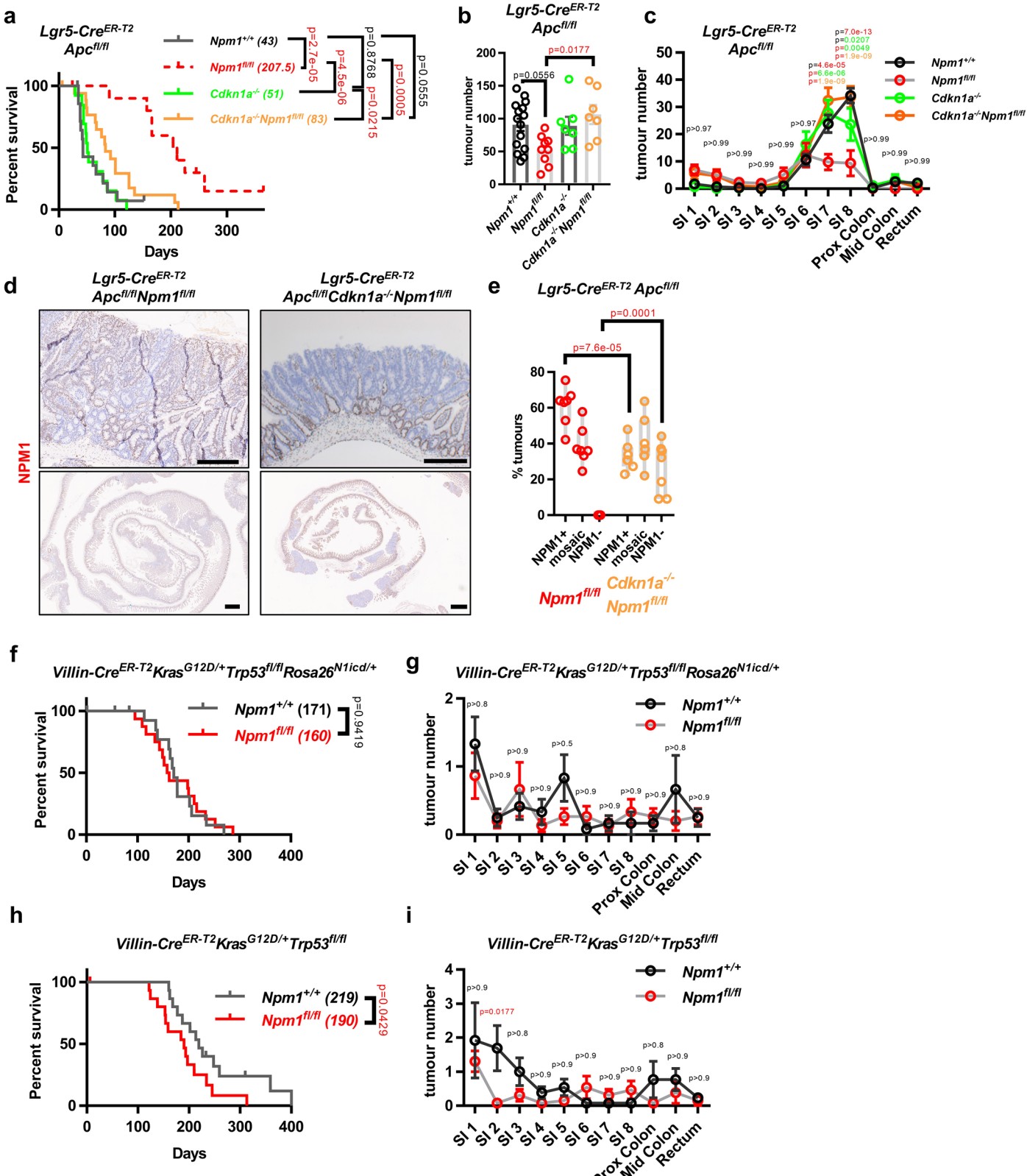

**Extended Data Fig. 7 | See next page for caption.**

**Extended Data Fig. 7 | *Cdkn1a* deletion abrogates the survival benefit of NPM1 loss in WNT-driven CRC models. a**, Survival curves of *Lgr5-Cre^ER-T2^Apc^fl/fl^* (n = 15), *Lgr5-Cre^ER-T2^Apc^fl/fl^Npm1^fl/fl^* (n = 11, also shown in Fig. 2h and Extended Data Fig. 6h), *Lgr5-Cre^ER-T2^Apc^fl/fl^Cdkn1a^−/−^* (n = 13) and *Lgr5-Cre^ER-T2^Apc^fl/fl^Cdkn1a^−/−^Npm1^fl/fl^* (n = 18) mice. Median survival (days) indicated in brackets. **b**, Tumor numbers from *Lgr5-Cre^ER-T2^Apc^fl/fl^* (n = 15), *Lgr5-Cre^ER-T2^Apc^fl/fl^Npm1^fl/fl^* (n = 9, also shown in Fig. 2i and Extended Data Fig. 6i), *Lgr5-Cre^ER-T2^Apc^fl/fl^Cdkn1a^−/−^* (n = 7) and *Lgr5-Cre^ER-T2^Apc^fl/fl^Cdkn1a^−/−^Npm1^fl/fl^* (n = 7) mice at clinical endpoint. Data, mean ± SEM. Data statistically assessed by one-way ANOVA followed by Tukey's multiple comparisons test. **c**, Tumor distribution within equidistant proximal to distal SI and colon sections from the groups shown in (b). Data statistically assessed by two-way ANOVA followed by Sidak's multiple comparisons test. Colour combinations denote groups compared. P-values in black indicate no statistical significance between groups. **d**, Representative staining for NPM1 on SI tissue sections from *Lgr5-Cre^ER-T2^Apc^fl/fl^Npm1^fl/fl^* (n = 7 – also shown in Fig. 2j and Extended Data Fig. 6k) and *Lgr5-Cre^ER-T2^Apc^fl/fl^Cdkn1a^−/−^Npm1^fl/fl^* (n = 7) animals at clinical endpoint. Scale bars, 100 µm (top panels) and 1 mm (bottom panels).

**e**, Quantification of percentage of tumours being positive, negative or mosaic for NPM1 expression in *Lgr5-Cre^ER-T2^Apc^fl/fl^Cdkn1a^−/−^Npm1^fl/fl^* animals at clinical endpoint (n = 7) compared to that of *Lgr5-Cre^ER-T2^Apc^fl/fl^Npm1^fl/fl^* (n = 7) (also shown in Fig. 2k and Extended Data Fig. 6l). Data statistically assessed by two-way ANOVA followed by Sidak's multiple comparisons test. Boxes in boxplots extend from the 25th to 75th percentiles, whiskers extend to the minimum and maximum values, and the line in every box is plotted at the median. **f**, Survival curves of *villin-Cre^ER-T2^Kras^G12D/+^Trp53^fl/fl^Rosa26^NIicd/+^* and *villin-Cre^ER-T2^Kras^G12D/+^Trp53^fl/fl^Rosa26^NIicd/+^Npm1^fl/fl^* (n = 16 per group). **g**, tumor number distribution within equidistant proximal to distal SI and colon sections from cohorts shown in (f) (*Npm1^+/+^*, n = 12; *Npm1^fl/fl^*, n = 15). **h**, Survival curves of *villin-Cre^ER-T2^Kras^G12D/+^Trp53^fl/fl^* (n = 15) and *villin-Cre^ER-T2^Kras^G12D/+^Trp53^fl/fl^Npm1^fl/fl^* (n = 16). **i**, tumor number distribution within equidistant proximal to distal SI and colon sections from cohorts shown in (h) (n = 13 in both groups). P-values in all survival data obtained by Log-rank (Mantel-Cox) tests. Censored mice in survival curves denoted as tick marks at indicated times post-induction. Statistically significant p-values are shown in red.

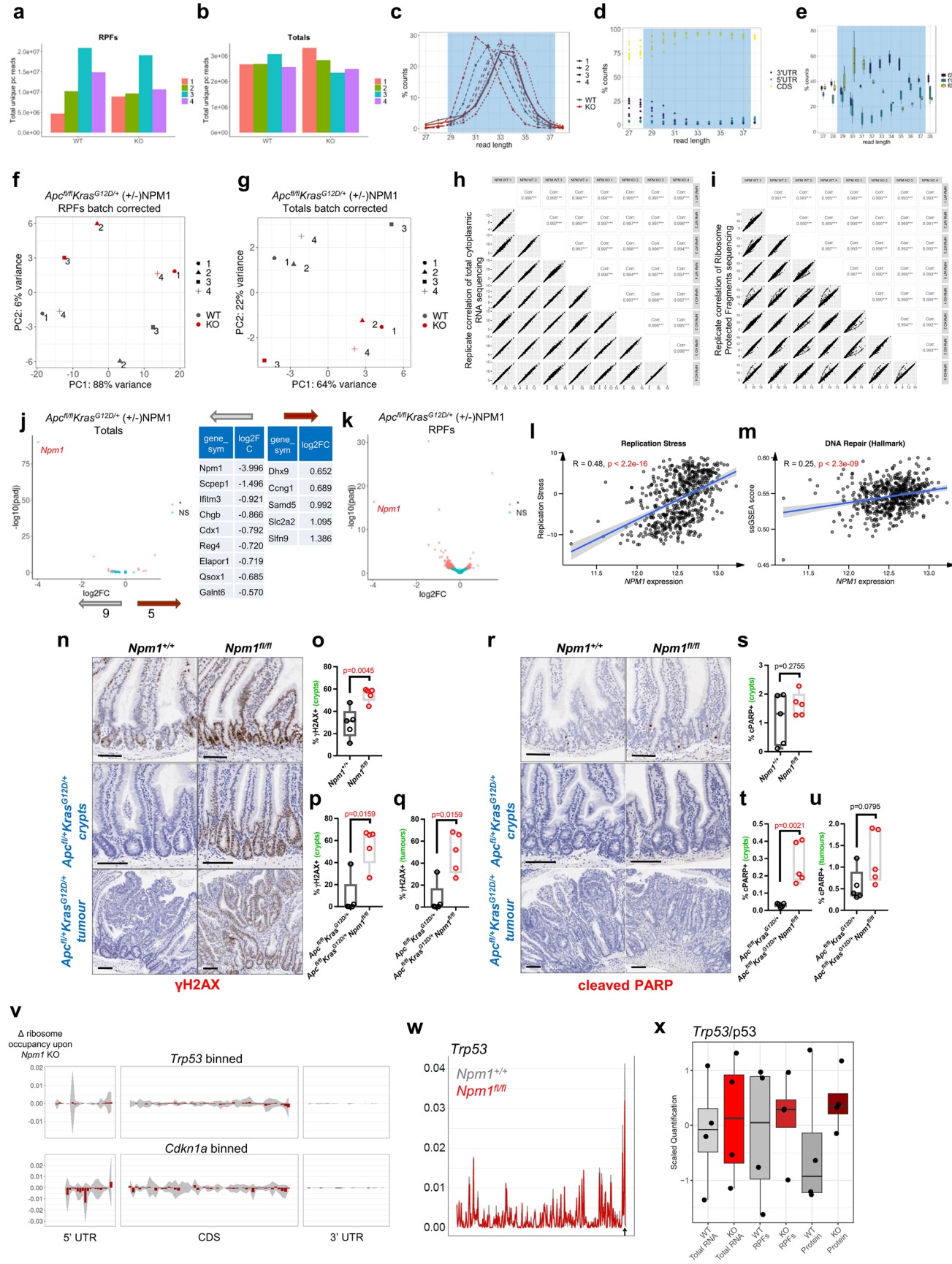

**Extended Data Fig. 8 | See next page for caption.**

**Extended Data Fig. 8 | NPM1 loss induces translational changes in *Apc*-deficient, *Kras*-mutant CRC model. a,b**, Protein-coding aligned (pc) unique reads for ribosome-protected fragments (RPFs) (a) and total cytoplasmic RNA (**b**) per replicate (n = 4/group; WT=$Apc^{fl/fl}Kras^{G12D/+}$, KO=$Apc^{fl/fl}Kras^{G12D/+}Npm1^{fl/fl}$ SI). All RNA reads used for downstream analyses. **c-e**, RPFs quality check: **c**, RPF read length distribution/replicate (pc unique reads/nucleotide length). **d**, Read coverage across transcript regions/read length: coding sequences (CDS-yellow), 5′-untranslated region (5′UTR-teal), 3′UTR (blue). >75% of reads align to CDSs. **e**, Read frame analysis/length. Lengths 27-28 lacked frame preference and were discarded, 29-31 had predominantly f0 and extracted with -12 offset, and 32-37 had f2 preference and extracted with -13 offset to correct codon position. Shading indicates lengths extracted for analysis. **f, g**, RPFs and total cytoplasmic RNA principal component analysis. Samples coloured by genotype and shaped by replicate. **h, i**, Correlation of rlog-normalised reads for total RNA, and RPFs across replicates. **j, k**, Volcano plots showing differentially expressed transcripts in total RNA (**j**) and RPFs (**k**) upon NPM1 loss. Significant transcripts indicated in red (adj. p-value < 0.1). Red/grey arrows and table

indicate transcripts increased/decreased by NPM1 loss at the RNA level. **l, m**, Scatterplots showing correlations between *NPM1* expression and replication stress or DNA repair status in human CRC tumours (n = 566, GSE39582 cohort). Two-sided Pearson correlation. Regression lines with 95% confidence intervals shaded and correlation coefficients (r) displayed. **n**, γH2AX staining on SI tissue from $Npm1^{+/+}$ and $Npm1^{fl/fl}$ mice sampled four days post-induction, and $Apc^{fl/+}Kras^{G12D/+}$ and $Apc^{fl/+}Kras^{G12D/+}Npm1^{fl/fl}$ endpoint mice (n = 5/group). **o-q**, γH2AX staining quantification in crypts and tumours of groups in (n). Unpaired two-tailed t test (o) and two-tailed Mann Whitney tests (p,q). **r-u**, Cleaved PARP staining and quantification as in (n-q). Unpaired two-tailed t tests. **v**, Metagene plots of *Trp53* and *Cdkn1a* ribosome occupancy across 5′UTR, CDS and 3′UTR. Shaded area indicates standard deviation across replicates (n = 4/group). **w**, Pause site plot for *Trp53* with resolved sites marked by arrow. **x**, Boxplots of p53 levels across total RNA, RPFs and proteomics. Boxplots boxes extend from 25th-75th percentiles, whiskers from minimum to maximum values, and the line denotes the median. Significant p-values in red. Scale bars, 100 µm.

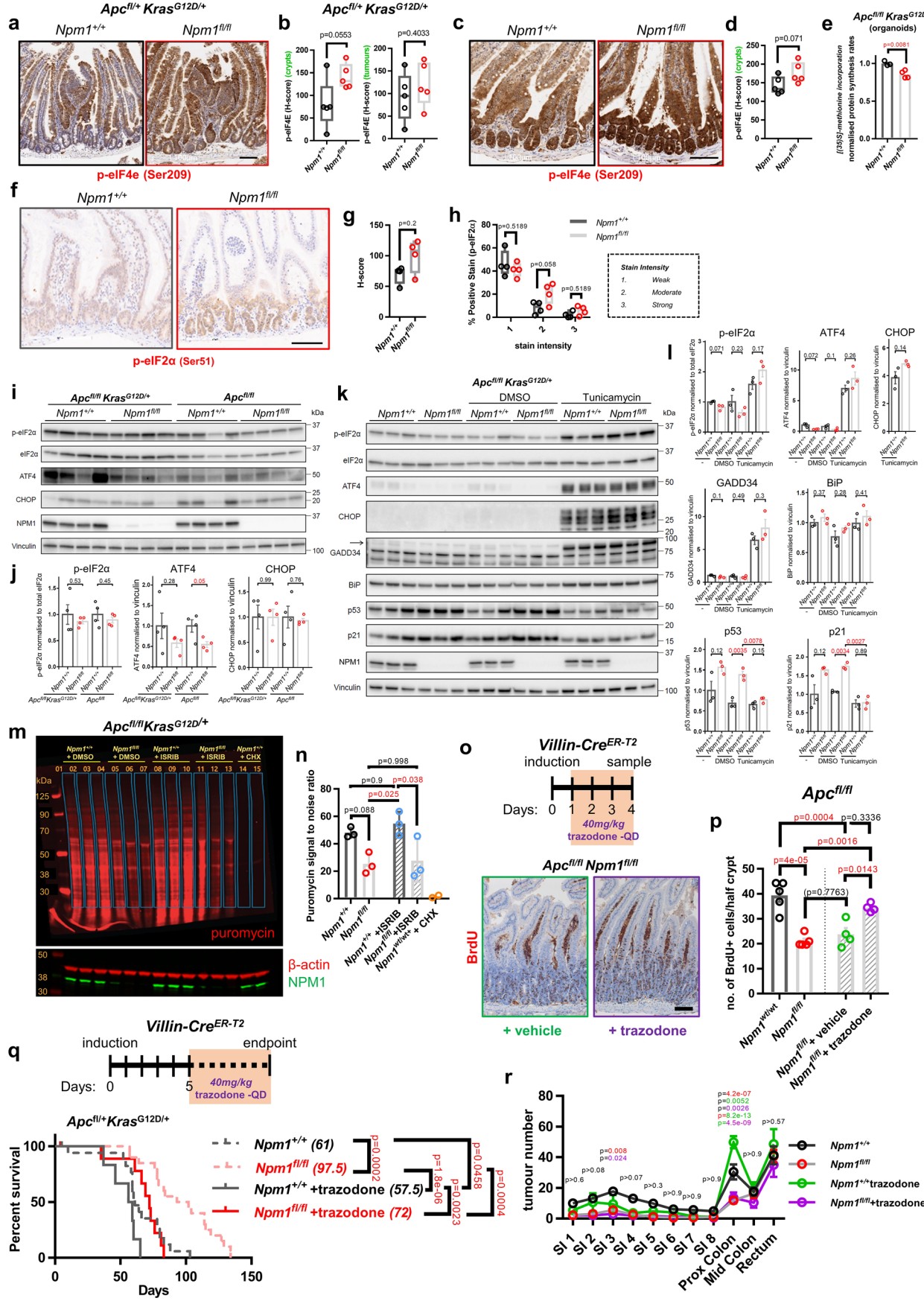

**Extended Data Fig. 9 | See next page for caption.**

**Extended Data Fig. 9 | NPM1 loss affects the translation pathway.**
**a, b**, p-eIF4e(Ser209) staining in SI from *Apc*^fl/+^*Kras*^G12D/+^ and *Apc*^fl/+^*Kras*^G12D/+^*Npm1*^fl/fl^ endpoint mice (n = 5/group), and quantification in crypts/tumours. Unpaired two-tailed t tests. **c, d**, p-eIF4e(Ser209) staining and quantification (crypts) in SI from *Npm1*^+/+^ and *Npm1*^fl/fl^ mice four days post-induction (n = 5/group). Unpaired two-tailed t test. **e**, ^35^S-methionine incorporation to assess protein synthesis in *Apc*^fl/fl^*Kras*^G12D/+^ (n = 3) and *Apc*^fl/fl^*Kras*^G12D/+^*Npm1*^fl/fl^ (n = 4) organoids. Data, mean ± SEM; unpaired two-tailed t test. **f-h**, p-eIF2α(Ser51) staining in SI from *Npm1*^+/+^ and *Npm1*^fl/fl^ mice four days post-induction (n = 4/group), and H-score/stain intensity quantification. Two-tailed Mann Whitney test (**g**); multiple two-sided t-tests with Holm–Sidak correction (**h**). **i, j**, Western blot for translation pathway components in epithelial extracts from *Apc*^fl/fl^, *Apc*^fl/fl^*Npm1*^fl/fl^, *Apc*^fl/fl^*Kras*^G12D/+^ and *Apc*^fl/fl^*Kras*^G12D/+^*Npm1*^fl/fl^ mice (n = 4/group), and quantification (**j**). Data, mean ± SEM. Unpaired two-tailed t tests. **k, l**, Western blot in *Apc*^fl/fl^*Kras*^G12D/+^ and *Apc*^fl/fl^*Kras*^G12D/+^*Npm1*^fl/fl^ untreated or DMSO/tunicamycin-treated organoids (n = 3/group), and quantification (**l**). Data, mean ± SEM. Unpaired two-tailed t tests. **m, n**, Western blot for puromycin incorporation in *Apc*^fl/fl^*Kras*^G12D/+^ and *Apc*^fl/fl^*Kras*^G12D/+^*Npm1*^fl/fl^ ± DMSO/ISRIB-treated organoids (n = 3/group), and quantification of boxed areas (**n**). Cyclohexamide used as negative control. Data, mean ± SEM. One-way ANOVA with Tukey's test. **o**, BrdU staining in SI from *Apc*^fl/fl^*Npm1*^fl/fl^ mice ±vehicle/trazodone treatment (n = 4/group). **p**, BrdU+ cell quantification in SI half-crypts from groups in (o), compared to untreated *Apc*^fl/fl^*Npm1*^+/+^ (n = 5) and *Apc*^fl/fl^*Npm1*^fl/fl^ (n = 5) mice (also presented in Fig. 2c). Data, mean ± SEM. One-way ANOVA with Tukey's test. **q**, Survival curves of untreated *Apc*^fl/+^*Kras*^G12D/+^ (n = 17) and *Apc*^fl/+^*Kras*^G12D/+^*Npm1*^fl/fl^ (n = 20) (also shown in Fig. 2n), and trazodone-treated mice (*Apc*^fl/+^*Kras*^G12D/+^ n = 7, *Apc*^fl/+^*Kras*^G12D/+^*Npm1*^fl/fl^ n = 10). Median survival (days) indicated in brackets. Log-rank (Mantel-Cox) tests. Censored mice denoted as tick marks. **r**, tumor distribution within equidistant proximal to distal SI and colon sections from untreated *Apc*^fl/+^*Kras*^G12D/+^ (n = 17) and *Apc*^fl/+^*Kras*^G12D/+^*Npm1*^fl/fl^ (n = 20) (also shown in Extended Data Fig. 5o) and trazodone-treated *Apc*^fl/+^*Kras*^G12D/+^ (n = 5) and *Apc*^fl/+^*Kras*^G12D/+^*Npm1*^fl/fl^ (n = 8) endpoint mice. Two-way ANOVA with Sidak's test. Colour combinations denote groups compared. P-values in black indicate no significance. Other significant p-values in red. Boxplots boxes extend from 25^th^-75^th^ percentiles, whiskers from minimum to maximum values, and the line denotes the median. Scale bars (**a,c,f**), 100 µm; (**o**), 50 µm.

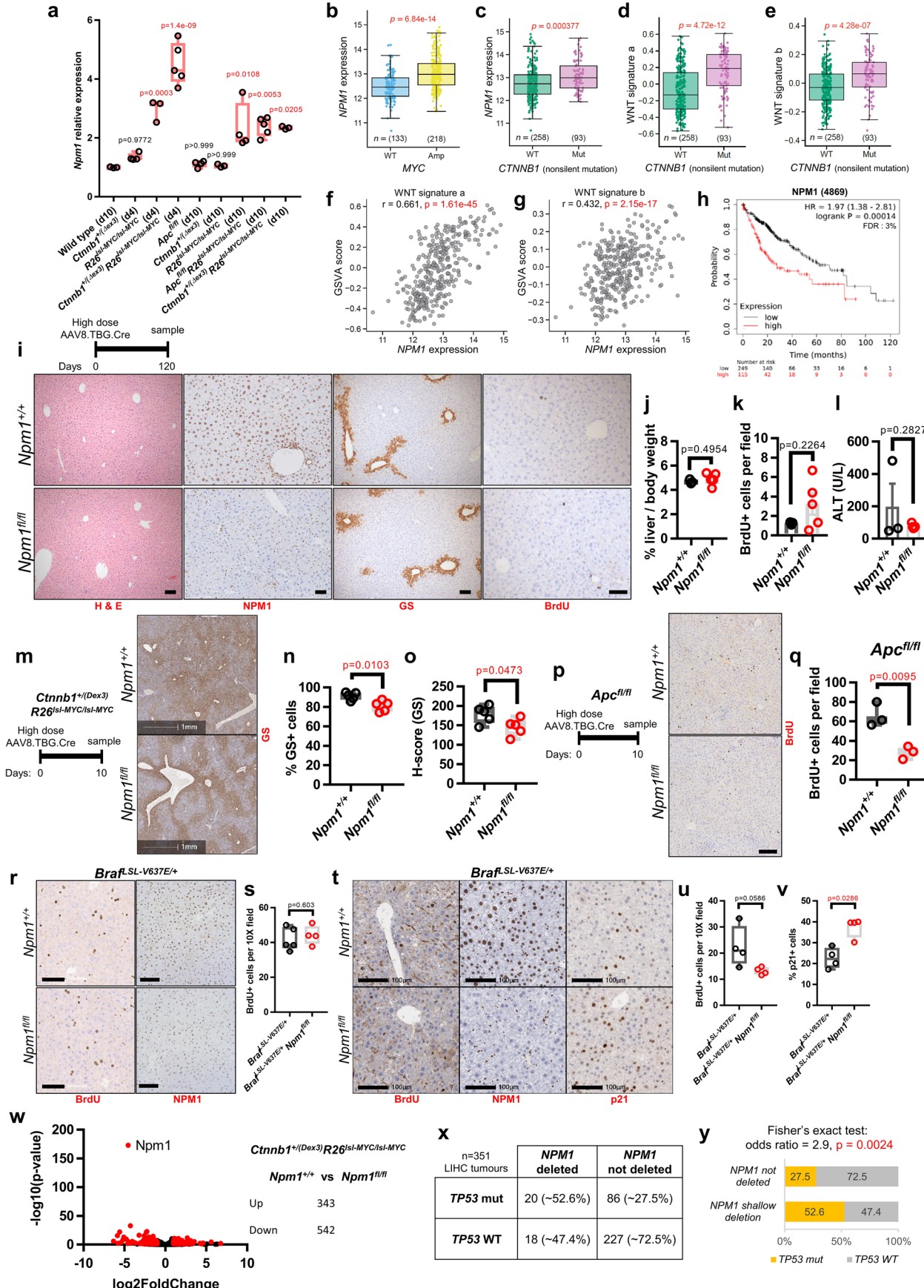

**Extended Data Fig. 10 | See next page for caption.**

**Extended Data Fig. 10 | *NPM1* is overexpressed in HCC and required for WNT-driven hepatocyte proliferation. a,** *Npm1* expression in murine livers of indicated genotypes and timepoints normalised to wild-type (GEO data: GSE230137) (wild-type(d10), *R26*^(lsl-MYC/lsl-MYC)(d4), *Ctnnb1*^(+/(Dex3))(d10), *Ctnnb1*^(+/(Dex3)) *R26*^(lsl-MYC/lsl-MYC)(d10), n = 3; *Ctnnb1*^(+/(Dex3))(d4), *Apc*^(fl/fl)(d10), *R26*^(lsl-MYC/lsl-MYC)(d10), n = 4; *Ctnnb1*^(+/(Dex3))*R26*^(lsl-MYC/lsl-MYC)(d4), *Apc*^(fl/fl)*R26*^(lsl-MYC/lsl-MYC)(d10), n = 5). One-way ANOVA with Tukey's test. **b, c,** *NPM1* expression in TCGA-LIHC tumours stratified by *MYC* amplification or *CTNNB1* mutation status. **d, e,** WNT signatures GSVA scores in wild-type and mutant *CTTNB1* TCGA-LIHC tumours. **f, g,** *NPM1* expression and WNT signatures GSVA scores correlation scatterplots in TCGA-LIHC tumours. Two-sided Pearson correlation. **h,** HCC patient survival with high/low *NPM1* (n = 249 low, n = 115 high). Log-rank test; tick marks: censored subjects. **i,** H&E, NPM1, glutamine synthetase (GS) and BrdU staining in *Npm1*^(+/+) (n = 3) and *Npm1*^(fl/fl) (n = 5) livers 120 days post high dose AAV8.TBG.Cre. Scale bars, 50 µm. **j-l,** Liver-to-body weight ratios, BrdU+ cells (9-13 10X fields of view (FOV)) and blood alanine transaminase (ALT) levels from groups in (**i**). **m-o,** GS staining, GS+ cells and H-score quantification in *Ctnnb1*^(+/(Dex3))*R26*^(lsl-MYC/lsl-MYC)±*Npm1*^(fl/fl) livers

ten days post high dose AAV8.TBG.Cre (n = 5/group). Scale bars, 1 mm. **p, q,** BrdU staining and quantification (six 10X FOV) in *Apc*^(fl/fl) and *Apc*^(fl/fl)*Npm1*^(fl/fl) livers ten days post high dose AAV8.TBG.Cre (n = 3/group). Scale bars, 100 µm. **r, s,** BrdU and NPM1 staining, and BrdU+ cell quantification (ten 10X FOV) in *Braf*^(t.SL-V637E/+) (n = 5) and *Braf*^(t.SL-V637E/+)*Npm1*^(fl/fl) (n = 4) livers four days post high dose AAV8.TBG. Cre. Scale bars, 100 µm. **t,** BrdU, NPM1 and p21 staining in *Braf*^(t.SL-V637E/+) and *Braf*^(t.SL-V637E/+)*Npm1*^(fl/fl) livers (n = 4/group) eight days post high dose AAV8.TBG.Cre. Scale bars, 100 µm. **u,** BrdU+ (ten 10X FOV) and **v,** p21+ hepatocytes from groups in (**t**). Unpaired two-tailed Mann Whitney test. **w,** RNAseq comparing *Ctnnb1*^(+/(Dex3)) *R26*^(lsl-MYC/lsl-MYC)±*Npm1*^(fl/fl) livers ten days post-induction (n = 4/group). Two-sided Wald test, Benjamini–Hochberg correction. Significant (adj p.value < 0.05, log2FC < -1 or >1) transcripts in red. **x, y,** Shallow *NPM1* deletion frequency and/or *TP53* mutations in TCGA-LIHC (n = 351). Two-sided Fisher's exact test. Data (bar charts), mean ± SEM. Boxplots boxes extend from 25th-75th percentiles, whiskers from minimum to maximum values; the line denotes the median. Statistics in (**b-e**), (**j-l**), (**n-o**), (**q**), (**s**), (**u**): unpaired two-tailed t tests. Significant p-values in red.

Owen Sansom

# Reporting Summary

## Statistics

For all statistical analyses, confirm that the following items are present in the figure legend, table legend, main text, or Methods section.

| n/a | Confirmed | |
|---|---|---|
| ☐ | ☒ | The exact sample size (*n*) for each experimental group/condition, given as a discrete number and unit of measurement |
| ☐ | ☒ | A statement on whether measurements were taken from distinct samples or whether the same sample was measured repeatedly |
| ☐ | ☒ | The statistical test(s) used AND whether they are one- or two-sided<br>*Only common tests should be described solely by name; describe more complex techniques in the Methods section.* |
| ☒ | ☐ | A description of all covariates tested |
| ☐ | ☒ | A description of any assumptions or corrections, such as tests of normality and adjustment for multiple comparisons |
| ☐ | ☒ | A full description of the statistical parameters including central tendency (e.g. means) or other basic estimates (e.g. regression coefficient) AND variation (e.g. standard deviation) or associated estimates of uncertainty (e.g. confidence intervals) |
| ☐ | ☒ | For null hypothesis testing, the test statistic (e.g. *F*, *t*, *r*) with confidence intervals, effect sizes, degrees of freedom and *P* value noted<br>*Give P values as exact values whenever suitable.* |
| ☒ | ☐ | For Bayesian analysis, information on the choice of priors and Markov chain Monte Carlo settings |
| ☒ | ☐ | For hierarchical and complex designs, identification of the appropriate level for tests and full reporting of outcomes |
| ☐ | ☒ | Estimates of effect sizes (e.g. Cohen's *d*, Pearson's *r*), indicating how they were calculated |

*Our web collection on statistics for biologists contains articles on many of the points above.*

## Software and code

Policy information about availability of computer code

| | |
|---|---|
| Data collection | ZEN 2 (blue edition, v2.0.0.0) (Carl Zeiss Microscopy)<br>Wallac MicroBeta TriLux 1450 scintillation counter with MicroBeta Workstation software (Version 4.0) (PerkinElmer)<br>Evident VS200 (version ASW 4.1.1) for fluorescence slide scanning<br>LI-COR - Odyssey CLx with Image Studio v6.0 (1.0.22) Image Studio (LICORbio)<br>BIO-RAD ChemiDoc Imaging System (Image Lab Software version 2.3.0.07)<br>Xcalibur software (Thermo Scientific, version 4.1.31.9)<br>Bio-Rad C1000 Touch CFX96 real time PCR system with CFX Maestro 2.3 Software (5.3.022.1030) |
| Data analysis | Microsoft Excel 2016 (16.0.5487.1000)<br>Prism version 7.0.4 (GraphPad)<br>HALO Image Analysis Platform version 3.6.4134 (Indica Labs, Inc.)<br>ImageJ (version 1.53q)<br>R (versions 4.2.2, 4.3.2 and 4.3.3)<br>RStudio (version 2023.12.1 Build 402) (Posit)<br>Image Studio v6.0 (1.0.22) (LICORbio)<br>Empiria Studio Software v3.0.0.173 (LICORbio)<br>Aperio ImageScope Version 12.4.3.5008 (Leica)<br>SPSS version 28.0.0 (IBM)<br>Kaplan-Meier Plotter (online tool)<br>Survminer package in RStudio version 3.0.3<br>Counts from TCGA data were normalised using the VST function from DESeq2 (version 1.36) |

ggplot2 (v.3.5.1)
ggpubr (v.0.6.0)
ggbeeswarm (v.0.7.2)
FastQC version 0.11.8
Trim Galore version 0.6.4
HISAT2 version 2.1.0
FeatureCounts version 1.6.4
DESeq2 version 1.22.2
ReactomePA version 1.36.0.
Visiopharm version 2024.07.2.17212 x64
QuPath Version: 0.5.1
gsva function from the Python package GSEApy (v1.1.8)
For copy number (CN) status, we used the absolute CN values calculated by GISTIC2 (v2.0.23)
WGCNA (v.1.70-3) R package
PDSclassifier (v.1.0.0) R package
ProliferativeIndex (v.1.0.1) R package
The gene signature DNA repair (Hallmark) was accessed via msigdbr (v.7.4.1) R package to obtain single sample gene set enrichment analysis (ssGSEA) score using GSVA (v.1.42.0) R package.
MS Raw data were processed with MaxQuant software version 1.6.14.0 and searched with Andromeda search engine.
MaxQuant output was used for protein quantitation analysis using Perseus software version 1.6.15.0.
Riboseq analysis was performed following the publicly available pipeline from the Bushell lab (https://github.com/Bushell-lab/Ribo-seq). Shell scripts analyses were run on a terminal in Ubuntu 20.04.6 LTS; R scripts were run in RStudio 2023.12.0, Build 369 using R versions 4.2.2, 4.3.2 (2023-10-31), and 4.3.3. The code used specifically in this project for Riboseq analysis and integration with the proteomics data is available on the project-specific GitHub page (https://github.com/ChiaraGiacomelli/NPM_CRC).

For manuscripts utilizing custom algorithms or software that are central to the research but not yet described in published literature, software must be made available to editors and reviewers. We strongly encourage code deposition in a community repository (e.g. GitHub). See the Nature Portfolio guidelines for submitting code & software for further information.

# Data

Policy information about availability of data

All manuscripts must include a data availability statement. This statement should provide the following information, where applicable:
- Accession codes, unique identifiers, or web links for publicly available datasets
- A description of any restrictions on data availability
- For clinical datasets or third party data, please ensure that the statement adheres to our policy

RNA sequencing data generated in this study have been deposited in the Gene Expression Omnibus (GEO) database under the accession numbers GSE230110, GSE309379 and GSE250047. Ribosome profiling data have been deposited at GEO under the accession number GSE249958. Proteomics data have been deposited to the ProteomeXchange Consortium via the PRIDE partner repository with the dataset identifier PXD062969. Murine liver Npm1 expression data presented in Extended Data Fig. 10a were derived from RNAseq data available at GEO under accession number GSE230137. Human cancer analyses were conducted using data that are in whole or part based upon data generated by the TCGA Research Network (https://www.cancer.gov/tcga), as well as the publically available colon cancer microarray dataset GSE39582. Source data are available with this manuscript.

# Research involving human participants, their data, or biological material

Policy information about studies with human participants or human data. See also policy information about sex, gender (identity/presentation), and sexual orientation and race, ethnicity and racism.

| | |
|---|---|
| Reporting on sex and gender | CRC PATIENT COHORT ASSESSED FOR NPM1 EXPRESSION LEVELS<br>Sex was assigned based on medical records and was not considered in the present study. |
| Reporting on race, ethnicity, or other socially relevant groupings | No socially constructed or socially relevant categorization variables were used in this study. |
| Population characteristics | NPM1 protein expression was assessed in a retrospectively collected cohort of stage I-III CRC patients (n=787). The cohort consisted of CRC patients undergoing surgical resection with curative intent within Greater Glasgow and Clyde National Health Service between 1997 and 2013. Data are deposited within the Glasgow Safehaven (GSH21ON009).<br>The age of the patients included in this cohort ranged from 21-98 years with a mean of 68.79 years. The cohort consisted of 45%/55% female/male patients. |
| Recruitment | This was a retrospectively collected cohort. Patients were excluded from the study due to mortality within 30 days of surgery, administration of neoadjuvant therapy and/or emergency presentation. |
| Ethics oversight | Ethical approval was in place for the study (MREC/01/0/3). |

Note that full information on the approval of the study protocol must also be provided in the manuscript.

# Field-specific reporting

Please select the one below that is the best fit for your research. If you are not sure, read the appropriate sections before making your selection.

☒ Life sciences ☐ Behavioural & social sciences ☐ Ecological, evolutionary & environmental sciences

For a reference copy of the document with all sections, see nature.com/documents/nr-reporting-summary-flat.pdf

# Life sciences study design

All studies must disclose on these points even when the disclosure is negative.

| | |
|---|---|
| Sample size | No formal power analyses were carried out for determining cohort sizes in the present study. All in vivo cohort sizes were determined based on power analyses in studies previously carried out in our lab in similar GEM models, respecting the limited use of animals in line with the 3R system: Replacement, Reduction, Refinement.<br>The human colorectal cancer cohort analysed for NPM1 expression as part of this study was based on historic patient samples and no prospective sample collection carried out as part of this study.<br>For in vitro experiments the minimum number of biological replicates required was used that permitted meaningful statistical comparisons. |
| Data exclusions | Patients represented on the human CRC TMA were excluded due to mortality within 30 days of surgery and/or administration of neoadjuvant therapy.<br>As part of QC in the Riboseq analysis, any transcript with average reads across all samples lower than 10 was excluded from any downstream analysis.<br>No other data were excluded from the analyses in the present study. |
| Replication | All in vivo experiments are performed in individual, biologically unique, animals. As such, all replicate values in all experiments presented correspond to independent replicate animals that confirm reproducibility of the effect of genetic manipulation and/or drug treatments on these animals. All numbers of individual biological replicates are stated for every experiment and denoted with 'n'.<br><br>35S methionine incorporation in organoids in vitro was performed in 3 independent organoid lines, derived from 3 independent animals (3 technical replicates each) |
| Randomization | To minimise genetic variability, all experimental and control animals were generated on inbred genetic backgrounds. Where possible, control and experimental animals were co-housed independent of genotype. All aging experiments included animals from both sexes. Animals were recruited to treatment groups in a partially randomised manner while taking these factors into account.<br><br>In vitro organoid experiments were performed on a single defined genetic background. As such, no randomization was performed. |
| Blinding | Researchers were not blinded to genotype during study and data collection. The viability of NPM1 knockout in adult organisms had not been previously assessed and, as such, the knockout animals needed close monitoring. In addition, the preclinical cancer models used require enhanced inspection to ensure animal welfare and compliance with Home Office regulations. The researchers were blinded to genotype and/or drug treatment during data analyses. |

# Reporting for specific materials, systems and methods

We require information from authors about some types of materials, experimental systems and methods used in many studies. Here, indicate whether each material, system or method listed is relevant to your study. If you are not sure if a list item applies to your research, read the appropriate section before selecting a response.

## Materials & experimental systems

| n/a | Involved in the study |
|---|---|
| ☐ | ☒ Antibodies |
| ☐ | ☒ Eukaryotic cell lines |
| ☒ | ☐ Palaeontology and archaeology |
| ☐ | ☒ Animals and other organisms |
| ☒ | ☐ Clinical data |
| ☒ | ☐ Dual use research of concern |
| ☒ | ☐ Plants |

## Methods

| n/a | Involved in the study |
|---|---|
| ☒ | ☐ ChIP-seq |
| ☒ | ☐ Flow cytometry |
| ☒ | ☐ MRI-based neuroimaging |

# Antibodies

| | |
|---|---|
| Antibodies used | Antibody - Company - Code - (Lot Number) - Dilution<br>Brdu - BD - Biosciences - 347580 - (3016583) - 1:250 (IHC)<br>Glutamine Synthetase - Sigma-aldrich - HPA007316 - (000016878) - 1:800 (IHC)<br>Lysozyme - Agilent - A099 - (41480045) - 1:200 (IHC)<br>NPM1 - Cell Signaling - 3542 - (5) - 1:50 (IHC); 1:1000 (WB) |

p21 - Abcam - ab107099 - (1029985-2) - 1:250 (IHC)
p53 (clone CM5) - Leica Biosystems - NCL-L-p53CM5p - (6087005) - 1:750 (IHC); 1:500 (IF-IHC)
cleaved PARP [E51] - Abcam - ab32064 - (1091190-1) - 1:1000 (IHC)
Phospho-Histone H2A.X (Ser139) (20E3) - CST - 9718 - (21) - 1:120 (IHC)
phospho-4E-BP1 - Cell Signaling - 2855 - (26) - 1:250 (IHC)
phospho-eEF2 - Cell Signaling - 2331 - (9) - 1:100 (IHC)
phospho-eIF2α - Cell Signaling - 3398 - (8) - 1:50 (IHC); 1:500 (WB)
phospho-eIF4e - Abcam - ab76256 - (1008447-6) - 1:150 (IHC)
β-catenin - BD Biosciences - 610154 - 1:50 (IHC)
Fibrillarin - CST - 2639 - (4) - 1:50 (IF-IHC)
Ki67 Clone MIB-1- Agilent - M7240 - (41584781) - 1:100 (IHC)
Mouse Envision - Agilent - K4001 - (11605675)
Rabbit Envision - Agilent - K4003 - (11623090)
Rat ImmPRESS kit - Vector Labs - MP-7444-15 - (ZK0118)
eIF2α - Cell Signaling - 9722 - (9) - 1:1000 (WB)
Vinculin - Abcam - ab129002 - 1:10000 (WB)
BiP - Abcam - ab21685 - (10) - 1:5000 (WB)
ATF4 - CST - 11815) - (6) - 1:1000 (WB)
GADD34 - Proteintech 10449-1-AP - (00102093) - 1:1000 (WB)
CHOP - CST - 2895 - (15) - 1:1000 (WB)
p53 [PAb 240] - Abcam - ab26 - 1:1000 (WB)
p21 - CST - 64016 - 1:1000 (WB)
puromycin - Sigma - MABE343 - (3305878) - 1:10000 (WB)
Beta-actin - Sigma - A2228 - (052M4816V) - 1:5000 (WB)
Goat anti-mouse IgG HRP - Invitrogen - A16078 (98-56-050423) - 1:10000 (WB)
Goat anti-rabbit IgG HRP - Invitrogen - A16110 - 1:10000 (WB)
Goat anti-Rabbit IgG (H+L) Alexa Fluor™Plus 800 - Thermo Scientific - A32735 - (ZB382373) - 1:10000 (WB)
Goat anti-Mouse IgG (H+L) Cross-Adsorbed Secondary Antibody, Alexa Fluor™ 680 - Thermo Scientific - A21057 - (2795341) - 1:10000 (WB)

Validation

All antibodies were used according to the manufacturers' instructions.

Anti-BrdU antibody, clone B44, is derived from hybridization of Sp2/0-Ag14 mouse myeloma cells with spleen cells from BALB/c mice immunized with iodouridine-conjugated ovalbumin. The antibody also reacts with iodouridine. Tissues derived from mice with no prior administration of BrdU have been stained as a negative control.

Glutamine Synthetase (Anti-GLUL) antibody produced in rabbit, is developed and validated by the Human Protein Atlas (HPA) project. Each antibody is tested by immunohistochemistry against hundreds of normal and disease tissues. These images can be viewed on the Human Protein Atlas (HPA) site. The antibodies are also tested using immunofluorescence and western blotting.

Lysozyme antibody staining on murine tissue has been verified in our recent paper in nature communications with DOI: 10.1038/s41467-023-44342-4, where oncogenic KRAS signalling suppresses Paneth cell differentiation.

NPM1 antibody has been validated within this study by comparing NPM1 proficient to NPM1 knockout tissue.

p21 antibody has been validated by the company by analysis of mouse p21WT fibrosarcoma and mouse p21KO fibrosarcoma tissue sections labeling p21 with ab107099.

p53 NCL-L-p53-CM5p is specific for mouse and rat p53 protein according to the manufacturer. It has also been validated within the CRUK Scotland Institute after staining murine sections following MDM2 knockout.

cleaved PARP antibody has been knock-out validated by the company were it was shown to react with Cleaved PARP in wild-type HAP1 cells treated with 1uM staurosporine for 3hrs in Western blot, with loss of signal observed in PARP knockout sample.

phospho-histone H2A.X antibody has been validated by Western blot analysis of extracts from untreated or UV-treated 293 cells by the company. It has also been validated to react with various cancerous tissues by IHC, and we also show increased reactivity to NPM1 KO murine intestinal tissue with lack of signal in corresponding control tissue in Extended Data Figure 8n.

phospho-4E-BP1 monoclonal antibody is produced by immunizing animals with a synthetic phosphopeptide corresponding to residues surrounding Thr37 and Thr46 of mouse 4E-BP1. It has been validated by the company to be reactive to murine tissue.

phospho-eEF2 has been verified for use in tissue sections in our lab and published material can be found in this published article doi: 10.7554/eLife.69729.

phospho-eIF2α antibody specificity has been confirmed by the company by treating cells with thapsigargin, which induces unfolded protein response, and confirming induction of phospho-eIF2α expression. Furthermore, In our manuscript we confirm the staining is absent when we treat with inhibitor against PERK, a kinase responsible for eIF2α phosphorylation.

phospho-eIF4e antibody has been validated by the company as well as in our lab in MNK1/2 knockout murine tissue samples (https://doi.org/10.1158/2159-8290.CD-20-0652).

For the β-catenin antibody we provide validation and specificity within our own data by showing nuclear translocation post APC loss in multiple murine tissues (see Extended Data Figure 1b).

Fibrillarin antibody specificity has been validated by CST for Western blot (WB) and immunofluorescence (IF/ICC) across multiple species (human, mouse, rat, and monkey), and it detects endogenous fibrillarin. Many peer-reviewed publications cite use of this

antibody and within our study it reliably localised to nucleoli.

Anti-Ki67 antibody is well established for immunohistochemichal use and cited widely. Some examples that confirm reactivity and reduced signal post drug treatments or genetic manipulations include doi.org/10.1038/s41388-019-1047-4, doi/10.1126/sciadv.abi7511 and doi.org/10.1038/s41467-019-08839-1.

eIF2α antibody has been assessed by western blot analysis of extracts from PC12 cells, untreated or thapsigargin-treated.

Vinculin antibody specificity confirmed with VCL knockout cell line validation by the company.

BiP antibody is validated for use in ICC/IF, IHC-P, WB in human, mouse, recombinant fragment samples by the company and has been cited over 646 times in peer reviewed journals.

ATF4 antibody has been validated by western blot analysis of extracts from 293 and HeLa cells, untreated (-) or tunicamycin-treated by the company, as well as by us using unicamycin-treated murine intestinal organoids.

GADD34 has been validated by WB anaysis of si-Control and si-GADD34 transfected PC-3 cells by the company.

CHOP antibody has subjected to western blot analysis of extracts from C6 and A-204 cells, untreated or treated with thapsigargin (300 nM, 2 hours) or tunicamycin (24 µg/ml, 2 hours) to promote stress response.

p53 [PAb 240] antibody has been KO validated for confirmed specificity by the company. ab26 was shown to specifically react with p53 in wild type HCT116 cells treated with irinotecan. No band was observed in p53 knockout HCT116 cells.

p21 (CST - 64016) antibody has been validated for Western blot by the company by analysis of extracts from vehicle control or Nutlin-3a-treated (10 µM, 24 hr) C2C12 cells and showing increased reactivity with the p21 antibody. This recent study also shows specificity to p21 in Cre-treated Mdm2ec/+;HRASG12V and p53fl/fl;HRASG12V MEFs exposed to 1µM milademetan compared to controls (https://doi.org/10.1038/s41586-024-08318-8).

Antibody against puromycin was validated in our own study by including lysates from organoids that had been treated with cyclohexamide to block protein synthesis along with puromycin, and they were clear of any signal when subjected to western blot analysis and probed against puromycin (see Extended Data Figure 9m).

Beta-actin antibody specificity was validated with western blot analysis in a study that performed b-actin cell knockout experiments (https://doi.org/10.1016/j.plasmid.2018.08.005).

# Eukaryotic cell lines

Policy information about cell lines and Sex and Gender in Research

| Cell line source(s) | Organoid lines from murine intestinal crypts were generated in house at the Cancer Research UK Scotland Institute. Npm1fl/fl lines comprised of 2 female and 1 male lines. Apcfl/fl;KrasG12D/+ lines comprised of 2 male and 1 female lines. Apcfl/fl;KrasG12D/+;Npm1fl/fl lines comprised of 2 male and 1 female lines. |
| --- | --- |
| Authentication | None of the cell lines that were used were authenticated since they were primary organoid lines generated in house from biologically independent mice. |
| Mycoplasma contamination | Cell lines were not tested for mycoplasma contamination. |
| Commonly misidentified lines (See ICLAC register) | n/a |

# Animals and other research organisms

Policy information about studies involving animals; ARRIVE guidelines recommended for reporting animal research, and Sex and Gender in Research

| Laboratory animals | All experiments were conducted on mice that had been bred on a C57BL/6 background for at least three generations (n≥3) for homeostasis experiments, and at least four generations (n≥4) for CRC models. For intestinal experiments male and female mice were induced between 8-15 weeks of age or as soon as they reached 20 g of body weight. For hepatocyte specific genetic recombination of the Npm1fl/fl alleles, male and female mice between 2-4 months of age were induced with adeno-associated virus expressing Cre under the control of the thyroxine binding globulin (TBG) promoter (AAV8.TBG.Cre). |
| --- | --- |
| Wild animals | The study did not involve wild animals. |
| Reporting on sex | Sex was not considered in study design for all homeostasis experiments and CRC models. Short term experiments assessing proliferation in the liver tumour models were performed only in male mice due to large cell proliferation variability between the sexes. For the long term liver cancer studies both sexes were represented and analysed and presented independently. Sex of all mice used in each experiment is indicated in the source data of related panels. |
| Field-collected samples | The study did not involve samples collected from the field. |

| Ethics oversight | All animal experiments were performed in accordance with UK Home Office regulations (project licences 70/8646 and PP3907577), with the approval, and under the oversight, of the animal welfare and ethical review board (AWERB) of the University of Glasgow. All experiments performed adhered to Institutional guidelines in full. |

Note that full information on the approval of the study protocol must also be provided in the manuscript.

## Plants

| Seed stocks | Report on the source of all seed stocks or other plant material used. If applicable, state the seed stock centre and catalogue number. If plant specimens were collected from the field, describe the collection location, date and sampling procedures. |
| Novel plant genotypes | Describe the methods by which all novel plant genotypes were produced. This includes those generated by transgenic approaches, gene editing, chemical/radiation-based mutagenesis and hybridization. For transgenic lines, describe the transformation method, the number of independent lines analyzed and the generation upon which experiments were performed. For gene-edited lines, describe the editor used, the endogenous sequence targeted for editing, the targeting guide RNA sequence (if applicable) and how the editor was applied. |
| Authentication | Describe any authentication procedures for each seed stock used or novel genotype generated. Describe any experiments used to assess the effect of a mutation and, where applicable, how potential secondary effects (e.g. second site T-DNA insertions, mosiacism, off-target gene editing) were examined. |

