## [Peer Review File · Nature Genetics]

Nucleophosmin supports WNT-driven hyperproliferation and tumour initiation

Corresponding Author: Professor Owen Sansom

Version 0:

Decision Letter:

25th Mar 2024

Dear Professor Sansom,

Your Article, "Nucleophosmin sustains WNT-driven cell proliferation and tumour initiation in vivo" has now been seen by 3 referees. You will see from their comments copied below that while they find your work of considerable potential interest, they have raised quite substantial concerns that must be addressed. In light of these comments, we cannot accept the manuscript for publication, but would be very interested in considering a revised version that addresses these serious concerns.

Reviewer #1 suggests increasing the relevance of your study to human disease by including human samples;

Reviewer #2 asks for Wnt-low controls, more validation and suggests expanding on a pan-cancer Npm1 dependency;

Reviewer #3 asks to expand the mechanistic insights on the link between NPM1 loss/ISR/p53 activation.

We hope you will find the referees' comments useful as you decide how to proceed. If you wish to submit a substantially revised manuscript, please bear in mind that we will be reluctant to approach the referees again in the absence of major revisions.

To guide the scope of the revisions, the editors discuss the referee reports in detail within the team, including with the chief editor, with a view to identifying key priorities that should be addressed in revision and sometimes overruling referee requests that are deemed beyond the scope of the current study. In this case, we invite you to address Reviewers' comments in full. Particularly, we would be interested in a revised version of the manuscript that includes human samples and a more developed mechanistic part on the NPM1-ISR-p53 activation link. We hope that you will find the prioritised set of referee points to be useful when revising your study. Please do not hesitate to get in touch if you would like to discuss these issues further.

If you choose to revise your manuscript taking into account all reviewer and editor comments, please highlight all changes in the manuscript text file. At this stage we will need you to upload a copy of the manuscript in MS Word .docx or similar editable format.

*2) If you have not done so already please begin to revise your manuscript so that it conforms to our Article format instructions, available [here](http://www.nature.com/ng/authors/article_types/index.html). Refer also to any guidelines provided in this letter.

*3) Include a revised version of any required Reporting Summary: <https://www.nature.com/documents/nr-reporting-summary.pdf>

Please be aware of our [guidelines](https://www.nature.com/nature-research/editorial-policies/image-integrity) on digital image standards.

Link Redacted

If you wish to submit a suitably revised manuscript we would hope to receive it within 6 months. If you cannot send it within this time, please let us know. We will be happy to consider your revision so long as nothing similar has been accepted for publication at Nature Genetics or published elsewhere. Should your manuscript be substantially delayed without notifying us in advance and your article is eventually published, the received date would be that of the revised, not the original, version.

Thank you for the opportunity to review your work.

Sincerely,
Chiara

Chiara Anania, PhD
Associate Editor
Nature Genetics
<https://orcid.org/0000-0003-1549-4157>

Referee expertise:

Referee #1: leukemia, nucleophosmin mutations expert, genomics

Referee #2: colorectal cancer, KRAS mutant cancer

Referee #3: signed review

Reviewers' Comments:

Reviewer #1:

Remarks to the Author:

This interesting study uses a mouse model-centric approach to describe a putative role for NPM1 in the context of several mouse solid tumors, and to define pathways that mediate this (Wnt, TP53). The study uses multiple complementary mouse models that are reported in detail and overall, convincing. I have two main comments before specific points: the degree of relevance to human disease is not clear limiting the appeal of the study in current form. This should be enhanced by observational analysis of human samples of the key mediators/pathways, and ideally, through mechanistic analysis (cell lines, organoids) to verify the results.

A second comment: the manuscript is long. As much of the results are reporting the mouse phenomena, disease latencies etc, it would be a more compelling and easier read if this could be condensed. One suggestion: the paragraph line 94 onward is results, and should be removed.

Specific points:

The MS opens with a bold statement about epigenetic variability as the driver of mutational variability in cancer. This is overreach and peripheral to the MS, and could be removed or toned down (or better supported by data if believed by the

authors).

Paragraph two is a bit out of date regarding the biology of NPM1c, where it is now known that there is residual chromatin bounds NPM1c that mediates HOX deregulation and transformation (recent papers from the Armstrong group et al in Cancer Discovery).

Human validation data is limited and not strong: e.g. upregulation of NPM1c in SF1d with very broad overlap in expression in comparison groups, no IHC/western etc.

Data for Figure 3c-d: why is the tumor number the same in the two groups?

Line 233 - the figure is the wrong number (should be 3k-l, not 2k-l)

Figure 4 a-c - expression of P53 is key, but it is very hard to see staining here.

I am not fully persuaded that the ISR is the main mechanism - there do appear to be some changes, but they are extremely subtle (Fig 5e).

Why is trazodone used? This is an SSRI, thus has at least two different mechanisms of action. Is a mechanism-specific drug available? How is this considered to result in P53 pathway activation? This seems central, but not explained (para line 366. Other mechanisms are mentioned (lin 481) but not tested.

Reviewer #2:

Remarks to the Author:

In this study, the authors investigated the relationship between Nmp1 and its role in intestinal and hepatic cancer in mice. They report a cell autonomous role of Nmp1 by which loss of Apc (or B-catenin activating mutations) upregulates Nmp1, and upon Nmp1 deletion, proliferation is suppressed resulting in reduced tumor formation. In addition, they show Nmp1 is dispensable for normal tissue homeostasis. The mechanism was traced back to Nmp1 loss leading to activation of an integrated stress response, which in turn, upregulates p53/p21 signaling. These findings suggest a potential intervention strategy for Wnt activated cancers, as Nmp1 inhibition may counteract the tumor promoting effects of Apc-loss or B-catenin activating mutations.

Overall, the conclusion that Nmp1 promotes tumor formation via attenuating the integrated stress response and p53 signaling are interesting and potentially important. However, there are several limitations which requires additional work to address.

1. Notably, I believe the authors have to validate the proposed mechanism linking Nmp1, p53, and the integrated stress response. While the authors demonstrate an epistatic relationship via co-deletion of Nmp1 and p53, it remains unclear how Nmp1 loss leads to elevated p53 and p21 protein expression as neither were upregulated in the RNAseq data and no increases were observed in ribosome occupancy of either gene. How do the authors think p53 and p21 are being upregulated?

2. Additionally, p53 activation invariably leads to p21 transcriptional upregulation. How do the authors reconcile this point as p21 was not upregulated at the mRNA level? Related to the above point, if p53 and p21 were playing a major role downstream of Nmp1 loss, you would expect the top pathways observed to be related to a DNA damage and cell cycle checkpoint response, however neither pathway emerged in the unbiased pathway analysis performed in figure 5C.

3. The authors should check whether Nmp1 loss leads to elevated DNA damage. Are there more γH2AX-positive cells in Nmp1 deficient cancer or normal tissue?

4. This study demonstrates the importance of Nmp1 in cancer growth. Are there pan-cancer dependencies of Nmp1? The authors should investigate publicly available CRISPR dependency datasets, such as DepMap. Additionally, is Nmp1 specific to Apc and b-catenin altered cancers? Does Nmp1 loss have an impact in p53 mutated cell lines?

5. Lastly, throughout the paper many claims are made regarding the specificity of Nmp1 loss being a vulnerability if Wnt-high cancer, but no "Wnt-low" tumor controls were used throughout the study. Additional in vivo experiments or analysis need to be performed with "Wnt-low" controls.

Reviewer #3:

Remarks to the Author:

In this article Kanellos et al, provide evidence suggesting that NPM1 is induced upon APC loss in WNT signaling-responsive tissues. The authors further show that abrogation of NPM1 does not inconspicuously affect most of the adult tissues, but that it prolongs survival in WNT-driven as CRC (+/- KRAS) and HCC (+/-MYC) models. Some evidence is also provided suggesting that NPM1 loss leads to induction of ISR and activation of p53/p21-dependent pathways, whereby abrogation of ISR lead to apparent decrease in p21 levels. Overall, it was thought that this article is of high potential interest as it may establish the role of NPM1 in solid tumors (e.g., beyond AML), while also proving previously unappreciated insights in oncogenic WNT signaling. To this end, it was thought that this article is likely to contribute significantly to improved understanding of the molecular underpinnings of HCC and CRC, and thus to be of broad interest to researchers from various biomedical disciplines ranging from cell and developmental to cancer biology. Overall, it was thought that most of the authors conclusions are adequately supported by the data, whereby the methodology was found to be sound in general. Notwithstanding these clear strengths of the report, several addressable issues mostly related to the lack of mechanistic data to support proposed model were raised. My specific comments are outlined below:

Major concerns:

1. Although several possibilities were discussed, it was thought that there is a lack of the mechanism whereby NPM1 impinges on p53 axis in the context of author's model. This may be of interest as NPM1 abrogation seemed to result in minimal changes in the transcriptome (changes are mostly in the translome), whereby it would be anticipated that these would be profound based on the perturbations in p53 function. To address this, it was stated that "Activation of p53 Following loss of NPM1 is subtle enough to avoid triggering a global p53 transcriptional program and permit homeostatic proliferation, while only affecting the increased proliferation levels required for oncogenic transformation", but having the experimental evidence how this occurs would further strengthen the applicant's model. To this end, notwithstanding that the previous literature was discussed, it may be of the interest to authors to further delineate the interplay between NPM1 and p53 in the context of dysregulated WNT signaling in CRC and HCC. This was thought to be of particular importance as many of different post-transcriptional and post-translational mechanism of p53 have been described in the literature. I appreciate that the effects of NPM1 loss on TP53 transcription or mRNA stability were largely excluded (via RNAseq data). I don't mean to torture the authors by making them conduct some extraordinarily complicated experiments as these may invariably result in leading them into rabbit holes, but few simple experiments such as looking at subcellular localization, interaction with HDM2/stability of p53 protein would be informative. Out of these types of experiments and sequencing data, it should be relatively trivial to identify the potential culprits. Of note, did TP53 mRNA pass the threshold in Ribo-seq arm of the study?
2. Notwithstanding that it is reasonable to propose that ISR may limit hyperproliferative phenotypes in e.g., intestine, when the tumor grows and outstrips the vasculature, ISR may play a protective role by allowing cancer cells to adapt to limited amounts of oxygen and nutrients. There is some evidence that ISR may have antitumorigenic activity in the experiments using trazodone and monitoring survival of the mice in the supplementary figure 6i, but the trazodone is quite pleiotropic and it is hard to pin them on ISR. Also, I could not find controls showing that phospho-eIF2alpha is indeed downregulated in these mice throughout tumor evolution nor information on tumor size in this experiment). Moreover, trazodone treatment was started quite early (5 days after induction), so these effects are likely to be illustrative of the effects of the drug on the initial tumor growth and not tumor maintenance wherein the ISR may have a protective role. It may thus be advantageous to investigate this potential antagonistic pleiotropy of ISR in the context of neoplasia by monitoring not only proliferative indexes, but also tumor growth and progression under conditions where ISR is continuously disrupted in the cells where NPM1 is deleted by either pharmacological or genetic (S51A eIF2alpha KI) approaches, and also disrupting ISR after tumors have already grown to certain size to investigate the effects on tumor maintenance. This would also be of importance to establish the role of proposed mechanisms during tumor evolution. Alternatively, the authors should consider limiting their conclusion to reduction of proliferation rather than tumor growth/progression.
3. It remains rather unclear how ISR is triggered upon NPM1 loss. The authors provide some evidence using PS that elongation may be stalled at the 5' ends of main ORFs, but it is not clear how the conclusion of imbalance of initiation and elongation rates was derived, as at least to me it remains unclear how the changes in the initiation rates were inferred? Notably, in the "classical" ISR the initiation rates should be strongly reduced. Is global protein synthesis reduced under these conditions? Based on this, perhaps the authors should consider that the response to NPM1 loss is distinct from "classical" ISR and constitutes a hitherto unappreciated translational reprogramming? This may be quite worthy of considering because mTORC1 appears to be simultaneously upregulated (and considering that this is KRAS driven model, one would expect phosphorylation of eIF4E also to go up) and thus, NPM1 loss may result in high eIF4F/low TC scenario that is likely to be distinct from the "classical" ISR. One of the key experiments that appears to be missing here is monitoring the levels of global protein synthesis +/- e.g., ISRIB as this may allow authors to distinguish between their observations and "classical" ISR. Another possible scenario may also be derived from this relatively simple experiment, which is that if the protein synthesis is upregulated in these tumors, induction of eIF2alpha phosphorylation may be secondary to overload of ER and activation of PERK and thus although pro-proliferative in the initial stages, might actually be an adaptive mechanism in fully grown tumors where the cells are exposed to stress (please see the comment above). Although protein synthesis was unchanged upon NPM1 loss in normal cells, considering that these models are driven by KRAS or MYC and that at least in the KRAS model mTORC1 seems to be hyperactive, it would not be unreasonable to expect that protein synthesis may be altered in this context.
4. Related to the above, validating at least some of the mRNAs and corresponding proteins identified to be differentially translated upon NPM1 loss seems to be warranted (looking at total mRNA and protein levels would suffice, or considering the complexity of the model even IHC for some of the targets alone would be sufficient). Also, reporting what happens to translation (from Ribo-seq data) and protein levels (e.g., by IHC) of "classical" ISR targets (e.g. ATF4, CHOP, GADD34) may

be informative.

Minor comments:

-GSK2606414 shows of target effects (PMID: 30931942) and as for the other similar inhibitors, it is expected that it will induce other eIF2alpha kinases in a concentration dependent manner (PMID: 37684277). Albeit this concern was largely mitigated by the experiments employing ISRIB, the authors should I think make a note of this to avoid further confusion in the literature.

-Using log-based ratios to determine translation efficiency comes with the perils of spurious correlation (<https://royalsocietypublishing.org/doi/abs/10.1098/rspl.1896.0076>), whereby methods based on the analysis of partial variance are likely to be superior (e.g., PMID: 21115840).

-Line 220 "Notably, the surrounding normal epithelium was negative for NPM1 expression, suggestive of its importance for tumour initiation". The authors should clarify this, as it is not completely clear to me why the loss of NPM1 in surrounding tissue would be indicative of NPM1 role in tumor initiation. I found this statement to be relatively confusing as across most of the models the NPM1 status does not appear to affect the number of tumors except in the KRAS/APC model, albeit NPM1 +ve escapers appears to be comparable across all the models.

-The authors should consider indicating the localization of the tumors throughout the article (i.e., do they form in small intestine, colon...?).

-Figure 5b covers about 1/2 of protein encoding genes, and this should perhaps be noted as there may still be genes that were regulated in the other 1/2 of the protein coding genes that were likely below the detection level.

-In Supp. Figure 5 the authors should consider adding additional QC data typically used for Ribo-seq (e.g., correlations between the replicates, reading frames/riboWaltz), and provide more detail in methodology regarding applied batch correction. Finally, the author may consider comparing their translational signatures to publicly available translational signatures (e.g., for ISR induction). This may be of interest as it is a bit surprising that there are no perturbations in the total mRNA levels which usually are quite perturbed under ISR (e.g., via ATF4 activation), and thus the stress response observed upon NPM1 loss may be novel and distinct from "classical" ISR (also see the comment # 3 under "major comments" section

-The authors should consider noting which phosphoacceptor sites on proteins were tested in figures and legends.

-It may be pertinent to perhaps look at the morphology of the nucleoli or indications of nucleolar stress in NPM1-abrogated vs. control cells, especially considering implication of p53.

I hope that the authors will find my comments constructive and of sufficient pathos.

Sincerely

I/Topisirovic

Version 1:

Decision Letter:

Our ref: NG-A64660R

15th Jul 2025

Dear Dr. Sansom,

Thank you for submitting your revised manuscript "Nucleophosmin supports WNT-driven hyperproliferation and tumour initiation" (NG-A64660R). It has now been seen by the original referees and their comments are below. The reviewers find that the paper has improved in revision, and therefore we'll be happy in principle to publish it in Nature Genetics, pending minor revisions to comply with our editorial and formatting guidelines.

Congratulations!

Sincerely,
Chiara

Chiara Anania, PhD
Associate Editor
Nature Genetics
<https://orcid.org/0000-0003-1549-4157>

Reviewer #1 (Remarks to the Author):

The authors have striven to provide extensive additional data and clarification, which is appreciated. I do think there remains uncertainty as to how NPM regulates transcriptional/translational perturbations, and as to why TP53 may be selectively different in this regard. But there are strong supportive correlative data from the mouse models and drug studies.

Reviewer #2 (Remarks to the Author):

We appreciate the authors' new data and their further explanation of the points raised in our review. We also commend the significant amount of genetic in vivo data presented in this study. Specifically, the authors have clearly addressed our concerns regarding the Npm1-p53 axis with the additional γ H2AX data. We believe they have provided sufficient evidence to address our original concerns, as well as those of the other reviewers.

Reviewer #3 (Remarks to the Author):

The authors have substantially revised that paper which was highly appreciated as it was thought that additional data significantly strengthened the article. I thought that the authors have addressed all of my questions and comments in a satisfactory manner, and to this end I have no further concerns.

Sincerely

I/Topisirovic

Version 2:

Decision Letter:

In reply please quote: NG-A64660R1 Sansom

11th Oct 2025

Dear Dr. Sansom,

I am delighted to say that your manuscript "Nucleophosmin supports WNT-driven hyperproliferation and tumour initiation" has been accepted for publication in an upcoming issue of Nature Genetics.

Your paper will be published online after we receive your corrections and will appear in print in the next available issue. You can find out your date of online publication by contacting the Nature Press Office (press@nature.com) after sending your e-

proof corrections.

Authors may need to take specific actions to achieve compliance with funder and institutional open access mandates. If your research is supported by a funder that requires immediate open access (e.g. according to [a Plan S principles](https://www.springernature.com/gp/open-science/plan-s-compliance) or the [NIH public access policy](https://www.springernature.com/gp/open-science/us-federal-agency-compliance)) then you should select the gold OA route, and we will direct you to the compliant route where possible. Because authors warrant under our subscription licensing terms that they haven't committed to licensing any version of their article under a licence inconsistent with the terms of our agreement – including the applicable embargo period – publication under the subscription model isn't suitable for authors whose funders require no embargo.

An online order form for reprints of your paper is available at [a https://www.nature.com/reprints/author-reprints.html](https://www.nature.com/reprints/author-reprints.html). Please let your coauthors and your institutions' public affairs office know that they are also welcome to order reprints by this method.

If you have not already done so, we strongly recommend that you upload the step-by-step protocols used in this manuscript to protocols.io. protocols.io is an open online resource that allows researchers to share their detailed experimental know-how. All uploaded protocols are made freely available and are assigned DOIs for ease of citation. Protocols can be linked to any publications in which they are used and will be linked to from your article. You can also establish a dedicated workspace to collect all your lab Protocols. By uploading your Protocols to protocols.io, you are enabling researchers to more readily reproduce or adapt the methodology you use, as well as increasing the visibility of your protocols and papers. Upload your Protocols at <https://protocols.io>. Further information can be found at <https://www.protocols.io/help/publish-articles>.

Sincerely,
Chiara

Chiara Anania, PhD
Associate Editor
Nature Genetics
<https://orcid.org/0000-0003-1549-4157>

Click here if you would like to recommend Nature Genetics to your librarian
<http://www.nature.com/subscriptions/recommend.html#forms>

** Visit the Springer Nature Editorial and Publishing website at http://editorial-jobs.springernature.com?utm_source=ejp_NGen_email&utm_medium=ejp_NGen_email&utm_campaign=ejp_NGen for more information about our career opportunities. If you have any questions please click [here](mailto:editorial.publishing.jobs@springernature.com).

We would like to thank the reviewers for their time and effort in evaluating our work, as well as for their constructive feedback. We are pleased that they found our methodology and data to be generally convincing and of significant interest. We agree that our findings represent an important advance in understanding the roles of NPM1 beyond AML and may open new avenues for future research.

We have carefully considered all reviewer comments and have implemented changes in the revised manuscript, which we believe has been significantly improved. Below we provide a point-by-point response to each comment. Our responses are shown in blue.

Of note, we made a minor adjustment to the original manuscript title, changing it from 'Nucleophosmin sustains WNT-driven cell proliferation and tumour initiation *in vivo*' to 'Nucleophosmin supports WNT-driven hyperproliferation and tumour initiation'. This change was made to distinguish pathological, WNT-driven hyperproliferation from normal, homeostatic WNT-mediated proliferation (e.g. in the intestinal epithelium).

Reviewers' Comments:

Reviewer #1:

Remarks to the Author:

This interesting study uses a mouse model-centric approach to describe a putative role for NPM1 in the context of several mouse solid tumors, and to define pathways that mediate this (Wnt, TP53). The study uses multiple complementary mouse models that are reported in detail and overall, convincing. I have two main comments before specific points: the degree of relevance to human disease is not clear limiting the appeal of the study in current form. This should be enhanced by observational analysis of human samples of the key mediators/pathways, and ideally, through mechanistic analysis (cell lines, organoids) to verify the results.

We would like to thank the reviewer for their insightful comments and suggestions and for appreciating the strength of our *in vivo* models. In response to the first main point regarding the degree of relevance to human disease, we have incorporated additional analyses of publicly available human datasets that we believe strengthen the translational relevance of our findings. We now show that *NPM1* is highly expressed across multiple tumour types, with colorectal cancer (CRC) exhibiting the highest degree of overexpression (Fig. 1e). We also show that *NPM1* expression is significantly correlated with the degree of WNT pathway activation across different tumour types, as well as within CRC samples specifically (Fig. 1f,g; Extended data Fig. 1d,e). Using an in-house built human CRC TMA we demonstrate that tumours with high *NPM1* expression have significantly increased Ki67 positivity. Additionally, analysis of an independent CRC cohort confirms a significant correlation between *NPM1* expression and proliferative index (Fig. 1h-k). To validate the relevance specifically to APC loss, we show that human tumours with *APC* mutations have higher *NPM1* expression (Extended data Fig. 5f). Furthermore, CRCs with the highest *NPM1* expression fall within the PDS1 subtype of the Pathway-Derived Subtype (PDS) classification, which is characteristic of high proliferative tumours enriched in *APC* mutations (Fig. 1l). Most notably, we show that p53-mutant cancers tend to express lower levels of *NPM1*. Importantly, however, nearly all (~91%) of CRCs harbouring shallow *NPM1* deletions, also possess *TP53* mutations. This mirrors our mouse model data highlighting the critical role of p53 in tumour establishment following *NPM1* loss (Fig. 3r-t). Similar findings

are also provided for human hepatic cancers (Extended data Fig. 10b-h; Extended data Fig. 10x,y), highlighting a conserved role of NPM1 in WNT-driven tumorigenesis across multiple tissues.

A second comment: the manuscript is long. As much of the results are reporting the mouse phenomena, disease latencies etc, it would be a more compelling and easier read if this could be condensed. One suggestion: the paragraph line 94 onward is results, and should be removed.

Despite incorporating additional data in response to the reviewers' comments, we have also succeeded in condensing the manuscript. We hope that the reviewers find the revised version clearer and improved in overall flow.

Specific points:

The MS opens with a bold statement about epigenetic variability as the driver of mutational variability in cancer. This is overreach and peripheral to the MS, and could be removed or toned down (or better supported by data if believed by the authors).

We apologise for the confusion caused by the opening statement. It was not our intention to imply that epigenetic variability is a direct driver of mutational variability in cancer. Rather, we meant to introduce the concept of tissue-specific mutation 'permissiveness', whereby the same oncogenic mutation might drive proliferation and transformation in one tissue but remain silent in another, likely due (in part) to epigenetic variability between tissues, as suggested by the referenced study. To avoid misinterpretation, we have revised the opening sentence which now reads 'the response to specific drivers of proliferation and oncogenic transformation varies greatly among tissues'. We hope this clarifies our intent.

Paragraph two is a bit out of date regarding the biology of NPM1c, where it is now known that there is residual chromatin binds NPM1c that mediates HOX deregulation and transformation (recent papers from the Armstrong group et al in Cancer Discovery).

This section has been condensed in the revised manuscript, however, the relevant sentence has been updated, and the studies mentioned have been cited.

Human validation data is limited and not strong: e.g. upregulation of NPM1c in SF1d with very broad overlap in expression in comparison groups, no IHC/western etc.

As noted in our response to the first main comment, we have substantially expanded the analyses supporting the human relevance of our findings. The figure initially referenced has been removed in the revised manuscript and replaced with more comprehensive analyses where the distinction between normal tissue and colorectal cancer samples is more clearly demonstrated (Fig. 1e,j). We would like to clarify that throughout the manuscript, we assess expression of the wild-type *NPM1*, not the mutated NPM1c isoform frequently expressed in AML.

Data for Figure 3c-d: why is the tumor number the same in the two groups?

In this figure we presented tumour number at the clinical endpoint and concluded that *Npm1* deletion suppresses tumour formation. This conclusion was based on the fact that the vast majority of tumours driven following the loss of *Apc* retain NPM1 expression. This selective pressure arguably contributes to the extension of survival observed following *Npm1* loss. Our interpretation is that more time is required for tumours to form from a reduced pool of *Npm1*⁺ cells, which are more readily transformed upon stochastic loss of the second *Apc* allele. This delay ultimately results in a comparable tumour burden that eventually causes anaemia, weight loss, and disease-related clinical endpoint. Although the total number of tumours does not reach statistically significant difference between the groups at endpoint, spatial analysis of tumour distribution throughout the intestine (Extended data Fig. 5k) reveals a significant suppression of tumour formation in regions that typically exhibit the greatest transformation potential.

Line 233 - the figure is the wrong number (should be 3k-l, not 2k-l)

We thank the reviewer for spotting this discrepancy. The text and figures have been substantially revised in the updated manuscript. We have carefully cited all figures to ensure they are currently referenced, but please do let us know if any discrepancies remain.

Figure 4 a-c - expression of P53 is key, but it is very hard to see staining here.

We appreciate that the p53 staining using chromogenic IHC detection is subtle. To address this, we have repeated the p53 staining using immunofluorescence and updated the corresponding quantification accordingly (Fig.3a-b, h-i). p53-positive cells are now clearly labelled. Of note, p53 positivity is markedly reduced following ISRIB treatment (Fig. 6m,o), which supports the specificity of the antibody used.

I am not fully persuaded that the ISR is the main mechanism - there do appear to be some changes, but they are extremely subtle (Fig 5e).

We appreciate the reviewer's concern and acknowledge that the figure referenced in the original manuscript may have given the impression that the observed differences were subtle. However, we would argue that the data demonstrated a substantial difference, which is a shift in ribosome distribution across nearly all translated mRNAs in *Npm1*-deleted compared to *Npm1*-proficient cells. Nevertheless, we have removed this figure from the revised manuscript to avoid any confusion. Instead, aside from the ribosome occupancy plot showing a clear shift in ribosome occupancy upon NPM1 loss (Fig. 4h), we also incorporated new analysis clearly highlighting the induction of ribosome pausing at the start of coding sequences (Fig. 4f). To further investigate the role of translational regulation we also included proteomic analysis. This revealed that top pathways are inversely enriched when comparing ribosome protected fragment (RPF) levels with protein levels, providing evidence of disrupted protein synthesis downstream of NPM1 loss (Fig. 4b-e). Finally, our ability to rescue suppressed proliferation upon NPM1 loss using three independent compounds (at least two of which targeting distinct parts of the pathway) makes us confident that protein synthesis stress is the major contributor to the observed phenotype.

Why is trazodone used? This is an SSRI, thus has at least two different mechanisms of action. Is a mechanism-specific drug available? How is this considered to result in P53 pathway activation? This seems central, but not explained (para line 366). Other mechanisms are mentioned (lin 481) but not tested.

We appreciate that many of these drugs that reverse the effects of ISR like trazodone, while effective in phenotypic rescue, are not very mechanism-specific and may have off-target effects. Precisely for this reason, and given the central role of ISR in the phenotypes following *Npm1* loss, we employed three different compounds with different actions on the same pathway - ISRIB (small-molecule inhibitor of the ISR), GSK2606414 (a PERK inhibitor that prevented phosphorylation of eIF2 α), and trazodone (identified in a drug screen for mimicking the effects of ISRIB¹) - to consistently reproduce rescue of *Npm1*-null intestinal cell proliferation. All three compounds successfully rescued proliferation in the short term, reinforcing our conclusion that ISR activation is a key mediator of the *Npm1*-loss phenotype.

While ISRIB is the more specific ISR inhibitor, it is poorly tolerated over prolonged treatment *in vivo*². We also observed this in our own experiments with *Npm1*-deficient animals showing increased sensitivity to the drug (Letter Fig. 1). Similarly, PERK inhibition is also not tolerated long term³. In contrast, trazodone is the only drug well-tolerated for extended treatment periods and was suitable for use in our long-term tumour models. While we agree that trazodone is not a specific p-eIF2 α inhibitor and may have additional, unrelated targets (as noted in our manuscript), we believe its inclusion in extended data complements our main findings and supports the central hypothesis that ISR activation mediates the *Npm1*-loss phenotype.

With regards to the p53 activation, this is a downstream consequence of *Npm1* deletion and our data indicate that it is ISR-dependent, since ISR inhibition prevented the accumulation of p53 and p21. To demonstrate this main mechanistic part we used ISRIB, the specific ISR inhibitor.

Letter Figure 1. Survival curves of untreated *Apc*^{fl/fl}*Kras*^{G12D/+} (n=17) and *Apc*^{fl/fl}*Kras*^{G12D/+}*Npm1*^{fl/fl} (n=20) mice, compared to that of mice of the same genotypes treated with ISRIB from the fifth day post-induction until clinical endpoint (*Apc*^{fl/fl}*Kras*^{G12D/+} n=6, *Apc*^{fl/fl}*Kras*^{G12D/+}*Npm1*^{fl/fl} n=6). Median survival in days indicated in brackets. P-values obtained by Log-rank (Mantel-Cox) tests. Censored mice denoted as tick marks at indicated times post-induction.

For the additional mechanisms discussed, we have now tested a few in follow up experiments. For instance, we explored the ribosome collision hypothesis further and found that there is marginal increase of phosphorylated p38 MAPK (a downstream marker of ZAK signalling, which is associated with ribosome collisions) (Letter Fig. 2a,b). However, treatment with vemurafenib, a known, though not specific, ZAK inhibitor⁴⁻⁶, did not rescue proliferation following NPM1 loss, arguing against ZAK-mediated stress signalling as a primary mechanism (Letter Fig. 2c,d). Given NPM1's roles as a molecular chaperone, we also investigated whether its loss led to protein aggregation and cellular stress. However, these experiments did not yield convincing evidence supporting this mechanism.

We appreciate that the full mechanistic functions of NPM1 in cellular homeostasis as well as in disease remain to be explored. However, as discussed above and in response to the other reviewers, we believe the current study has already made substantial advances. Incorporation of the proteomic data and new analysis on ribosome pauses on mRNAs, provide additional evidence that NPM1 loss leads to deregulated protein synthesis but without directly affecting p53 and p21 that are spared from this deregulation. The NPM1 biology proves complex and offers exciting opportunities for future investigation. However, we believe that our key findings - (1) the identification of NPM1 as critical for solid tumour initiation downstream of WNT signalling, (2) the surprising tolerance of NPM1 loss in adult tissues, suggesting potential for therapeutic targeting, and (3) the central role of p53 coupled with a protein synthesis stress response in mediating NPM1 dependency - are well supported by our current data and represent important advances. Further investigation of NPM1's broader roles will be the focus of future studies.

Letter Figure 2. a, Western blot analysis from epithelial extracts of *Apc^{fl/fl}Kras^{G12D/+}* and *Apc^{fl/fl}Kras^{G12D/+}Npm1^{fl/fl}* (n=4 per group) and (b) quantification relative to controls. Data, mean \pm SEM. Statistical significance determined by unpaired two-tailed t test. c, Representative staining for BrdU (top) and NPM1 (bottom) on SI tissue sections from *Apc^{fl/fl}Kras^{G12D/+}* and *Apc^{fl/fl}Kras^{G12D/+}Npm1^{fl/fl}* animals treated with vemurafenib daily (n=4 per group) and collected four days post-induction. d, Quantification of BrdU+ cells in SI half-crypts of animals from the groups shown in (c) and compared to untreated animals of the same genotypes. Data, mean \pm SEM. Data statistically assessed by one-way ANOVA Tukey's multiple comparisons test.

Reviewer #2:

Remarks to the Author:

In this study, the authors investigated the relationship between *Nmp1* and its role in intestinal and hepatic cancer in mice. They report a cell autonomous role of *Nmp1* by which loss of *Apc* (or B-catenin activating mutations) upregulates *Nmp1*, and upon *Npm1* deletion, proliferation is suppressed resulting in reduced tumor formation. In addition, they show *Nmp1* is dispensable for normal tissue homeostasis. The mechanism was traced back to *Npm1* loss leading to activation of an integrated stress response, which in turn, upregulates p53/p21 signaling. These findings suggest a potential intervention strategy for Wnt activated cancers, as *Npm1* inhibition may counteract the tumor promoting effects of *Apc*-loss or B-catenin activating mutations.

Overall, the conclusion that *Nmp1* promotes tumor formation via attenuating the integrated stress response and p53 signaling are interesting and potentially important. However, there are several limitations which requires additional work to address.

We would like to thank the reviewer for their critical assessment of our work and for recognising its interest and potential significance. Please find below our detailed responses to the individual comments.

1. Notably, I believe the authors have to validate the proposed mechanism linking *Nmp1*, p53, and the integrated stress response. While the authors demonstrate an epistatic relationship via co-deletion of *Nmp1* and p53, it remains unclear how *Npm1* loss leads to elevated p53 and p21 protein expression as neither were upregulated in the RNAseq data and no increases were observed in ribosome occupancy of either gene. How do the authors think p53 and p21 are being upregulated?

We agree with the reviewer that this was a puzzling aspect of our findings. As the reviewer points out a 'classical' p53 response typically inflicts transcriptional changes, which are not evident in our RNA-seq or RPF data. We believe this reflects the subtleness of this response when *Npm1* is deleted. Consistent with this, and as noted in the manuscript, general homeostasis remains largely unaffected following *NPM1* loss, which supports a limited transcriptional response. Irrespective of the lack of transcriptional/translational upregulation, we believe there is strong evidence supporting a critical functional role for p53 downstream of *NPM1* loss, since genetic deletion of p53 fully rescues the phenotypes associated with *Npm1* deletion. However, to address this concern we have incorporated a new piece of evidence in the revised manuscript that could help explain the elevated p53 and p21 protein levels. While *NPM1* loss generally induces ribosome pausing and leads to a shift in ribosome occupancy toward the start of the coding sequence, we observed that the transcripts encoding p53 and p21 do not follow this trend. In fact, in the case of p53, *NPM1* loss appears to resolve ribosome pausing sites, arguably allowing for more efficient translation (Extended Data Fig. 8v-x). This suggests that despite a global translation deregulation, specific transcripts encoding p53 and p21 have been spared, and highlight a potential explanation for the observed disconnect between transcription, ribosome profiling data and protein levels.

2. Additionally, p53 activation invariably leads to p21 transcriptional upregulation. How do the authors reconcile this point as p21 was not upregulated at the mRNA level?

Related to the above point, if p53 and p21 were playing a major role downstream of *Npm1* loss, you would expect the top pathways observed to be related to a DNA damage and cell

cycle checkpoint response, however neither pathway emerged in the unbiased pathway analysis performed in figure 5C.

As discussed in the previous point, the reviewer is right in suggesting that generally p53 activation is associated with transcriptional upregulation of p21. As noted in our response to the previous comment, we believe that one way p21 is upregulated in this context is by being spared from a translation deregulation and being efficiently translated in the absence of NPM1. Unfortunately, p21 itself was not detected in our proteomic data, however, we have sufficient evidence throughout the manuscript of its enhanced expression following NPM1 loss, as well as of its significance (p21 loss phenocopies *Trp53* deletion and rescues tumorigenesis in NPM1-depleted guts). We were reluctant ourselves to fully accept this data, but once two independent RNA sequencing experiments and qPCRs in two different genetic models produced the same result there was strong evidence supporting the lack of transcriptional response.

The reviewer is correct that in the referenced graph DNA damage related pathways were not enriched in the analysis shown. However, that specific enrichment analysis was based on Gene Ontology terms for Molecular Function and Biological Process using a preranked list by Translational Efficiency ($\log_2\text{FC RPFs} - \log_2\text{FC total mRNA}$). When we perform pathway enrichment using preranked RPFs $\log_2\text{FC}$ alone and not normalised for changes in total mRNA, DNA repair does emerge as an enriched pathway (Fig. 4b). We also observed a moderate correlation between Replication stress and DNA repair signatures with *NPM1* expression in human CRC samples (Extended Data Fig. 8l,m). Since implementation of proteomic data revealed this inverse pathway enrichment between RPF and proteomic datasets, which complement our histological findings in supporting a protein synthesis stress response, we have now removed all Translational Efficiency-based analyses and replaced them with analyses based solely on RPF data. We hope this will reduce confusion and improve data interpretation.

3. The authors should check whether *Npm1* loss leads to elevated DNA damage. Are there more γH2AX -positive cells in *Npm1* deficient cancer or normal tissue?

We thank the reviewer for this suggestion. We have now performed IHC staining for γH2AX in NPM1-depleted intestinal tissues (Extended Data Fig. 8n-q). Consistent with the point above highlighting DNA repair pathway enrichment following NPM1 loss, we confirm a marked increase in γH2AX positive cells, both in normal tissue as well as downstream of oncogenic activation. Notably, this increase does not coincide with a corresponding increase in apoptosis, as indicated by cleaved-PARP staining (Extended Data Fig. 8r-u). These results are in line with our conclusion that NPM1 loss elicits a subtle p53 response that does not disrupt general tissue homeostasis.

4. This study demonstrates the importance of *Npm1* in cancer growth. Are there pan-cancer dependencies of *Npm1*? The authors should investigate publicly available CRISPR dependency datasets, such as DepMap. Additionally, is *Npm1* specific to *Apc* and *b-catenin* altered cancers?

Does *Npm1* loss have an impact in p53 mutated cell lines?

This is something we were curious about ourselves since we share the belief that NPM1 may play a broader role in supporting tumour growth across multiple cancer types. However, in our attempt to investigate *Npm1* pan-cancer dependencies we encountered notable inconsistencies in the NPM1 data across versions of DepMap. The most recent DepMap

release (24Q3) includes 1,150 cell lines. However, only 333 of them have NPM1 data available. By comparison, the 20Q1 (2020 quarter 1 release) includes 739 cell lines, with 726 of them having NPM1 dependency scores available. Looking at the overlapping data between the two versions, the CRISPR gene effect for *NPM1* is not consistent between the two versions (Letter Fig. 3, left panel). This inconsistency appears specific to NPM1 as, for example, *KRAS* shows strong correlation across releases (Letter Fig. 3, right panel). Nevertheless, we performed some pan-cancer analysis using the more complete 20Q1 dataset. However, because of these inconsistencies mentioned above, and given that the manuscript is already very data-rich, we opted not to include these results in the main text, but summarise them here instead.

Letter Figure 3. *NPM1* (left panel) and *KRAS* (right panel) CRISPR gene effect across the two DepMap releases indicated.

Across tumour types, bowel, liver, and oesophagus/stomach consistently ranked among the top cancer types with high WNT activation as expected (Letter Fig. 4a,b). However, the average CRISPR gene effect score for *NPM1* (~ -0.75) (Letter Fig. 4c,d) suggests that *NPM1* is broadly essential across many cancer types, so potentially not limiting to just APC/b-catenin altered cancers.

We also examined the relationship between *NPM1* dependency and *TP53* mutation status. While there was a trend for many cell lines to be slightly more dependent on *NPM1* with wild-type *TP53*, this was not statistically significant across most cancer types except skin (Letter Fig. 4e). Importantly though, the CRISPR gene effect of *NPM1* strongly anti-correlates with the gene effect of *TP53* (Letter Fig. 4f) and *CDKN1A* (Letter Fig. 4g) across tumour types (note that the gene effect scores of *TP53* and *CDKN1A* across most tumour types are positive, indicating that their loss supports cell survival). This suggests that *NPM1* becomes more essential — that is, *NPM1* loss suppresses cell proliferation — in contexts where *TP53* and *CDKN1A* are indeed the rate limiting step of proliferation. This is in agreement with our own murine data showing that *Npm1* deletion suppresses proliferation and tumourigenesis in tissues with intact *Trp53/Cdkn1a*.

Moreover, in bowel cancer cell lines dependency on *NPM1* is particularly strong when *APC* is mutated and *TP53* remains intact (Letter Fig. 4h), supporting our conclusion that *NPM1* dependency is modulated by both WNT pathway activation and functional p53. This is also consistent with the human CRC sample data, which show that *NPM1* deletions are predominantly observed (thus tolerated) only when *TP53* is inactivated (Fig. 3r-t).

Letter Figure 4. **a, b**, GSVA score for WNT signatures *a* and *b* across tumour types in DepMap cell line data. **c**, CRISPR gene effect and **(d)** expression of NPM1 across tumour types. **e**, Cancer types sorted by mean NPM1 CRISPR gene effect difference between TP53 wild-type (0) and TP53 mutant (1) cell lines. P-values shown above boxes determined by two-sided t-test. **f, g**, Scatterplots showing the correlation between the CRISPR gene effect of NPM1 and TP53 (**f**) or CDKN1A (**g**). Statistical assessment performed by two-sided Pearson correlation. Correlation coefficient (*r*) is displayed to indicate the degree of association. **h**, NPM1 gene effect in bowel cancer cell lines with intact (0) or mutated (1) APC and/or TP53. P-values shown above boxes determined by two-sided t-test.

5. Lastly, throughout the paper many claims are made regarding the specificity of Npm1 loss being a vulnerability in Wnt-high cancer, but no “Wnt-low” tumor controls were used throughout the study. Additional in vivo experiments or analysis need to be performed with “Wnt-low” controls.

This is an important point raised by the reviewer. To address the specificity of NPM1 loss in the context of high WNT activity we performed several additional experiments and analyses. Firstly, to evaluate the tissue specificity of WNT activation and functional importance of the enhanced *Npm1* expression, we co-deleted *Apc* and *Npm1* across mouse tissues using the Rosa-Cre system. These were compared to *Apc* deletion alone as well as to wild-type control tissue (Extended Data Fig. 3). As expected tissues along the gastrointestinal tract, particularly the intestine and liver, showed the strongest proliferative responses to APC loss, consistent with their WNT-permissive nature. Importantly, these were also the tissues that NPM1 loss most effectively suppressed proliferation, supporting the link between WNT activity and NPM1 dependency. In addition, we show that *NPM1* expression positively correlates with WNT activation levels in human CRC samples as well as across multiple tumour types (Fig. 1f,g; Extended Data Fig. 1d,e; Extended Data Fig. 10c-g). We also provide evidence that multiple murine models of CRC not driven by *Apc* deletion do not exhibit *Npm1* upregulation (Fig. 2a), nor do endpoint tumours driven by oncogenic BRAF (Extended Data Fig. 5a-c). NPM1 loss does not suppress transcriptional activation of *Anxa1* in the colon following oncogenic BRAF activation (Extended Data Fig. 5d,e), nor BRAF-driven hepatocyte proliferation (Extended Data Fig. 10r-u). Finally, we examined murine models of CRC driven by *Kras* and *Trp53* mutations without APC loss and found that NPM1 depletion does not provide a survival benefit (Extended Data Fig. 7f-i). Together, these data provide multiple lines of evidence that support NPM1 dependency downstream of high WNT activity, and strengthen the conclusion that NPM1 functions as a WNT effector in this context. However, while our findings support this specificity, we agree that NPM1 may also be important in other cancers, particularly in highly proliferative tumours that rely on increased biosynthetic and translational capacity.

Reviewer #3:

Remarks to the Author:

In this article Kanellos et al, provide evidence suggesting that NPM1 is induced upon APC loss in WNT signaling-responsive tissues. The authors further show that abrogation of NPM1 does not inconspicuously affect most of the adult tissues, but that it prolongs survival in WNT-driven as CRC (+/- KRAS) and HCC (+/-MYC) models. Some evidence is also provided suggesting that NPM1 loss leads to induction of ISR and activation of p53/p21-dependent pathways, whereby abrogation of ISR lead to apparent decrease in p21 levels. Overall, it was thought that this article is of high potential interest as it may establish the role of NPM1 in solid tumors (e.g., beyond AML), while also proving previously unappreciated insights in oncogenic WNT signaling. To this end, it was thought that this article is likely to contribute significantly to improved understanding of the molecular underpinnings of HCC and CRC, and thus to be of broad interest to researchers from various biomedical disciplines ranging from cell and developmental to cancer biology. Overall, it was thought that most of the authors conclusions are adequately supported by the data, whereby the methodology was found to be sound in general. Notwithstanding these clear strengths of the report, several addressable issues mostly related to the lack of mechanistic data to support proposed model were raised. My specific comments are outlined below:

We sincerely thank the reviewer for their thorough and thoughtful evaluation of our manuscript. We greatly appreciate the recognition of the potential multidisciplinary significance of our findings regarding the role of NPM1 in WNT-driven tumorigenesis, as well as in cell and developmental biology, and the constructive comments. Please see below for our responses to the specific points raised.

Major concerns:

1. Although several possibilities were discussed, it was thought that there is a lack of the mechanism whereby NPM1 impinges on p53 axis in the context of author's model. This may be of interest as NPM1 abrogation seemed to result in minimal changes in the transcriptome (changes are mostly in the translome), whereby it would be anticipated that these would be profound based on the perturbations in p53 function. To address this, it was stated that "Activation of p53 Following loss of NPM1 is subtle enough to avoid triggering a global p53 transcriptional program and permit homeostatic proliferation, while only affecting the increased proliferation levels required for oncogenic transformation", but having the experimental evidence how this occurs would further strengthen the applicant's model. To this end, notwithstanding that the previous literature was discussed, it may be of the interest to authors to further delineate the interplay between NPM1 and p53 in the context of dysregulated WNT signaling in CRC and HCC. This was thought to be of particular importance as many of different post-transcriptional and post-translational mechanism of p53 have been described in the literature. I appreciate that the effects of NPM1 loss on TP53 transcription or mRNA stability were largely excluded (via RNAseq data). I don't mean to torture the authors by making them conduct some extraordinarily complicated experiments as these may invariably result in leading them into rabbit holes, but few simple experiments such as looking at subcellular localization, interaction with HDM2/stability of p53 protein would be informative. Out of these types of experiments and sequencing data, it should be relatively trivial to identify the potential culprits. Of note, did TP53 mRNA pass the threshold in Ribo-seq arm of the study?

We appreciate the reviewer's comments and share his concerns that further clarification of the interplay between NPM1 and p53 would strengthen our model. As discussed above in reviewer's #2 second comment, we were reluctant ourselves to accept the lack of transcriptional changes, especially so given the importance of p53 downstream of NPM1 loss. Our genetic data clearly demonstrate that *Trp53* deletion fully rescues the phenotypes elicited by *Npm1* loss, underscoring the functional importance of p53 in this context. As noted above the newly included pausing site analysis from the ribosome profiling data shows that *Trp53* and *Cdkn1a* (p21) transcripts are notably spared from ribosome pausing (Extended Fig. 8v). In the case of *Trp53*, we even observed a resolved pause site (Extended Fig. 8w), suggesting more efficient translation in the absence of NPM1. Additionally, we observed increased γ H2AX staining following *Npm1* deletion (per Reviewer #2's suggestion), which could contribute to low-level p53 activation. While we tried to assess p53 subcellular localization, this did not yield conclusive results, and we were unable to validate MDM2 expression or localization in tissue sections due to technical limitations with available antibodies.

We appreciate that additional mechanistic insight would be valuable. It has long been recognised that ribosomopathies often lead to p53 activation, and *TP53* is frequently mutated to circumvent the deleterious effects of such stress. Although our data do not indicate major deregulation of ribosomal RNA precursors, the possibility remains that NPM1 loss (given its role as a major nucleolar component) triggers mild nucleolar stress, activating a nucleolar surveillance pathway and stabilizing p53. This would explain the subtle activation of the p53 pathway under homeostatic conditions, which becomes significantly more detrimental in the context of oncogene-driven proliferation. Furthermore, as discussed in the manuscript, NPM1 has been identified as a component of actively translating ribosomes^{7,8}. Future studies will benefit from further dissecting its role in ribosome function and translation regulation. While beyond the scope of the current study, these will be crucial for fully elucidating the post-transcriptional mechanisms underlying p53 activation following NPM1 depletion. To conclude, while we acknowledge that additional experiments would be needed to fully elucidate the mechanistic details, our current data provide strong support for a selective, translational mechanism of p53 activation upon NPM1 loss.

2. Notwithstanding that it is reasonable to propose that ISR may limit hyperproliferative phenotypes in e.g., intestine, when the tumor grows and outstrips the vasculature, ISR may play a protective role by allowing cancer cells to adapt to limited amounts of oxygen and nutrients. There is some evidence that ISR may have antitumorigenic activity in the experiments using trazodone and monitoring survival of the mice in the supplementary figure 6i, but the trazodone is quite pleiotropic and it is hard to pin them on ISR. Also, I could not find controls showing that phospho-eIF2alpha is indeed downregulated in these mice throughout tumor evolution nor information on tumor size in this experiment). Moreover, trazodone treatment was started quite early (5 days after induction), so these effects are likely to be illustrative of the effects of the drug on the initial tumor growth and not tumor maintenance wherein the ISR may have a protective role. It may thus be advantageous to investigate this potential antagonistic pleiotropy of ISR in the context of neoplasia by monitoring not only proliferative indexes, but also tumor growth and progression under conditions where ISR is continuously disrupted in the cells where NPM1 is deleted by either pharmacological or genetic (S51A eIF2alpha KI) approaches, and also disrupting ISR after tumors have already grown to certain size to investigate the effects on tumor maintenance. This would also be of importance to establish the role of proposed mechanisms during tumor

evolution. Alternatively, the authors should consider limiting their conclusion to reduction of proliferation rather than tumor growth/progression.

We fully agree with the reviewer that the ISR may play context-dependent roles during tumour evolution, acting as a brake on proliferation in early stages while later supporting tumour cell adaptation to stress conditions. We also agree on the pleiotropic nature of trazodone, and we have acknowledged this limitation in both the manuscript and our response to Reviewer #1. We have also included the tumour data for the trazodone long-term experiment now in Extended Data Fig. 9r. We apologise for its omission in the first version of the manuscript.

As the reviewer correctly points out, the timing of ISR inhibition is likely to have differential effects. In fact, we now include new data showing that ISRIB treatment in established tumours of *Apc^{fl/+}Kras^{G12D/+}* animals significantly affects tumour cell proliferation (Fig. 6i,j), in contrast to its limited effect in short-term proliferation (Fig. 6d). On the contrary, in NPM1-deficient tumours, ISR inhibition with ISRIB restores proliferation to levels comparable to untreated NPM1-proficient controls, both in the early and established tumour settings. This suggests that while the ISR is supporting growth of established tumours in our model, following NPM1 loss it plays a critical role in mediating the anti-proliferative effects in both early proliferation as well as tumour maintenance.

Regarding phospho-eIF2 α levels, ISRIB acts downstream of eIF2 α phosphorylation and does not necessarily reduce phospho-eIF2 α itself. As such, we observed no significant changes in p-eIF2 α levels in ISRIB-treated established tumours or in intestinal crypts (Fig. 6k,l).

Lastly, we would like to clarify our arguments regarding tumour growth/establishment in the absence of NPM1 as presented in the original version of the manuscript. Our interpretation was based on the fact that most tumours arising after *Apc* deletion retain NPM1 expression, suggesting that NPM1 is required for tumour initiation unless additional oncogenic drivers, such as *Kras* activation and/or *Trp53* inactivation, are present to overcome this dependency. In the revised manuscript we have refrained from references around ‘tumour growth’, as this was not directly assessed, and instead focus specifically on tumour establishment/initiation.

3. It remains rather unclear how ISR is triggered upon NPM1 loss. The authors provide some evidence using PS that elongation may be stalled at the 5' ends of main ORFs, but it is not clear how the conclusion of imbalance of initiation and elongation rates was derived, as at least to me it remains unclear how the changes in the initiation rates were inferred? Notably, in the “classical” ISR the initiation rates should be strongly reduced. Is global protein synthesis reduced under these conditions? Based on this, perhaps the authors should consider that the response to NPM1 loss is distinct from “classical” ISR and constitutes a hitherto unappreciated translational reprogramming? This may be quite worthy of considering because mTORC1 appears to be simultaneously upregulated (and considering that this is KRAS driven model, one would expect phosphorylation of eIF4E also to go up) and thus, NPM1 loss may result in high eIF4F/low TC scenario that is likely to be distinct from the “classical” ISR. One of the key experiments that appears to be missing here is monitoring the levels of global protein synthesis +/- e.g., ISRIB as this may allow authors to distinguish between their observations and “classical” ISR. Another possible scenario may also be derived from this relatively simple experiment, which is that if the protein synthesis is upregulated in these tumors, induction of eIF2 α phosphorylation may be secondary to overload of ER and activation of PERK and thus although pro-proliferative in the initial

stages, might actually be an adaptive mechanism in fully grown tumors where the cells are exposed to stress (please see the comment above). Although protein synthesis was unchanged upon NPM1 loss in normal cells, considering that these models are driven by KRAS or MYC and that at least in the KRAS model mTORC1 seems to be hyperactive, it would not be unreasonable to expect that protein synthesis may be altered in this context.

We thank the reviewer for these insightful comments and suggestions. We apologise for the confusion, but we did not intend to imply changes in the actual rates of initiation or elongation. Rather, our original speculation referred to a translation imbalance following NPM1 loss at the pathway level, driven by seemingly competing signals with increased translation initiation cues (based on elevated p-4E-BP1 and now also p-eIF4E; Extended Data Fig. 9a-d), alongside signals that may suppress elongation (via increased p-eEF2), and an accumulation of ribosomes at the 5' ends of main ORFs. To avoid confusion, we have removed all references to specific initiation/elongation 'rates' from the manuscript. Instead, we now refer more generally to 'imbalanced translation regulation' based on the signalling context observed.

In response to the reviewer's suggestion, we have assessed global protein synthesis in NPM1-proficient and deficient *Apc^{fl/fl}Kras^{G12D/+}* organoids. Despite increased phosphorylation of translation initiation factors, protein synthesis is reduced in the absence of NPM1 (Extended Data Fig. 9e). However, unlike the canonical ISR, we do not observe induction of classical ISR downstream targets such as ATF4, CHOP, or GADD34 (Extended Data Fig. 9i-l). In fact, ATF4 levels are slightly reduced following NPM1 loss, and global protein synthesis is not restored by ISRIB treatment in NPM1-deficient organoids (Extended Data Fig. 9m,n). Notably, *Npm1*-null organoids retain the capacity of eliciting a classical ISR induced upon tunicamycin treatment (Extended Data Fig. 9k,l), suggesting that it is potentially not activated in the canonical manner upon NPM1 loss. Interestingly, in this context, induction of p53 and p21 were suppressed. Taken together, these data support the reviewer's suggestion that the response to NPM1 loss may represent a previously unappreciated, non-canonical translational stress response that is distinct from the classical ISR. We now note this in both the Results and Discussion sections of the revised manuscript. We agree that further dissection of this mechanism represents an important direction for future study.

4. Related to the above, validating at least some of the mRNAs and corresponding proteins identified to be differentially translated upon NPM1 loss seems to be warranted (looking at total mRNA and protein levels would suffice, or considering the complexity of the model even IHC for some of the targets alone would be sufficient). Also, reporting what happens to translation (from Ribo-seq data) and protein levels (e.g., by IHC) of "classical" ISR targets (e.g. ATF4, CHOP, GADD34) may be informative.

As already mentioned above, we have now removed all Translational Efficiency-based analyses and replaced them with analyses based solely on RPF data. To complement the Ribo-seq dataset, we have now incorporated proteomic data generated from the same biological samples, and show inverse enrichment of certain pathways at the translational and protein levels. Furthermore, as outlined in response to the above comment, we have also assessed the expression of the classical ISR targets mentioned in both organoids and epithelial extracts used to perform the Ribo-seq experiment (Extended Data Fig. 9i-l). As discussed, we do not observe increased expression of these ISR components upon NPM1 loss, which supports the idea that the observed translational stress response is distinct from the classical ISR.

Minor comments:

-GSK2606414 shows of target effects (PMID: 30931942) and as for the other similar inhibitors, it is expected that it will induce other eIF2alpha kinases in a concentration dependent manner (PMID: 37684277). Albeit this concern was largely mitigated by the experiments employing ISRIB, the authors should I think make a note of this to avoid further confusion in the literature.

This is a great point raised by the reviewer. We have revised the relevant Results section of the manuscript to include this clarification and have cited the relevant studies to help avoid potential confusion.

-Using log-based ratios to determine translation efficiency comes with the perils of spurious correlation (<https://royalsocietypublishing.org/doi/abs/10.1098/rspl.1896.0076>), whereby methods based on the analysis of partial variance are likely to be superior (e.g., PMID: 21115840).

We thank the reviewer for raising this important point. In response, we tested the Anota2seq (A2S) pipeline on our dataset. We have now excluded any form of analyses downstream translation efficiency as none of the genes was considered significantly altered neither by our analysis nor by A2S. Since our focus in functional analyses was on the genes ranked by RPF differences, we checked whether our own DEseq2-based analysis differed significantly from the A2S one. As it can be appreciated by the two scatter plots in Letter Fig. 5, correlation between A2S and DEseq2 results is extremely high, particularly so for the RPFs. We therefore proceeded to keep the results as analysed by our pipeline in the revised manuscript.

Letter Figure 5. Scatter plots visualising the correlation between Anota2seq and DEseq2 based analysis of total cytoplasmic RNA (left) and RPFs (right).

-Line 220 “Notably, the surrounding normal epithelium was negative for NPM1 expression, suggestive of its importance for tumour initiation”. The authors should clarify this, as it is not completely clear to me why the loss of NPM1 in surrounding tissue would be indicative of NPM1 role in tumor initiation. I found this statement to be relatively confusing as across most of the models the NPM1 status does not appear to affect the number of tumors except in the KRAS/APC model, albeit NPM1 +ve escapers appears to be comparable across all the models.

We have revised the relevant sentence to now read: '*Npm1* deletion significantly extended survival at clinical endpoint, although most tumours escaped recombination and retained NPM1 expression (Fig. 2d-g; Extended Data Fig. 5k). Notably, the surrounding normal epithelium maintained *Npm1* deletion, suggesting a tumour-specific requirement for NPM1 during initiation.'

Our rationale is that the persistence of *Npm1* deletion in the surrounding normal epithelium, alongside the frequent retention of NPM1 expression in tumours, suggests that NPM1 is specifically required for tumour establishment, rather than for the maintenance of normal epithelial tissue. This points to a tumour-selective dependency on NPM1. As also discussed in response to reviewer #1, although the total number of tumours does not differ significantly across groups at endpoint, spatial analysis of tumour distribution (Extended data Fig. 5k) reveals a significant suppression of tumour formation in intestinal regions that are typically most prone to transformation. Thus, our emphasis is not solely on tumour count, but rather on the fact that tumours must retain NPM1 expression to establish. As the reviewer points out, when KRAS activation enables them to bypass this dependency, overall tumour numbers do remain lower in the absence of NPM1.

-The authors should consider indicating the localization of the tumors throughout the article (i.e., do they form in small intestine, colon...?).

Following the reviewer's suggestion, tumour distribution data along the intestinal tract for animals sampled at clinical endpoint is now provided as part of the Extended Data.

-Figure 5b covers about ½ of protein encoding genes, and this should perhaps be noted as there may still be genes that were regulated in the other ½ of the protein coding genes that were likely below the detection level.

Indeed, during differential expression (DE) analysis we conventionally exclude genes with raw reads <10 average across samples. We have now clearly stated this in the method sections. "Any transcript with average reads across all samples lower than 10 was excluded from DE analysis and downstream processing. Of the 20,622 transcripts in the most abundant transcript (MAT) table, 10,861 were retained for cytoplasmic RNA, and 11,279 were retained for RPFs. Merging the data tables to evaluate changes at RPF and cytoplasmic RNA concomitantly retains 10,261 transcripts".

We tested the outcome of no pre-filtering on DEseq2. Unfiltered DEseq2 results for RPFs has data for 16,737 transcripts, while for totals has 20,622, which means all the transcripts from the MAT are kept. Inner joining them to evaluate changes at both levels has data for all the 16,737 genes from the RPFs set. Plotting these data is extremely noisy (left scatter). Therefore, another option is to plot what has a numerical adjusted p-value in RPFs or totals as calculated by DEseq2 (i.e. excluding genes which have padj NA in both RPFs and totals) (right scatter). This would exclude the noisy transcripts. By doing so we are left with data for 5,385, therefore further halving what we are currently analysing. We therefore consider that our current analysis is stringent enough but not excessively so.

Letter Figure 6. Scatter plot visualising changes in NPM1-proficient and deficient *Apc^{fl/fl}Kras^{G12D/+}* samples at both total RNA and RPFs levels. The left panel includes all 16,737 genes from the unfiltered DESeq2 results (RPFs), while the right panel shows the 5,385 genes with a numerical adjusted p-value in either the RPF or total RNA dataset, as calculated by DESeq2.

-In Supp. Figure 5 the authors should consider adding additional QC data typically used for Ribo-seq (e.g., correlations between the replicates, reading frames/riboWaltz), and provide more detail in methodology regarding applied batch correction. Finally, the author may consider comparing their translational signatures to publicly available translational signatures (e.g., for ISR induction). This may be of interest as it is a bit surprising that there are no perturbations in the total mRNA levels which usually are quite perturbed under ISR (e.g., via ATF4 activation), and thus the stress response observed upon NPM1 loss may be novel and distinct from “classical” ISR (also see the comment # 3 under “major comments” section

In response to the reviewer, we have now included correlation plots between the replicates for both cytoplasmic RNA sequencing and RPF datasets (Extended Data Fig. 8h,i). We observe a very high correlation across our whole dataset for all samples. Analysis of reading frames, as well as read lengths and coverage of coding sequences vs UTRs are now moved to Extended Data Fig. 8a-e. Regarding batch correction, additional details have now been provided in the Methods section.

As discussed in response to earlier comments, we agree that the lack of changes in total mRNA levels is indeed striking and may reflect a stress response that is distinct from the canonical ISR. To explore this further, we investigated whether in our context we also detect a so-called split-ISR, similar to the one described in your recent study with the Larsson and Hatzoglou labs, by extracting the genes belonging to the GO enrichment terms Up or Downregulated upon shEIF2B5 (FDR <1e-2) (Supplementary table 2)⁹. Visual inspection of the distribution of these genes in our data (Letter Fig. 7) suggests that the ISR program activated upon *Npm1* deletion is further distinct from the split-ISR, supporting the notion that this represents a novel mode of stress adaptation and highlighting it as an important direction for future mechanistic investigation.

Letter Figure 7. Scatter plot visualising changes in NPM1-proficient and deficient *Apc^{fl/fl}Kras^{G12D/+}* samples at both total RNA and RPFs levels, with genes belonging to the GO enrichment terms Up or Downregulated upon *shEIF2B5* (FDR < 1e-2)⁹ highlighted in colour.

-The authors should consider noting which phosphoacceptor sites on proteins were tested in figures and legends.

This information has now been incorporated in all relevant figures and figure legends.

-It may be pertinent to perhaps look at the morphology of the nucleoli or indications of nucleolar stress in NPM1-abrogated vs. control cells, especially considering implication of p53.

In response to the reviewer's comment we have performed a comprehensive analysis on nucleolar size, number and circularity, in NPM1-depleted intestinal cells compared to controls (**Extended Data Fig. 4h-q**). We observe a modest reduction in nucleolar circularity following NPM1 loss in wild-type intestine, however, this is not significant following oncogene activation. However, given NPM1's roles in nucleolar structure and function, we cannot exclude the possibility that its loss induces mild nucleolar stress, which may activate the nucleolar surveillance pathway and contribute to p53 stabilization. Nonetheless, the robust rescue of the phenotypes and suppression of p53 induction by ISR inhibition with multiple compounds, suggests that even if mild nucleolar stress occurs, it is likely peripheral to the dominant translational stress response triggered by NPM1 loss.

I hope that the authors will find my comments constructive and of sufficient pathos.

Sincerely

I/Topisirovic

References

1. Halliday, M. *et al.* Repurposed drugs targeting eIF2 alpha-P-mediated translational repression prevent neurodegeneration in mice. *Brain* **140**, 1768-1783 (2017).
2. Halliday, M. *et al.* Partial restoration of protein synthesis rates by the small molecule ISRIB prevents neurodegeneration without pancreatic toxicity. *Cell Death & Disease* **6**(2015).
3. Moreno, J.A. *et al.* Oral Treatment Targeting the Unfolded Protein Response Prevents Neurodegeneration and Clinical Disease in Prion-Infected Mice. *Science Translational Medicine* **5**(2013).
4. Vin, H. *et al.* BRAF inhibitors suppress apoptosis through off-target inhibition of JNK signaling. *Elife* **2**(2013).
5. Mathea, S. *et al.* Structure of the Human Protein Kinase ZAK in Complex with Vemurafenib. *Acs Chemical Biology* **11**, 1595-1602 (2016).
6. Silva, J. *et al.* Ribosome impairment regulates intestinal stem cell identity via ZAKalpha activation. *Nature Communications* **13**(2022).
7. Bartsch, D. *et al.* mRNA translational specialization by RBPMS presets the competence for cardiac commitment in hESCs. *Science Advances* **9**(2023).
8. Simsek, D. *et al.* The Mammalian Ribo-interactome Reveals Ribosome Functional Diversity and Heterogeneity. *Cell* **169**, 1051-+ (2017).
9. Chen, C.W. *et al.* Plasticity of the mammalian integrated stress response. *Nature* **641**(2025).